# Wastewater-based epidemiology predicts COVID-19-induced weekly new hospital admissions in over 150 USA counties

Xuan Li[1], Huan Liu[1], Li Gao[2], Samendra P. Sherchan[3,4], Ting Zhou[1], Stuart J. Khan [5], Mark C. M. van Loosdrecht [6] & Qilin Wang [1]✉

Although the coronavirus disease (COVID-19) emergency status is easing, the COVID-19 pandemic continues to affect healthcare systems globally. It is crucial to have a reliable and population-wide prediction tool for estimating COVID-19-induced hospital admissions. We evaluated the feasibility of using wastewater-based epidemiology (WBE) to predict COVID-19-induced weekly new hospitalizations in 159 counties across 45 states in the United States of America (USA), covering a population of nearly 100 million. Using county-level weekly wastewater surveillance data (over 20 months), WBE-based models were established through the random forest algorithm. WBE-based models accurately predicted the county-level weekly new admissions, allowing a preparation window of 1-4 weeks. In real applications, periodically updated WBE-based models showed good accuracy and transferability, with mean absolute error within 4-6 patients/100k population for upcoming weekly new hospitalization numbers. Our study demonstrated the potential of using WBE as an effective method to provide early warnings for healthcare systems.

The coronavirus infectious disease (COVID-19) has created a severe public health crisis globally. During the peaks of the pandemic in the United States of America (USA), COVID-19 infections overwhelmed healthcare systems in most states, occupying up to 90% of their capacity[1]. Unexpected and heavy burdens from COVID-19 exhausted frontline healthcare workers in 60–75% of hospitals or clinics[2], subsequently leading to increased fatality rates[3]. Even in recent months (December 2022 to February 2023), COVID-19-induced hospitalizations still occupied an average of 10–20% of beds in healthcare systems in many counties, and up to 60% in some counties[4]. Reliable predictions of hospitalization numbers are thus crucial for adequate public health decision-making and evaluation, and healthcare system preparedness.

To date, the prediction of hospitalization admissions due to COVID-19 is majorly at the state or national level, relying on confirmed COVID-19 cases or historical records of daily or weekly COVID-19-induced admissions as the key indicators[5,6]. However, with the end of the COVID-19 public health emergency in many countries, changes in test availability, behavior, and reporting strategies reduced the certainty of COVID-19 infection numbers, especially for asymptomatic infections[1]. In addition, clinical testing may only capture a portion of the true infections in the community due to factors such as insurance coverage, individual willingness to be tested, and socioeconomic status in the area[7,8]. In clinical settings, it is common that some patients have been admitted to hospitals before obtaining positive COVID-19 tests[5]. Ensembled probabilistic forecasts for daily incident hospitalizations were also provided based on the forecast from multiple teams at state and national levels[9]. However, hospitalization rates and patterns can vary significantly at the county level due to differences in

[1]Centre for Technology in Water and Wastewater, School of Civil and Environmental Engineering, University of Technology Sydney, Ultimo, NSW 2007, Australia. [2]South East Water, 101 Wells Street, Frankston, VIC 3199, Australia. [3]Department of Biology, Morgan State University, Baltimore, MD, USA. [4]Department of Environmental Health Sciences, School of Public Health and Tropical Medicine, Tulane University, New Orleans, LA, USA. [5]Water Research Centre, School of Civil and Environmental Engineering, University of New South Wales, Sydney, NSW 2052, Australia. [6]Department of Biotechnology, Delft University of Technology, Julianalaan 67, 2628 BC Delft, the Netherlands. ✉e-mail: Qilin.Wang@uts.edu.au

population demographics, healthcare resources, etc., even within the same state[10]. More granular insights for predicting hospitalization at county-level are more ideal for practical application.

Wastewater-based epidemiology (WBE) is considered an efficient approach for COVID-19 case surveillance, providing unbiased infection estimations at the community level with limited cost (0.7–1% of the population-wide testing)[11–14]. Many studies have successfully quantified and correlated SARS-CoV-2 concentrations ($C_{RNA}$) in wastewater to COVID-19 cases[11–13,15]. Few studies have reported the association between $C_{RNA}$ in wastewater (or primary sludge) with hospitalizations[16,17] and endeavored to create surveillance models for forecasting hospital admissions with various leading times ranging from 1 to 8 days[18–20]. Nevertheless, these observations and models were developed using data from only a few localities for a short period (a couple of months). A recent study revealed the predictive potential for state-level hospitalization occupancy (census hospitalizations) with a leading time of 8–18 days in Austria[21]. However, population demographics (such as race/ethnicity, vaccination, chronic conditions, etc.) that have been clinically observed impacting the COVID-19 symptom severity[22–27] were not considered in all these precedent prediction models. This limits their temporal and geographic scope, thus making it uncertain whether they (both the model and the hospitalization indicators predicted) could be generalized to other areas. Considering that hospitals/healthcare facilities often allocate their resources and workers on a weekly basis for upcoming patients[28], a large-scale (temporal and geographic) prediction system for hospitalizations at the county level on a weekly basis would be more informative for local healthcare facilities, which unfortunately is lacking.

In this work, we used county-level weekly WBE data from the recent 20 months (June 2021 to January 2023) covering 159 counties from 45 states in the USA (Fig. 1) with their corresponding county-level hospital admission records, vaccination records, and weather conditions. The county-level population demographics were incorporated from COVID-19 Community Vulnerability Index (CCVI)[23,29], which is in use by the Centers for Disease Control and Prevention (CDC), for easy-adaption and transfer in different regions. Random forest models were established using these factors to predict the county-level

hospitalization indicators over the course of the upcoming week, as well as the second, third, and fourth weeks after the wastewater sampling to address the following: (1) The feasibility of using WBE for predicting hospital admission numbers in healthcare systems: which hospitalization indicator can be predicted by WBE-based prediction and how accurate are the predictions in comparison to the current approaches (cases-based prediction and record-based predictions)? (2) The contribution of CCVI indexes, vaccination, and weather factors for the prediction: how are they affecting the WBE-based prediction? (3) For real applications, is a periodic update of the model necessary? (4) The transferability of the models to other counties and states: how accurate is the model prediction for other counties and how to improve the accuracy? (Fig.1). Our results would help improve the preparedness of healthcare systems and vulnerable counties in the USA in coping with the COVID-19 pandemic or endemic.

## Results
### Geographic, socioeconomic, and epidemiological characteristics of the counties involved in the model establishment
The 99 counties involved in the model establishment (Fig. 2a) covered 40 states in the USA, with 1–8 counties involved in each state. The population size in each county ranged from 0.02 to 3.4 M (Supplementary Table S1), covering nearly 60 M population in total. The CCVI indexes in these counties ranged from 0.02 to 0.99, which are representative of most USA counties (Supplementary Fig. S1)[23,29]. Most of the counties (interquartile range, IQR) had the overall VI at 0.31–0.74, CCVI in socioeconomic status at 0.25–0.65, minority and language at 0.57–0.89, household and transportation at 0.23–0.60, epidemiological factors at 0.15–0.35, healthcare system at 0.22–0.61, high-risk environment at 0.28–0.63, and population density at 0.80–0.96 (Fig. 2b).

Three indicators for hospitalization numbers were used including: 1) weekly new admission, 2) total number of patients who stayed in an inpatient bed during the week (census inpatient sum), and 3) daily average number of patients who stayed in an inpatient bed in the week (census inpatient average). The weekly new admission, census inpatient sum, and census inpatient average had a range of 0–100 patients/100k population, 0–1220 patients/100k

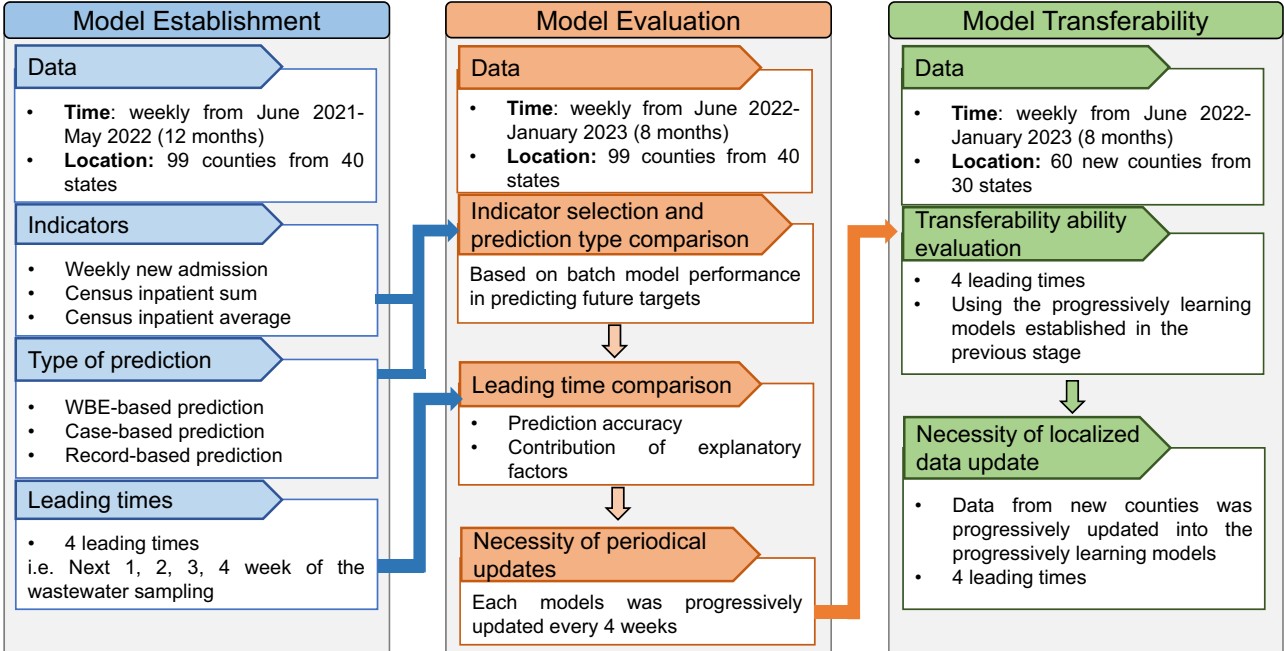

**Fig. 1 | Flow chart of the paper methodology, process, and structure.** The data, indicators, leading times, and prediction approaches used for model establishment, evaluation, and transferability stage in this study.

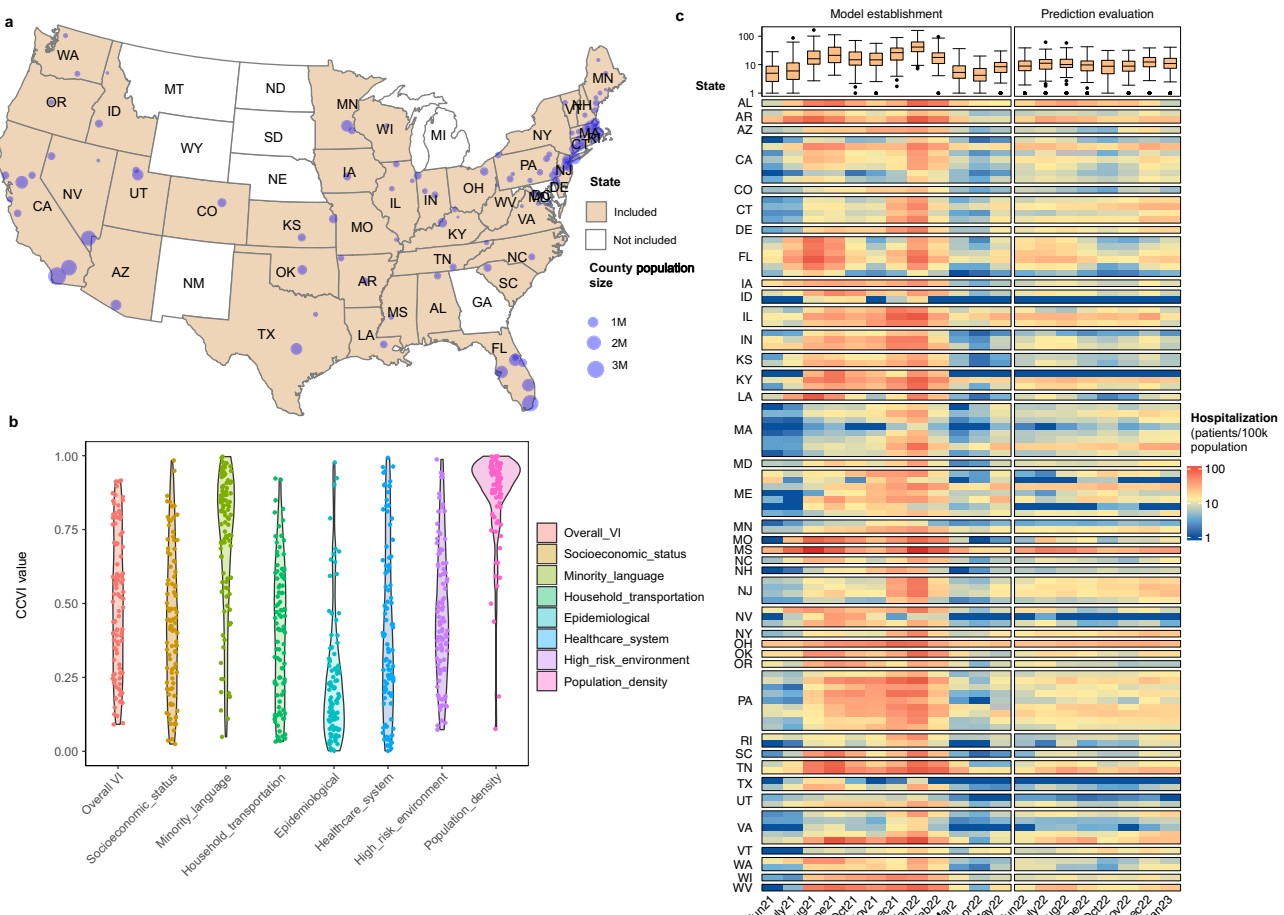

**Fig. 2 | Geographic location, COVID-19 Community Vulnerability Index (CCVI), and average weekly new COVID-19-induced hospitalizations in each month in the 99 counties involved. a** The state (filled in color) and counties (indicated by the dot with dot size reflecting the population size of the county) involved in the model establishment. The data for the map was obtained from 'USmap' package in R, where the shape data was provided by the USA Census bureau. **b** The CCVI of the counties involved in the model establishment is represented by a violin plot for each index. **c** The average weekly new hospitalization admission numbers of each month from these 99 counties. The data before June 2022 (12 months) were used for model establishments while data after June 2022 (8 months) were used for model evaluation. In the box plot (top of subplot **c**), the colored box indicates the 25th and 75th percentiles, and the line in the box indicates median. The whiskers represent 1.5× the interquartile range and dots indicate outliers. *N* = 99 for each box.

population, and 0–175 patients/100k population, respectively. The highest peaks were observed during August 2021 to February 2022 (Fig. 2c, Supplementary Fig.S2). The $C_{RNA}$ of wastewater samples ranged from 0.4 to 9000 copies/mL (IQR:101.54–546.53 copies/mL) (Supplementary Fig. S2). The weekly new COVID-19 cases ranged from 0 to 4065 incidence/100k population (IQR: 48–271 incidence/100k population). The hospitalization indicators and $C_{RNA}$ were skewed to higher ranges (Supplementary Fig. S2), which is consistent with the inherent development of the outbreak. The ratio of vaccinated people among the population in these counties increased from 4.5–84.8% (IQR: 45.2–61.9%) in June 2021 to 42.7–95.0% (IQR: 71.2–92.5%) in January 2023 for the first does (Vaccine_1st). Meanwhile, the ratio of vaccinated people among the population for the second dose (Vaccine_2nd) increased from 4.0–69.6% (IQR: 39.3–69.6%) in June 2021 to 37.9–92.5% (IQR: 63.15–77.2%) in January 2023. The major vaccines used were Pfizer/BioNTech and Moderna during the time of the study. The average daily air temperature ($T_a$), average daily precipitation, ('precipitation' hereafter), and average daily wastewater temperature ($T_w$) were −16.5–32.7 °C (IQR: 4.0–22.1 °C), 0–1.1 mm (IQR: 0.1–0.2 mm), and 7.3–32.8 °C (IQR: 15.1–29.9 °C), respectively. At the national level, the major variants of COVID-19 shifted progressively from Alpha and Beta in June 2021 to Delta in June–November, 2021 and

Omicron in December 2021 to April 2022 with 33 different lineages occurred (Supplementary Fig. S3).

## Correlations between explanatory factors and hospitalization indicators for model establishment

The records of these three hospitalization indicators over the course of the upcoming week (Hos1w), as well as the second (Hos2w), third (Hos3w), and fourth weeks (Hos4w), were used as prediction targets, providing prediction leading times of 1, 2, 3 and 4 weeks, respectively. Under the same leading time, $C_{RNA}$ exhibited slightly stronger correlations with weekly new admissions (Hos_wn, R = 0.47–0.61), than census inpatient average (Hos_ca) and census inpatient sum (Hos_cs) (R = 0.46–0.56). Notably, the correlation between $C_{RNA}$ and the targets in the first two weeks (Hos1w and Hos2w) was higher than that in the 3rd and 4th weeks (Hos3w and Hos4w) for the same type of indicator (Fig. 2). This suggests that the predictive performance of WBE may differ for various indicators and at different leading times. Other explanatory factors, including population size, and factors associated with vaccination, CCVI, and the weather showed significant correlations (|R| of 0.1–0.4) with at least one of the targets (Fig. 3a). Considering the randomness of random forest algorism (see Methods), all these 15 explanatory factors were used for establishing WBE-based prediction models.

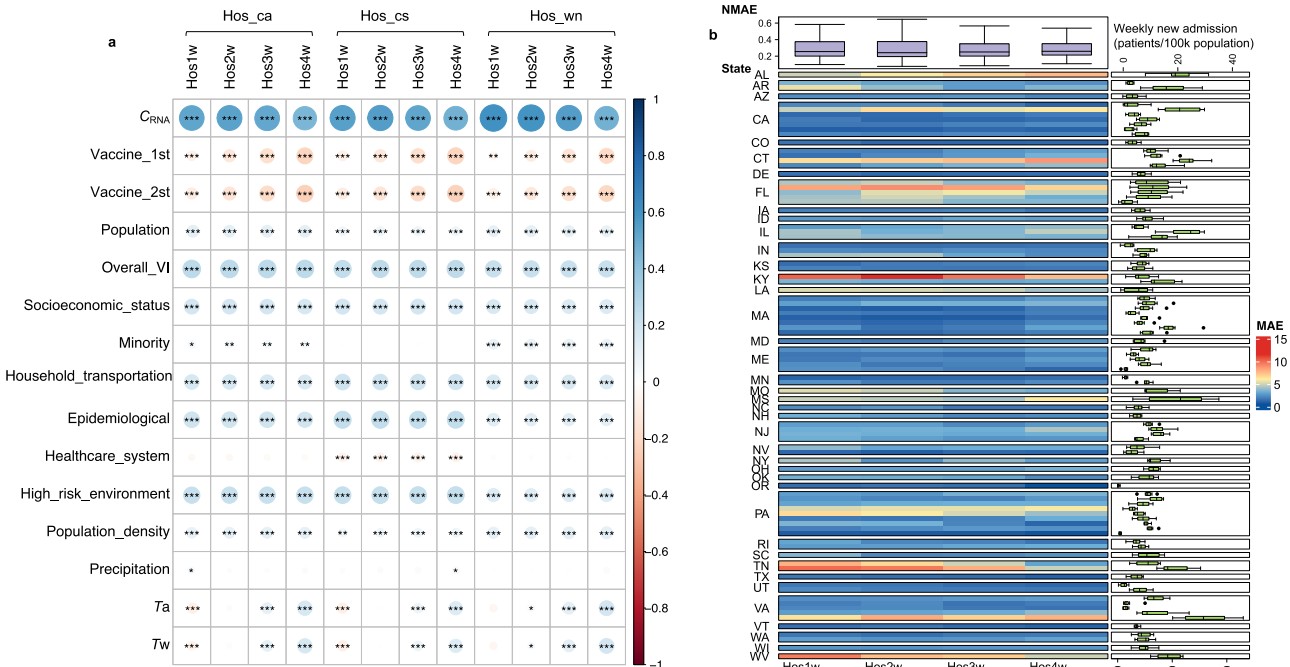

**Fig. 3 | Correlation between explanatory factors and hospitalization records and performance of WBE-based batch model for predicting future weekly new hospital admissions. a** Spearman's correlation between all the explanatory factors and hospital admission records (three types of hospitalization indicators (i.e. Hos_wn: weekly new admission, Hos_cs: census inpatient sum, and Hos_ca: census inpatient average) under 4 leading times (Hos1w, Hos2w, Hos3w, Hos4w: the upcoming week, the second, third and fourth week after the wastewater sampling, respectively). The color and circle size indicate the strength of the correlation (bigger circle = stronger correlation; blue color = positive correlation and red color = negative correlation). The significance of the correlation is determined through two-side *t*-test, and marked as *, **, and *** representing a *p* value of ≥0.01 and <0.05, ≥0.001 and <0.01 and <0.001, respectively. The detailed *p* values are provided in Table S6. **b** The mean absolute error (MAE) of the established batch model for predicting weekly new hospital admissions in these 99 counties from June 2022 to January 2023. The main heatmap shows the MAE (reflected by the color) between the prediction and the actual admission record for each county. The box plot on the right shows the weekly new admissions (patients/ 100k population) for each county during June 2022–January 2023. The colored box indicates the 25th and 75th percentiles, and the whiskers indicate the 1.5× the interquartile range. The line in the box indicates median and dots represent outliers. *N* = 31 for each county. The top box plot summarizes the normalized MAE (NMAE) for the prediction at different leading times (Hos1w, Hos2w, Hos3w, and Hos4w). The NMAE is calculated as the MAE divided by the mean of weekly new admission numbers (see methods for equations). The colored box indicates the 25th and 75th percentiles, and the whiskers indicate the 1.5× the interquartile range. The line in the box indicates median and dots represents outliers. *N* = 99 for each prediction leading time (Hos1w-Hos4w).

## Performance and leading time of the established models in predicting future admissions

WBE-based prediction models were established for all twelve targets (3 indicators ×4 leading times) using the data obtained from June 2021 to May 2022 (Fig. 1). The model performance was evaluated using correlation coefficients (R), mean absolute error (MAE), and normalized MAE (NMAE) between model predictions and targets. For all three types of hospitalization indicators (i.e. weekly new admission, census inpatient sum, and census inpatient average), the established WBE-based model well described the pattern of data observed from June 2021 to May 2022 with overall R values over 0.90 and NMAE within 0.30 (Supplementary Table S2). When applying the established batch models for predicting the future hospitalization indicators in June 2022–January 2023, the model performance for weekly new admission was greatly better than census inpatient sum, and census inpatient average (Table 1). The prediction accuracy achieved R of 0.81–0.82 and NMAE of 0.32–0.37 for predicting weekly new admission, but only R of 0.59–0.67 and NMAE of 0.53–0.76 for census inpatient sum and R of 0.66–0.69 and NMAE of 0.51–0.65 for census inpatient average (Table 1). This indicates that WBE-based predictions are likely more capable of capturing the weekly new admissions rather than the census average or sum of inpatients in the week.

Using WBE-based predictions for weekly new admission, the batch models achieved a reasonable performance with an overall MAE of 4 patients/100k population for weekly new admissions in the next 1–4 weeks (Table 1). In these 99 counties, the prediction

performance for weekly new admission in four leading times was comparable, with a MAE of 1–19 patients/100k population for the first and second week (Hos1w and Hos2w), 2–18 patients/100k population for the third week (Hos3w), and 2–16 patients/100k population for the fourth week (Hos4w) after the wastewater sampling (Fig. 3b). Higher MAE was observed in counties with higher weekly new admissions. Overall, the NMAE of most counties was within 0.2–0.4 (Fig. 3b).

To facilitate comparison, additional prediction models were established using random forest algorithms based on weekly new COVID-19 cases and test positivity (referred to as case-based predictions) and the relevant weekly records for each hospitalization indicator (referred to as record-based predictions) at the county level. For model establishments, both case-based models (R = 0.81–0.97, NMAE = 0.25–0.41) and record-based models (R = 0.80–0.96, NMAE = 0.23–0.43) showed comparable or slightly worse performance than WBE-based predictions (R = 0.90–0.97, NMAE = 0.22–0.30) in describing the patterns in the data for all three hospitalization indicators (Table S2, Supplementary Fig. S4). When being applied to predict the future targets in June 2022–January 2023, both case-based or record-based models showed slightly better prediction for weekly new admission than census inpatient sum and census inpatient average (Table 1). The NMAE values of our county-level case-based (0.40–0.42) and record-based (0.38–0.45) models for weekly new admission were comparable to previous case-base or record-based (or ensembled) prediction for daily new admissions at the state or national level in the

**Table 1 | Performance of WBE-based, case-based and record-based batch models predicting the future targets in June 2022–January 2023 under 4 leading times (Hos1w-Hos4w)**

| Indicators | Model | Hos1w | | | Hos2w | | | Hos3w | | | Hos4w | | |
|---|---|---|---|---|---|---|---|---|---|---|---|---|---|
| | | R | MAE | NMAE | R | MAE | NMAE | R | MAE | NMAE | R | MAE | NMAE |
| Weekly new hospitalization | WBE | 0.82 | 3.65 | 0.35 | 0.81 | 3.84 | 0.37 | 0.82 | 3.59 | 0.34 | 0.82 | 3.30 | 0.32 |
| | Record | 0.78 | 3.90 | 0.38 | 0.70 | 4.05 | 0.39 | 0.65 | 4.20 | 0.40 | 0.56 | 4.63 | 0.45 |
| | Case | 0.51 | 4.25 | 0.41 | 0.41 | 4.23 | 0.40 | 0.40 | 4.46 | 0.42 | 0.44 | 4.28 | 0.41 |
| Census inpatient sum | WBE | 0.60 | 61.74 | 0.76 | 0.59 | 58.37 | 0.72 | 0.67 | 46.61 | 0.57 | 0.62 | 42.71 | 0.53 |
| | Record | 0.78 | 25.78 | 0.32 | 0.69 | 32.82 | 0.40 | 0.61 | 35.68 | 0.43 | 0.47 | 38.25 | 0.47 |
| | Case | 0.56 | 34.39 | 0.43 | 0.80 | 33.71 | 0.42 | 0.56 | 34.99 | 0.43 | 0.63 | 34.11 | 0.42 |
| Census inpatient average | WBE | 0.69 | 7.26 | 0.65 | 0.68 | 6.84 | 0.61 | 0.67 | 6.22 | 0.55 | 0.66 | 5.77 | 0.51 |
| | Record | 0.87 | 3.71 | 0.34 | 0.56 | 4.74 | 0.42 | 0.55 | 5.46 | 0.48 | 0.54 | 5.11 | 0.45 |
| | Case | 0.69 | 4.49 | 0.40 | 0.65 | 4.61 | 0.41 | 0.53 | 7.58 | 0.67 | 0.62 | 4.63 | 0.41 |

Note: Hos1w, Hos2w, Hos3w, and Hos4w represent the first, second, third, and fourth week after wastewater sampling, respectively. R denotes the correlation coefficient, MAE indicates the mean absolute error, and NMAE refers to the normalized mean absolute error (refer to the Methods section for detailed calculations). WBE means the wastewater-based epidemiology.

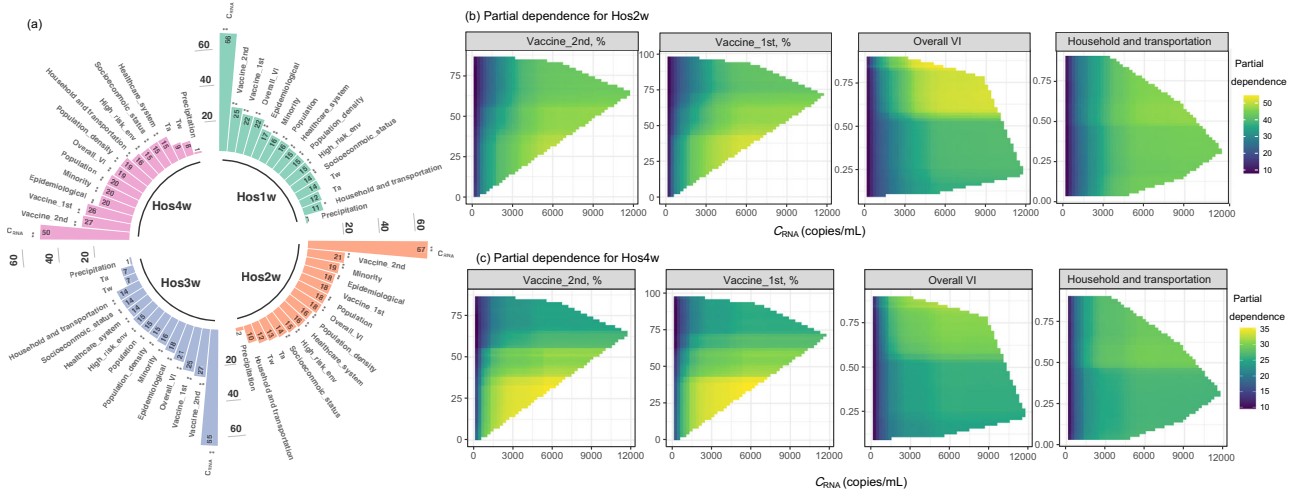

**Fig. 4 | Importance and contribution of the explanatory factors to the established model for weekly new admissions prediction. a** The importance of explanatory factors was ranked by the increase in %MSE (percent change in mean square error when the explanatory factor is permuted). A higher increase in %MSE corresponds to higher importance. The significance of the explanatory factors was marked as *, **, and *** representing a $p$ value of ≥0.01 and <0.05, ≥0.001 and <0.01 and <0.001, respectively. Hos1w, Hos2w, Hos3w, Hos4w are the upcoming week, the second, third and fourth week after the wastewater sampling, respectively. The two-factor partial dependence for predicting weekly new admissions at the second week (Hos2w, subfigure **b**) and fourth week (Hos4w, subfigure **c**), on $C_{RNA}$ and four significant explanatory factors used in the models. The horizontal axis represents the values of $C_{RNA}$, whereas the vertical axis represents the values of the other four explanatory factors (as shown in the title). The color gradients in the figure indicate the partial dependence of the predicted target concerning a specific x-value and y-value combination.

USA (NMAE = 0.35–0.45, leading time of 2–3 weeks)[30,31]. Nonetheless, our WBE-based models showed superior performance compared to case-based or record-based models for weekly new admission prediction, including those from previous studies, with lower NMAE (0.32–0.37) and longer leading time (1–4 weeks).

## Contribution of explanatory factors for WBE-based prediction for weekly new admissions

The importance of explanatory factors for models established for weekly new admission prediction was evaluated by the increase in mean squared error (MSE, %) of predictions when the value of a certain explanatory factor was permuted[32]. Regardless of the leading time, $C_{RNA}$ was found to be the most important factor for predicting weekly new admissions, contributing to a significant increase in MSE (50–67%, $p = 0.010$) (Fig. 4a). Vaccination coverage (Vaccine_1st and Vaccine_2nd) also played a crucial role, contributing to a 19–28% increase in MSE ($p = 0.01$–0.10), with Vaccine_2nd being more important (21–28% increase in MSE, $p = 0.01$). Most CCVI indexes showed significant

contributions of a 10–23% increase in MSE for predicting weekly new admissions. As the leading times increased, there was a decrease in the significance of $C_{RNA}$ in predicting weekly new admissions, going from 66–67% for the first three weeks to 51% for the fourth week after the wastewater sampling. Meanwhile, the importance of CCVI in household and transportation increased from 10–11% for predicting the weekly new admissions at the first two weeks to 14–17% for predicting the weekly new admissions at the third and fourth week after the wastewater sampling. Population density showed an increase from 15–16% for predicting the weekly new admissions at the first three weeks to 19% for predicting the fourth week after wastewater sampling (Fig. 4a). This suggests that COVID-19 transmission-related information is more critical for predicting weekly new admissions in later weeks. Other CCVI indexes showed comparable importance of a 15–25% increase in MSE, regardless of the leading time. The $T_w$, and $T_a$ showed limited contributions (7–14% increase in MSE, $p = 0.28$–0.99). While precipitation showed negligible contribution to the model prediction (1–2%, $p = 0.3$–0.9), which is likely due to that $C_{RNA}$ being

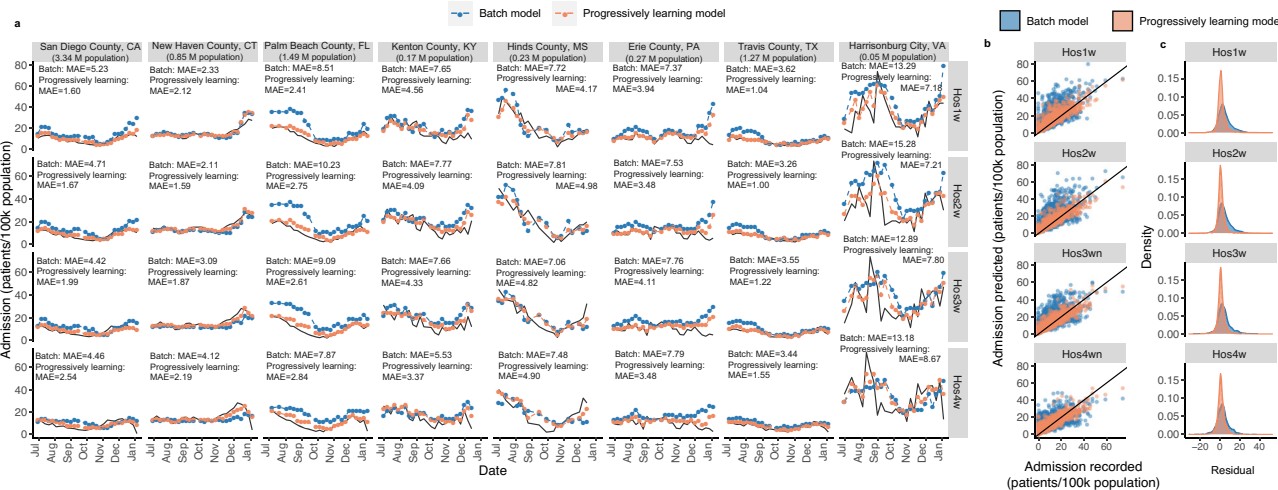

**Fig. 5 | Comparison between actual admission records and the prediction results from batch models and progressive learning models for data in June 2022- January 2023. a** The prediction results from the batch model (in blue) and progressive learning model (in orange) and the actual admission records (in black) for weekly new admissions in eight representative counties. Hos1w, Hos2w, Hos3w, Hos4w are the upcoming week, the second, third and fourth week after the wastewater sampling, respectively. **b** The prediction results from the batch model (in blue) and progressive learning model (in orange) verse the actual admission records for weekly new admissions. **c** The error distribution between prediction results and actual admission records for the batch model (in blue) and progressive learning model (in orange) for predicting weekly new admissions.

normalized to pepper mild mottle virus (a fecal indicator) to minimize any potential dilution-related variations[15].

Through one-factor partial dependence analysis, clear non-linear relationships were observed between the weekly new admissions and explanatory factors (Supplementary Fig. S5), with the overall increasing or decreasing trend consistent with the positive or negative correlation observed in Fig. 3a. Considering the key role of $C_{RNA}$ in WBE as it reflects the infection status in the community, two-factor partial dependence analysis was conducted based on $C_{RNA}$ and explanatory factors with significant contributions (as shown in Fig. 4a). Vaccination showed a clear impact on reducing weekly new admissions under the same infection status (reflected by $C_{RNA}$) for all eight targets (Fig. 4b, c and Supplementary Fig. S6), especially when Vaccine_2nd was over 60%. A higher vulnerability in overall VI, household and transportation, epidemiological factors, or socioeconomic status increased the weekly new admissions under the same infection status, with a more pronounced impact observed when these CCVI indexes exceeded 0.5 (Fig. 4b, c, and Supplementary Fig. S6). In addition, population size, while significantly impacting the increase of MSE, had a negligible impact on changes in weekly new admissions under the same infection status (Supplementary Fig. S6).

**The necessity of periodical updates of WBE-based models**
The random forest models developed in the previous sections for predicting weekly new admissions under different leading times were progressively updated every four weeks between June 2022 and January 2023, considering the healthcare system settings (Fig. 1). The performance of the models improved greatly through progressively learning compared to the batch model (Fig. 5b, c). The MAE reduced from 4 patients/100k population in the batch models to 3 patients/100k population in the progressively learning models, and the NMAE decreased from 0.32–0.37 in the batch models to 0.28–0.29 in the progressive learning models (Supplementary Table S3). The errors between the model predictions and actual clinical records in both batch and progressive learning models followed a normal-like distribution, with a mean value of 1.68–3.77 in the batch models and 0.88–1.39 in the progressively learning models (Fig. 5b, c). For each leading time, the peaks of the error distribution were closer to 0 in the progressively learning models than in the batch models (Fig. 5b, c). The

autocorrelation functions (ACF)[33] confirmed that the residuals (errors) were merely white noise with no significant serial correlation and were not dependent on an adjacent observation (Supplementary Fig. S7). The MAE observed in each county also reduced from 1–19 patients/100k population (Fig. 3b) to 1–12 patients/100k population in the progressively learning models (Fig. 6a), regardless of the leading time. This implies that a progressive update of the model is essential for improving the prediction accuracy.

Specifically, the prediction performance of the batch and progressively learning models were illustrated in eight representative counties (selected based on population size). Predictions from both batch models and progressively learning models reached good agreements with the actual admission records (Fig. 5a), regardless of the leading time. Compared with batch models, progressively learning models reduced the MAE by 10–70% for a certain county and showed better prediction capability towards the rapid changes in the trends (both sudden rise and drops) (Fig. 5a). The population size in the county did not appear to have a clear impact on the model's accuracy, with most counties achieving comparable NMAE (0.14–0.35) (Fig. 5a). Although both batch models and progressively learning models tended to underestimate some peaks in Harrisonburg city, which had the smallest population size (Fig. 5a), this was more likely due to higher admission numbers recorded in the county, which were less frequently presented in the datasets (Fig. 2c). This resulted in fewer data points for models to learn and subsequently predict the peaks.

**Transferability of the progressively updated WBE-based models**
The progressive learning models (established in above sections) using the data from 99 counties were applied for predicting the weekly new admissions in another 60 different counties from 30 states in the USA with a population size ranging from 0.2 M to 10.0 M (nearly 40 M population in total, Supplementary Table S4). These 60 counties were all unknown to the model (not included in the model establishment process), and 7 of them were from 5 new states (i.e. DC, GA, NM, ND, SD, Supplementary Table S4). From June 2022 to January 2023, the progressively learning models reasonably predicted the weekly new admissions in these 60 counties in the next 1-4 weeks after the wastewater sampling (Fig. 6b, c), with an average MAE of 7-8 patients/100k

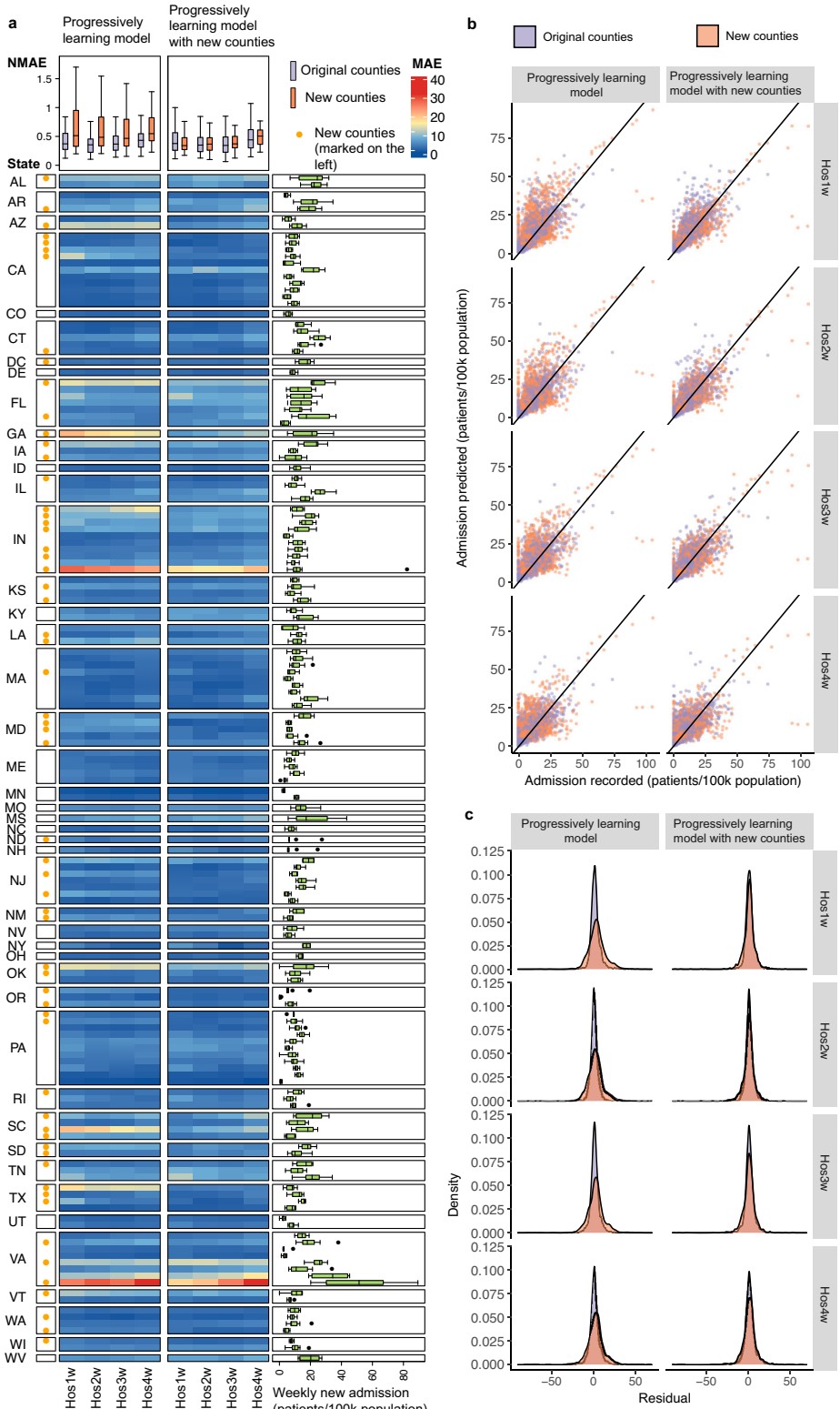

population and an average NMAE of 0.43–0.48 (Fig. 6a). In six representative counties, although the progressively learning models captured the overall trends of the data, the models were insensitive to sudden changes in the patterns (drop or rise), especially for the counties from a new sate to the model (the first three counties in Fig. 7) at longer leading times (Hos3w and Hos4w) (Fig. 7).

We further included the data of these 60 different counties from June 2022 to January 2023 into the progressively learning models with the same update frequency (4 weeks). With the data of new counties

included, the MAE of the prediction for these 60 counties reduced to 4–5 patients/100k population with an average NMAE of 0.31–0.35 for the next 1–3 weeks, and MAE of 6 patients/100k population and NMAE of 0.45 for Hos4w. The inclusion of data from new counties did not affect the prediction performance for the original 99 counties with comparable MAE at 3 patients/100k population and NMAE of 0.27–0.28 for the first three weeks (Hos1w, Hos2w and Hos3w), but slightly increased the MAE to 4 patients/100k population (NMAE of 0.35) for the fourth week (Hos4w).

**Fig. 6 | The performance of progressively learning models with and without the data from 60 new counties in June 2022–January 2023. a** The MAE of the progressively learning models with and without the data from new counties for predicting the weekly new admissions in these 159 counties (99 original counties and 60 new counties) in June 2022–January 2023. The 60 new counties are labeled with yellow dot on the left. The color of each cell in the main heatmap indicates the MAE between the prediction and the actual admission record in each county. The box plot in the right presents the weekly new admissions (patients/100k population) in each county during June 2022–January 2023. The colored box indicates the 25th and 75th percentiles, and the whiskers indicate the 1.5× the interquartile range. The line in the box indicates median. *N* = 31 for each county. The top box plot summarizes the NMAE for the prediction in different leading times (Hos1w-Hos4w: the

upcoming week, the second, third and fourth week after the wastewater sampling) for the original 99 counties (in purple, *N* = 99) and the 60 new counties (in orange, *N* = 60). The colored box indicates the 25th and 75th percentiles, and the whiskers indicate 1.5× the interquartile range. The line in the box indicates median. **b** The prediction results from the progressively learning models with (on the right) and without (on the left) the data from new counties for predicting the weekly new admissions in the original 99 counties (in purple) and the 60 new counties (in orange). **c** The error distribution between prediction results and actual admission records from the progressively learning models with (on the right) and without (on the left) the data from new counties for predicting the weekly new admissions in the original 99 counties (in purple) and the 60 new counties (in orange).

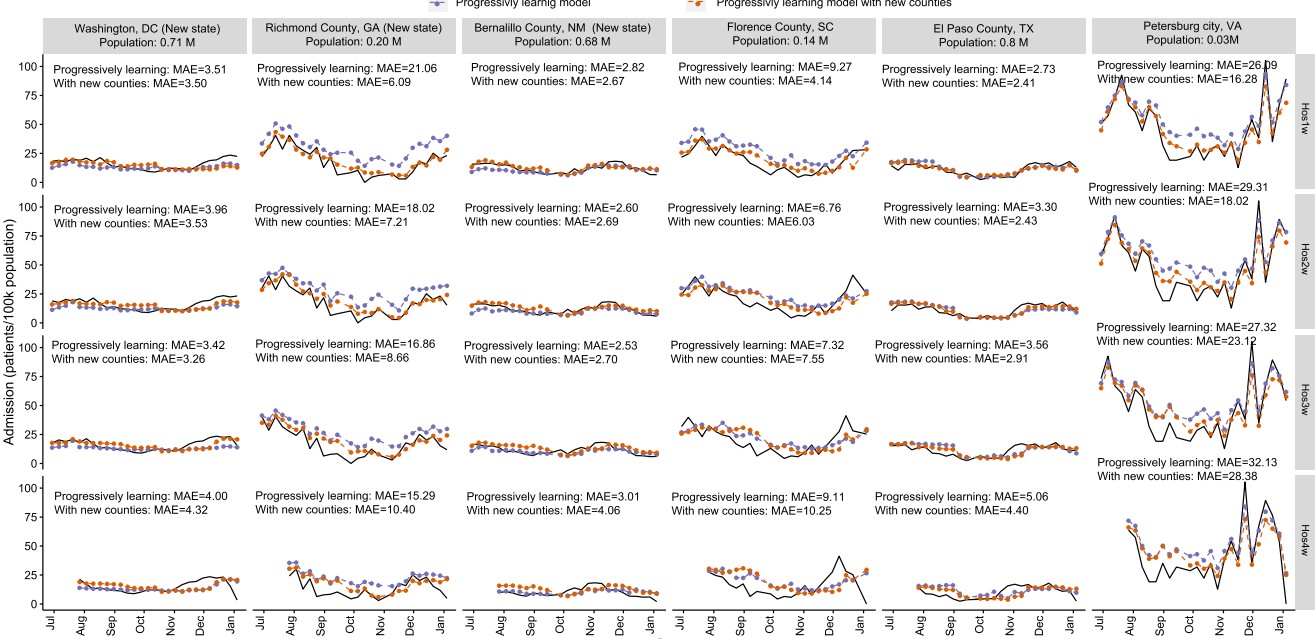

**Fig. 7 | Transferability of progressively learning models.** The prediction results from the progressively learning model without (in purple) and with (in orange) the data from new counties for weekly new admissions in six representative counties.

Hos1w, Hos2w, Hos3w, and Hos4w: the upcoming week, the second, third and fourth week after the wastewater sampling, respectively.

## Discussions

The COVID-19 pandemic has profoundly impacted the healthcare system with additional burdens to hospitalization demands. Using wastewater surveillance data from the past 20 months from 99 counties, our studies revealed the feasibility of using WBE to predict the county-level weekly new hospitalizations with a leading time of 1–4 weeks. The early warning capability of WBE for predicting the weekly new hospital admission in the healthcare system is likely related to viral RNA shedding from COVID-19 patients to sewers and the transmission of COVID-19 within the population. Sputum and feces have been identified as the major shedding sources of SARS-CoV-2 RNA in wastewater, with the shedding load peaked ($10^2$–$10^3$ higher than other times) in the first couple of days before, to a week after, the symptom onset[34–37]. Thus, the changes of $C_{RNA}$ in wastewater samples are more sensitive to the variations in the numbers of COVID-19 infections at their early infection stages. Furthermore, recent meta-analyses revealed that COVID-19 patients remain contagious for around 12 days and the median time between symptom onset and hospitalization was 7 days (IQR: 5–10 days)[25,38,39]. Thus, the $C_{RNA}$ of a certain wastewater sample likely 1) directly reflects the newly infected patients, and 2) indirectly reflects the future COVID-19 patients in the following 12 days due to the close contact with the current patients.

Depending on the severity of the symptoms, part of these newly and future infections are likely admitted to hospitals in the next 14 days, and 14–26 days after wastewater sampling, respectively. This is consistent with the 1–4 weeks of leading time in WBE-based predictions for weekly new admissions in our study and also reflected by the contributions of explanatory factors.

$C_{RNA}$ was found as the most crucial explanatory factor, followed by population-health-related information (i.e. Vaccine_2nd, Vaccine_1st, and CCVI index in epidemiology factors) and COVID-19 transmission-related information (i.e. CCVI in population density and household and transportation). While population-health information showed comparable importance regardless of leading times, COVID-19 transmission-related information became increasingly important for longer leading times. Under the same $C_{RNA}$ (infection status), a higher CCVI index in population density or household and transportation increased weekly new hospitalizations, especially when they are over 0.5. These two factors reflect the proximity to and interaction with other people and exposure to diseases, which directly relates to the transmission probability (impacting the number of future cases)[22–24]. Under the same infection status, a higher Vaccine_2nd, higher Vaccine_1st, or lower CCVI in epidemiological factors reduced the weekly new hospitalizations, particularly under Vaccine_2nd > 60% or CCVI in

epidemiological <0.5. This is consistent with the clinical observations of over 1 million patients, where a single dose and two doses of any vaccine (i.e. Pfizer-BioNTech, Oxford-AstraZeneca, Moderna that commonly used in the USA) were associated with a 35% and 67% reduction in the risk of hospitalization, respectively[40]. The CCVI in epidemiological factors considers high-risk populations for COVID-19 such as elderly adults and individuals with underlying health conditions (e.g. respiratory or heart conditions) that have been shown to be associated with more severe COVID-19 symptoms in clinical observations[22–24]. This supports our observations that health-related information was critical for predicting COVID-19-induced hospitalizations under all four leading times, while transmission-related information was more important for models with longer leading times.

The WBE-based predictions more accurately captured the weekly new hospitalizations compared to the daily census average or census sum patient numbers in the week. The census admission numbers for a particular week encompass both new admissions and continuing admissions from previous weeks. Hospital stays can vary significantly from a few days to as long as 41 days, depending on factors such as prescribed treatments, chronic conditions (like diabetes and hypertension), nutritional risks (such as body mass index and cognitive impairment), etc.[41–44]. Accurately capturing and integrating these variables at the population level into WBE-based predictions (or any other existing approaches) may be challenging. This is also commonly observed in case-based or record-based models (the existing approaches), where better prediction accuracy was achieved for new admissions rather than census inpatient numbers[31,45]. More importantly, our WBE-based predictions (NMAE = 0.32–0.37) outperformed the record-based or case-based models in terms of the accuracy and leading time, for county-level predictions (our study, NMAE = 0.38–0.45, leading time up to 4 weeks) and state/national-level predictions (previous studies, NMAE = 0.35–0.45, leading time of 2–3 weeks)[30,31]. The sub-optimal performance of case-based predictions may be attributed to the potential bias of clinical testing, where only part of the infections in the community can be captured[7,8]. For record-based prediction, the inherent lag between the infection and hospitalization might also affect the prediction accuracy, especially for rapid changes in the infection status[31]. In contrast, WBE unbiasedly captures the infection status among the population at the early stage of the infection[34–37]. The WBE-based prediction approach established in our study offers a promising alternative or complementary approach to provide early warning for future COVID-19-induced admissions, allowing a preparation window of 5–28 days in healthcare systems (considering several hours to 2 days of turnover time for wastewater sample analysis).

For the application of the WBE-based models, progressively updating the model with the most recent datasets greatly improved the prediction accuracy, reducing the MAE by 10–70% for a certain county in comparison to the batch models, reaching an overall NMAE of 0.28–0.29 under a leading time of 1–4 weeks in these 99 counties. The progressive learning models also showed reasonable transferability to other 60 counties from 30 states in the USA, with slightly higher NMAE of 0.43–0.48. After incorporating the data from new counties on a monthly basis into the progressively learning models, the updated model reached comparable prediction accuracy towards all 159 counties, with a NMAE of 0.31–0.35 for the next 1–3 weeks, and 0.45 for the fourth week. Thus, for future applications, the progressive learning model with the most recent datasets from relevant counties is highly recommended, and the methodology established in our study has a huge potential to be applied in other regions/counties.

The necessity of periodic updates of localized data from relevant counties is likely related to the variation and evolution of immunity and SARS-CoV-2 variants in different counties, as well as the nature of machine-learning approaches. As discussed in the above sections, vaccination coverage showed a significant contribution to predicting weekly new hospitalizations. However, the effect of vaccination on immune protection typically declines over time due to antibody neutralization[46]. The effectiveness of Pfizer or Moderna vaccines decreased from around 65–70% to approximately 10%, 20 weeks after the second dose[40]. Moreover, SARS-CoV-2 variants evolve over time and exhibit distinct regional patterns across the nation[47]. Reduced risks of progression to severe clinical outcomes (i.e. hospitalization) were observed with Omicron infections than with Delta infections[48]. Even during the Omicron infections, the effectiveness of vaccines and the probability of hospitalization also varied against different Omicron subvariants[48–50]. Thus, the number of hospitalizations under the same infection status may also depend on the remaining immunity from vaccinations and subvariants of infections in each county over time. The progressively learning model provides the most up-to-date information, allowing the model to adjust its structure to accommodate new changes.

There are several limitations in this study. The community's immunity is affected by several factors, such as booster shots' recipient coverage and the time interval between booster shots and the second dose of vaccination, as well as infection-induced immunity[51]. Unfortunately, such information on booster shots was not available at the county level, and the effectiveness and duration of infection-induced immunity remain largely unknown[46,48]. Thus, such information was not included in our models. For future research, it is recommended to incorporate time-weighted vaccination and prior infections to evaluate community immunity to predict hospital admissions. Additionally, immune protection from vaccination or prior infections varies against different subvariants[48]. Since reports on the proportion of infections from different variants/subvariants often delay due to the time required for clinical and wastewater analyses (which can take up to months depending on analytical capabilities), such information was not included in our study. However, it is encouraged for future investigations when timely information becomes available.

It is worth noting that the WBE data used in this study is retrospective. Risks of severe clinical outcomes and the time between the infection/symptom onset and the admission likely vary with the population structure changes (aging, relocation, and seasonal population movements)[48–51]. Adjustment of the model structure based on the localized conditions should be considered in future studies. Additionally, although normalization techniques that use endogenous population biomarkers can reduce the potential noise caused by the population size captured by the wastewater sample[17,52,53], the uncertainty caused by population mobility cannot be avoided in WBE-based predictions, as well as case-based or record-based predictions[53–55]. Recently, researchers have employed mobility surveillance data, such as cell phone mobility data, to enhance prediction accuracy[31,56]. Although this information is not included in our models due to its unavailability at the county level during the study period, it is highly recommended for future studies when the data becomes accessible. In addition, considering the regional variations in the leading time and the turnover time for sample analysis (up to several days), the WBE-based models predicted hospitalizations on a weekly basis. Although this meets the weekly resource allocation and staff arrangement in most healthcare systems, for certain regions where a high-resolution (such as daily) prediction is required, the case-based or record-based prediction might be more suitable than WBE-based predictions.

## Methods

### County-level wastewater surveillance data in the US

Wastewater surveillance data was obtained from the Biobot Nationwide Wastewater Mentoring Network (biobot.io/data), the largest publicly available dataset on SARS-CoV-2 RNA concentrations in wastewater. The Biobot Nationwide Wastewater Mentoring Network was selected by the USA Department of Health and Human Services for wastewater-based monitoring, covering 30% of the USA population. The detailed sampling, analytical, and data process protocol for the

wastewater data were described in Duvallet, Wu[15], and Supplementary information Text 1. Briefly, the concentration of SARS-CoV-2 RNA detected in each wastewater sample was normalized to pepper mild mottle virus (a fecal indicator) to minimize any potential noise caused by the dilution, population size, and wastewater flow[15]. The normalized SARS-CoV-2 RNA concentration was further aggregated based on county and sample amount to preserve the anonymity of participating utilities, providing one SARS-CoV-2 RNA concentration ($C_{RNA}$) per week for each county (details provided in Supplementary Text 1). Considering the progress of the vaccination and experience in the sample analysis, the county-level weekly SARS-CoV-2 concentrations in wastewater ($C_{RNA}$) from June 2021 to January 2023 were obtained from Biobot and used in this study.

### County-level hospitalization data in the USA

Three indicators for hospitalization numbers were used including: 1) weekly new admission, 2) the total number of patients who stayed in an inpatient bed during the week (census inpatient sum), and 3) the daily average number of patients who stayed in an inpatient bed in the week (census inpatient average). The data for weekly new hospitalizations, census inpatient sum, and census inpatient average was retrieved from HealthData.gov. This dataset is derived from reports with facility-level granularity across two main sources: (1) the Department of Health and Human Services (HHS) TeleTracking, and (2) reporting provided directly to HHS Protect by state/territorial health departments on behalf of their healthcare facilities. By combining data from these sources, the dataset from HealthData.gov ensured a comprehensive and validated data collection. Briefly, facility-level data for hospital utilization in each county was reported on a weekly basis, along with the corresponding county where each facility is located. In the dataset, when there are fewer than 4 patients in a data field, the cell is redacted and replaced with -999999. To ensure the accuracy of the prediction, we removed such missing values from the data in our study. The county-level values for each indicator were then obtained from the aggregation of facilities within the same week and county. Considering the preparation window, records for each indicator in the next 1–4 weeks of the wastewater sampling were summarized for each county and used in this study. The hospitalization numbers used in this study are anonymous, which do not require ethical approval.

### County-level population-related and weather data

For better management and policy-making, COVID-19 Community Vulnerability Index (CCVI) was established by Surgo Foundation and used by CDC for COVID-19-related response in the USA[57]. The CCVI is adapted from Social Vulnerability Index (SVI) from CDC with modifications regarding COVID-19-related risk factors (such as high-risk population and environment)[23,29]. The CCVI is also widely used for evaluating the epidemiological impacts/responses under COVID-19[58,59]. At the county level, CCVI considers 40 measures from census data, covering 7 themes including i) socioeconomic status; ii) minority status and language, iii) housing type, transportation, household composition, and disability ("household and transportation" hereafter), iv) epidemiological factors, v) healthcare system, vi) high-risk environment, and vii) population density, with an overall VI summarizing these 7 themes[23,29]. The CCVI overall score as well as the 7 theme indices range from 0 to 1, with 1 representing the most vulnerable area and 0 representing the least vulnerable area[23,29]. The CCVI indexes of the overall score and 7 themes were obtained from the publicly available website (https://precisionforcovid.org/ccvi). We chose to use CCVI to reflect the population demographic rather than incorporating multiple measures from population census data to ensure that the model/approach could be easily adapted to most regions based on their existing management systems, thereby promoting the transferability of the established approach.

The ratio of vaccinated people (%) recorded on Monday of the sampling week for the first dose (Vaccine_1st) and the second dose (Vaccine_2nd) among the total population in each county was obtained from the CDC record (https://data.cdc.gov/Vaccinations). For comparison purposes, the case-based prediction was also established. The daily COVID-19 cases (cases/100k population), and test positivity (positive tests/total tests) were collected from publicly available records in USAFacts (https://usafacts.org/visualizations/coronavirus-covid-19-spread-map) and aggregated on a weekly basis.

Considering the potential dilution of wastewater due to precipitation, the daily average precipitation (mm) in the week of wastewater sampling was obtained from the USA Environmental Protection Agency for each county (https://www.ncdc.noaa.gov/cdo-web/datatools/lcd). During in-sewer transportation, potential decay of SARS-CoV-2 RNA occurs and is impacted by the wastewater temperature[60,61]. Although the wastewater temperature was not reported in the Biobot data, previous studies revealed that it can be calculated from air and soil temperature[62]. Thus, the daily air temperature in the week of the wastewater sampling was obtained from the USA Environmental Protection Agency for each county (https://www.ncdc.noaa.gov/cdo-web/datatools/lcd). The average air temperature of the week ($T_a$, °C) was summarized and used to further calculate the average wastewater temperature ($T_w$, °C) using the method described by Hart and Halden[62].

### Model establishment using random forest algorithm

Random forest is a non-parametric machine learning approach to modeling the relationship between the potential explanatory factors (input variables) and the target[63,64]. Random forest algorithm relies on establishing a group of individual decision trees to optimize model fit. Two approaches are incorporated to ensure the randomness and diversity of the decision trees: i) bootstrapping the training data so that each tree grows with a different sub-sample; ii) selecting features randomly to generate different subsets of explanatory variables for splitting nodes in a tree[65]. The correlations between observations in the data generally do not affect the individual trees or the final model. Thus, autocorrelation is not typically considered an issue for random forest models[63,64].

Prediction models were established for each hospitalization indicator (i.e., weekly new hospitalizations, census inpatient sum, census inpatient average) using three types of prediction (i.e. WBE-based, record-based, and case-based), under four leading times (i.e., Hos1w, Hos2w, Hos3w, and Hos4w). For model establishment, data from June 2021 to May 2022 (3162 data points for each target, 12 months) were utilized to describe the patterns for each target through the random forest algorithm in R (ver 4.2.0, R Foundation for Statistical Computing, http://www.R-project.org/). For each hospitalization indicator, 13 common explanatory factors were used between the WBE-based models and case-based or record-based models. These 13 common factors included: CCVI indexes (8 factors); county-level vaccination coverage (Vaccine_1st and Vaccine_2nd, %); population size of the county; and weather ($T_a$, °C, and precipitation, mm). In addition to these 13 common factors, the weekly new COVID-19 cases (cases/100k population) and test positivity (positive tests/total tests) were used for case-based predictions, $C_{RNA}$ and wastewater temperature ($T_w$, °C) were used for WBE-based predictions, and hospitalization records for each indicator (i.e., weekly new hospitalizations, census inpatient sum, census inpatient average) in the week of wastewater sampling were used for record-based prediction. The correlation between the hospitalization indicators and the explanatory factors for case-based and record-based models were provided in the supplementary Text 3. As the skewness of the data does not affect the structures and performance of random forest models[66] (which was also demonstrated in Supplementary Table S5), transformations for data were not included in our study. Considering that $C_{RNA}$ used in the study was normalized to

pepper mild mottle virus (a fecal indicator) to minimize any potential dilution-related variations[15], the interaction between precipitation and $C_{RNA}$ was not included as a factor in the WBE-based models.

To establish each of the 36 models (comprising 3 types of prediction ×3 hospitalization indicators ×4 leading times), the data was randomly divided into three parts, training set (70% of data), validation set (15% of data), and test set (15% of data), regardless of their counties and time points. The training set was used to train the random forest models, and the validation set was used in conjunction to optimize model structures. The test set was then used to evaluate the model's prediction capability over unseen data during the model establishment stage. The rationale behind the randomized data selection was to ensure that the models developed could accurately describe the generalized patterns within the datasets from June 2021 to May 2022.

The performance of the model was assessed using the correlation coefficient (R), mean absolute error (MAE), and normalized mean absolute error (NMAE) as Eqs. (1) and (2). These evaluation criteria, particularly NMAE, have been extensively employed in previous prediction studies[30,31], allowing for inter-study comparisons.

$$MAE = \frac{\sum_{i=1}^{n}(|y_i - \hat{y}_i|)}{n} \tag{1}$$

$$NMAE = \frac{\sum_{i=1}^{n}(|y_i - \hat{y}_i|)}{\sum_{i=1}^{n}(y_i)} \tag{2}$$

where $y_i$ is the i[th] observation of $y$ and $\hat{y}_i$ the predicted $y_i$ value from the model. The $n$ is the total number of data points.

### Significance and contribution of explanatory factors in established models

The significance and contribution of each explanatory factor in the model were determined using the frPermute package in R through a 5-fold cross-validation with 5 replicates[32]. For a certain set of data, the importance score for each explanatory factor was determined as the percentage increase in mean square error (%MSE) observed when the value of an explanatory factor was permuted, compared to when no metrics were permuted. The partial dependence between the output (target) and explanatory factors (input variables) was analyzed using the Pdp package in R. The partial dependence depicts the marginal effect of one or two explanatory factors on the outputs while controlling for other explanatory factors[67]. Mathematically, the partial dependence function for regression is defined as (Eq. (3)).

$$\hat{f}_S(x_S) = E_{X_C}\left[\hat{f}(x_S, X_C)\right] = \int \hat{f}(x_S, X_C)dP(X_C) \tag{3}$$

The $x_S$ are the features of explanatory factors that we are interested in, and $X_C$ are the other explanatory factors used in the machine learning model $\hat{f}$. The mathematical expectation is denoted by $E$ and probability by $P$. The partial function $\hat{f}_S(x_S)$ shows the relationship between $x_S$ feature and the predicted targets. The partial function $\hat{f}_S(x_S)$ is estimated by calculating averages in the training data, also known as Monte Carlo method as Eq. (4):

$$\hat{f}_S(x_S) = \frac{1}{N}\sum_{i=1}^{N}\hat{f}(x_S, X_{iC}) \tag{4}$$

Where $\{X_{1C}, X_{2C},...X_{NC}\}$ are the values of other variables $X_C$ in the dataset, $N$ is the number of instances. The partial dependence method works by averaging the machine learning model output over the distribution of the features in set C, allowing the function to illustrate the relationship between the features in set S (of interest) and the predicted outcome. By averaging over the other features, we obtain a function that is dependent solely on the features in set S. In other words, partial dependence reveals the relationship between the targets (outputs) and the explanatory factors in $x_S$ (explanatory factors that we are interested in).

### Model evaluation and comparison

The 36 models established using the data from June 2021 to May 2022 were employed to forecast hospitalization indicators from June 2022 to January 2023 ('future' data to the model, 2308 data points for each model) using relevant explanatory factors. The prediction accuracy of the models was evaluated using MAE and NMAE to compare and select the types of prediction (i.e. WBE-based, case-based, and record-based), hospitalization indicators (i.e. weekly new hospitalizations, census inpatient sum, census inpatient average) and leading times (i.e. 1–4 weeks).

### Necessity of periodic updates

The WBE-based models for weekly new hospitalizations (selected based on the model evaluation results) under 4 leading times were further used to investigate the need for periodic updates to the model structure. In progressively learning models, the training dataset used for random forest models was progressively updated every four weeks from June 2022 to January 2023. This means that at week i, a new set of models was established utilizing the data from the previous weeks up to week i − 1 and used for prediction until the next update (in week i + 4). The construction of the progressive learning models followed the procedure described in the previous section, with 80% of the data used for training and 20% used for testing. The performance of the batch and progressive learning models was assessed using MAE and NMAE to compare the predicted results to the actual admission record.

### Transferability of progressive learning models

The transferability of progressive learning models established in the section above was tested in another 60 counties from 30 states in the USA from June 2022 to January 2023 (details provided in Supplementary Table S4, 1459 data points for each model). The wastewater surveillance data for these counties was obtained from Biobot, while other explanatory factors (e.g. CCVI indexes, precipitation etc.) and hospitalization records were obtained from relevant sources as described in earlier sections.

Additionally, the study investigated the impact of localized data updates on model transferability. Data in these 60 counties from June 2022 to January 2023 was progressively incorporated into the existing progressive learning model under the same update frequency. This means, at week i, the data in these 60 counties from June 2022 to week i − 1, was incorporated into the dataset used for establishing the progressively learning model, providing the prediction till the next update (week i + 4). Model predictions were compared with actual admission records and evaluated using MAE, and NMAE.

### Software used for statistical data analysis and data visualization

We conducted all analyses and data visualizations using R (version 4.2.0, R Foundation for Statistical Computing, http://www.R-project.org/) with packages including reshape2 (version 1.4.4), dplyr (version 1.0.9), tidyr (version 1.2.2), randomForest (version: 4.7-1.1), frPermute (version 2.5.1), Pdp (version 0.8.1), ggplot2 (version 3.3.6), corrplot (version 0.92), ComplexHeatmap (version 2.15.1), and USmap (version 0.6.2).

### Reporting summary

Further information on research design is available in the Nature Portfolio Reporting Summary linked to this article.

## Data availability

We conducted secondary data analyses of publicly available data with data source listed below.

County-level wastewater surveillance data: biobot.io/data

County-level hospitalization data: COVID-19 Reported Patient Impact and Hospital Capacity by Facility | HealthData.gov

County-level COVID-19 Community Vulnerability Index (CCVI) indexes: https://precisionforcovid.org/ccvi

County-level vaccination coverage: https://data.cdc.gov/Vaccinations

Daily county-level COVID-19 cases: https://usafacts.org/visualizations/coronavirus-covid-19-spread-map

Daily temperature and precipitation in each county: Environmental Protection Agency https://www.ncdc.noaa.gov/cdo-web/datatools/lcd

The data for USA map in Fig. 2a was sourced from R package 'USmap' (version 0.6.2) where relevant shape data was provided by the US Census bureau (open access): https://data.census.gov/map?layer=VT_2021_040_00_PP_D1&loc=43.3751,-113.1138,z2.6270

Secondary data (wastewater surveillance data and relevant weather, CCVI, and hospitalization data) used in the analyses could be shared by contacting the corresponding author.

## Code availability

The code for analysis and figures is provided at[68]: https://doi.org/10.5281/zenodo.8128697.

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

## Acknowledgements

X.L. thanks the funding support from the Australian Academy of Science through W H Gladstones Population and Environment Fund. Q.W. acknowledges the Australian Research Council Future Fellowship (FT200100264). The authors also acknowledge the valuable suggestions provided by Prof. Dayong Jin at University of Technology Sydney during the manuscript preparation.

## Author contributions

X.L. conceptualized the study, curated the data, developed the methodology, conducted formal analysis, wrote and edited the manuscript. H.L., L.G., and T.Z. contributed to the data curation. S.S, S.K., and M.V. reviewed and edited the manuscript. Q.W. contributed to the study design, supervision, and reviewing and editing of the manuscript.

## Competing interests

The authors declare no competing interests.
