## [Peer Review File · Nature Communications]

Wastewater-based epidemiology predicts COVID-19-induced weekly new hospital admissions in over 150 USA countiesREVIEWER COMMENTS

Reviewer #1 (Remarks to the Author):

This study evaluates models to predict COVID-19 hospitalizations in the United States using sewage data and other publicly available sources. This is an important topic and could potentially be a useful contribution. However, further work is needed to clarify the methods and present the results in such a way that will be useful and acceptable for the public health community.

The description of the model development and evaluation is not clear. Are the training/test/validation sets based on groups of counties or groups of time points? Is the training done in a way that is blind to the future (e.g., by sequentially adding time points). If not, the validity of the models and validation data would be in question. It sounds like this is maybe what is being done with the progressive learning model

The structure of the paper is hard to parse. There is a lot of description of the methods in the results section, but not enough detail to evaluate what is being done.

The authors state that "The underestimation of peaks is likely related to the skewed data distribution to higher ranges for hospitalization and ICU admission records" Why did they opt not to use a log-linked GLM (Poisson or negative binomial) for the random forest, or at a minimum transform the data to be less skewed prior to model fitting?

The COVID forecasting Hub has established a set of standardized forecasting targets that are used by groups using a variety of methods (https://covid19forecasthub.org/reports/single_page.html#National_level). It would be helpful to align prediction targets to these standardized targets

Fig 1B would be more useful if it also show distribution of counties not included (this would show if SVI in included counties is representative of entire US)

Fig 1C...I think what this is trying to show is case 0-7, 0-14, 0-21, 0-28 days. This should be relabeled for clarity. Not sure panel 1C is particularly useful.

Descriptive stats of CCVI—these numbers are on an arbitrary scaling and meaningless to most people (e.g., what does a CCVI of 0.7 mean? Is that high? Average?)

The Prediction target is effectively a rolling average of hospitalization (e.g., average, 0-28 days out). This means prediction targets are overlapping. If you predict from week 1, window is weeks 2-5), if predict starting week 2, window is 3-6, etc. This might influence things when training model. There might also be other windows of relevance not captured with this approach. For example, if there is a 1 week lag between sewage and hospital increases, then including days 0-6 in the prediction window will affect ability to capture patterns accurately.

A time series plots of wastewater and hospitalization data would be useful, perhaps from some representative counties

4B—some further guidance on interpreting these figures is needed. This is not a commonly used analysis/plot type in the epidemiological literature, so some guidance in the text on interpretation would be helpful

Are interactions among variables included in the model (rain*sewage)

Fig 5 hard to interpret...might be more useful to show time series from representative counties with overlay of different predictions.

Since all of the data are publicly available, analytical code needs to be included in a repository so that other can validate and use this approach

There is some precedent of using wastewater data to predict hospitalizations. See Peccia Nature Biotechnology, which used a distributed lags model to link sewage concentrations with hospitalization

The paper needs some grammatical editing.

Reviewer #2 (Remarks to the Author):

The authors have used Biobot data of wastewater surveillance for SARS-CoV-2 to forecast hospitalizations. It is a natural evolution of the wastewater surveillance data. One big question I have is if the authors have checked in with Biobot who generated all the wastewater surveillance data. I realize it is in the public domain, but as a courtesy the authors should reach out. I have seen some presentations from the Biobot data scientists, and I would be highly surprised if they are not working on forecasting hospitalizations using their data. This feels like a bit of a scoop, without proper acknowledgement/collaboration. I could be wrong – perhaps Biobot is more than happy to see their data analyzed and applied. I have a number of further questions regarding the article, as well as a few concerns:

1. Line 23, should be “easing”?
2. Line 25 – I’m not sure I would claim that low-cost COVID-19 prediction is essential. Helpful, certainly.
3. Line 25 – this is not the first time to use wastewater surveillance to predict hospitalizations. I recommend allowing the article to stand on its own merits and not the false claim of “first”.
4. Lines 25-29 – perhaps break up this sentence?
5. Lines 31-33 – for the results, can the authors place these in terms of relative percentage? With an absolute mean error of 20 hospitalizations, how much percent error is that? Is the average 20 hospitalizations, so a 100% error? Or is the average 100 hospitalizations so only a 20% error? From figure 1C it looks like an error of 20 hospitalizations is quite a lot.
6. From the abstract, I am left wondering how well wastewater forecasts compare to forecasts using clinical surveillance data. Clinical surveillance is “free” to local health departments (cost passed onto individual health insurance) compared to wastewater surveillance which cannot pass this cost onto individual health insurance. And so if the goal is to make a low-cost prediction for hospitalizations, why not use clinical surveillance? How does wastewater compare to test positivity and incidence in terms of accuracy of predicting hospitalizations?
7. Lines 54-57 don’t follow too well. The changing in test availability, behavior, and reporting is what is driving the poorer clinical surveillance data. Not the change in contact tracing, mask mandates, and vaccine promotion.
8. Line 68. None of these studies has compared the cost of wastewater surveillance to clinical surveillance. It is an added cost to public health budgets, whereas from a public health budget perspective clinical surveillance is free. Lines 71-74 are more accurate, as compared to a random community sample the cost is lower.
9. Lines 74-75. A number of studies have already shown that levels of SARS-CoV-2 in wastewater correlates from hospitalizations, some as early as 2020. The authors need to do a better literature review, cite these studies, and then show how their study builds upon them. Some examples from a very quick and cursory google search: Nattino et al. in JAMA, Zhan et al. in ACS EST Water, Galani et al. in Sci Total Environment.
10. The authors are using the Covid-19 community vulnerability index, and cite a pre-print from 2021 that has not been published as of 2023. The CDC created the social vulnerability index which does correlate with COVID-19 measures. How is the CCVI different from the CDC’s SVI? I recommend the authors use the SVI as the CCVI is not peer-reviewed.
11. Line 90. Did the authors collect any data? The wastewater data was publicly available – what about the other measures the authors used? It looks like the authors conducted secondary data analyses of publicly available data, a very important distinction.
12. Line 494, 500. Should read data were retrieved or obtained, not collected. (As is done in line 504). Wu et al. should be cited in the introduction.

13. Line 512 – the authors state the CCVI was developed by the US CDC. Do they mean the SVI? It looks like it was actually developed by the Surgo Foundation and includes both proprietary and non-proprietary data. Digging into the Melvin et al. article from 2020 it looks like SVI plus human movement. The SVI would be more relevant for the time period the authors are analyzing (once human movement was less disrupted).
14. Lines 526-529. Dilution from precipitation would only matter for combined sewers. Do the authors not have access to wastewater treatment plant flow data?
15. The authors have not indicated how they handled the temporal or spatial scales of wastewater. Wastewater data would have varying temporalities with some sites providing weekly data and other sites providing more frequent data. The methods suggest a weekly temporal scale of data analysis using 7, 14, 21, and 28-day measures of hospitalizations. Hospitalizations would be on a daily scale. How have the authors matched those temporal scales? Also, on the spatial scale – how have the authors matched up the county data to the wastewater data? Numerous counties in the Biobot data have multiple wastewater treatment plants. Including them all with a direct match to hospitalization data would artificially inflate the dataset. The authors have also not considered wastewater surveillance coverage either in their analysis nor in their discussion.
16. For the out of model predictions, did the authors randomly select data or did they randomly select treatment plants or time periods? I would prefer the authors to randomly select treatment plants or time periods rather than simply randomly selecting data. I see this now in lines 333, but would the authors state so in the methods?
17. From Figure 2 the correlations between wastewater levels and hospital measures are either 1 or near 1, which to me suggest some type of serial autocorrelation going on – either temporal or spatial. How have the authors accounted for this autocorrelation? Could the artificial duplication of data also be contributing to these hard to believe perfect correlations?
18. From Figure 4 it looks like the authors broke out their CCVI – this was not stated in the methods.
19. Wastewater had the greatest explanatory power for hospitalizations with a 7-day lead time. Why then do the authors select 14- and 28-day lead times to highlight? Considering the biology the authors highlight lines 386-393, the seven-day lead time makes sense. (Fecal shedding about when someone would test positive).
20. Figure 1 looks okay. The authors might consider log-transforming the measures as in 1C they are so skewed.
21. Figure 2 looks good
22. Table 1. I've never seen an R-squared of 0.98 when modeling hospitalizations – there is just too much random chance. And all of the outcomes are above 0.9. It raises a bit of an eyebrow – it would be wise to dig into the modeling and data to make sure these are real. I have a hard time reconciling that result with Figures 3, 4a, and 5.

Reviewer #3 (Remarks to the Author):

Summary

In "Wastewater-based epidemiology predicts COVID-19-induced hospital and ICU admission numbers in over 100 USA counties," the authors explore the predictability of hospital and ICU rates across the United States using a wide variety of potential predictors including wastewater-based estimates of disease. They explore relationships between the predictor and response variables, construct random forest prediction models, test the fits of these models, and validate their models with additional data held back from the fitting procedure. Overall I find the paper to be well-written and topical, as wastewater based epidemiology is a nascent field and there remain many questions regarding its utility as a surveillance system. However, I have major concerns about the data and methodology as presented in the current work, which I describe in detail below.

Major comments

Based on the title and text, the study purports to predict county-level COVID-19 hospital and ICU admission counts. I investigated the "University of Minnesota COVID-19 Hospitalization Tracking Project" cited by the authors (note that the reference number cited does not link to the data

source), and I am fairly certain that it provides access to hospitalization and ICU census counts, not admission counts. Hospital census describes the raw number of people hospitalized or in the ICU with COVID-19 on a given day, where admissions describe the new patients arriving on to a hospital on a given day. The authors should strongly check which data they are using - from my eye, the values in figure 3 seem to be too large and too smooth to be daily admissions. If I'm correct, then the authors should change the text to reflect this difference. More importantly, this dramatically impacts the interpretation of the results. There are much higher correlations between day-to-day hospital and ICU census counts overall, as the counts from today impact the counts from tomorrow. This causes issues with model fitting, parameter estimation, and forecasting as discussed in (<https://royalsocietypublishing.org/doi/10.1098/rspb.2015.0347>). For these reason, all forecasting efforts for COVID-19 have focused on hospital admission counts (e.g. <https://github.com/reichlab/covid19-forecast-hub> and <https://www.pnas.org/doi/10.1073/pnas.2111870119>). Otherwise they've focused on daily new reported cases or deaths. The authors could theoretically switch the analysis to use WBE for predicting case counts instead of the hospitalization/ICU census counts.

The time periods for the analysis are confusing as written, hindering the ability to evaluate all of the results fully. It appears that the authors have split up the time period for training, testing, and validation based on the table, but they also appear to show in-sample fits for the whole time period (Figure 3). On top of that they also describe 5-fold cross validation for understanding the importance of explanatory factors. I think it would greatly enhance the clarity of the paper if the authors outline how all of these components fit together with one another alongside including the dates of analysis for each figure/table caption as well. For example, the predictions in Figure 5 for the batch model don't appear to perform that well, but it was my understanding that is the same prediction model described in the above figures showing strong predictive ability.

The predictions shown in Figure 3 and in the table are remarkably good. However, it is difficult to fit these results into the findings of the larger forecasting field. Are the authors claiming that their model can make extremely accurate 4 week predictions? If so, this would be above and beyond what other teams have been able to do in the COVID-19 forecast hub, for example see: <https://forecasters.org/blog/2021/09/28/on-the-predictability-of-covid-19/>. Given this performance, it would be useful to understand how these predictions compare with alternative models such as the null model used in the forecast hub, alongside models that are built on other predictors. For example, are the hospitalization predictions equally as good if one only uses previous hospitalization data to make the predictions, or does one really need WBE data? In general, the question is, why have the author developed such accurate forecasts, is it the data, the model, something else? Any explanation and comparison with the field would be helpful. Furthermore, It would be useful to include more of the predictor variables in the analysis of variable importance. Would WBE still be chosen as most important over the more traditional data streams (e.g. cases, hospitalizations, or ICU counts)?

Minor comments:

For the statement: "Our study demonstrated the potential of using WBE as a cost-effective method to provide early warnings for healthcare systems." The authors did not include any cost-effective analysis comparing predictor variables, so I would suggest removing that from the claim. It would be extremely helpful to include some example time-series for all of the time-dependent predictor variables. I am surprised that WBE performs as well as case counts in predicting hospitalizations/ICU census, and it would be useful to see the time-series to better visualize the relationship.

Figure questions

Fig 1C shows time-series data as points and a box-plot. It is difficult to glean any information from these plots and I would suggest that simple time-series might be more interpretable. Also, shouldn't the 7d, 14d, 21d, and 28d healthcare values all have the same appearance? A 14d value is just a 7d value lagged by 7 days.

Fig 2 - I think percent positivity is more interpretable and common than the reverse positive metric. This would also help put the correlation for that metric positive than the negative one

currently shown

Fig 4 needs further explanation. Why do explanatory factors go to 100? What does explanatory factors mean? What is partial dependence and how is it defined?

It's not clear that Fig 6 is showing strong out-of-sample prediction performance. There is a clear bias towards positive residuals for hospitalizations and negative residuals for ICU. Also, the blue outlier county may be throwing off evaluation metrics for hospitalizations. For example, if you remove that county, then the relationship for hospitalizations looks pretty much like a horizontal line. In general I would suggest using a larger data set for this out-of-sample analysis.

Dear Reviewers,

We are grateful for the constructive comments received from the reviewers, which have helped us to further improve the quality and clarity of the manuscript.

We appreciate the opportunity to revise this manuscript and have carefully evaluated and addressed all comments and amended the manuscript accordingly. Manuscript ID: NCOMMS-22-51169.

Below are our detailed responses to the reviewers' comments point by point. The comments from reviewers are in **black**, responses from the authors are in **blue**, and revisions to the manuscript are in **red**.

We would be happy to address any further comments that you might have.

Kind regards,

Prof. Qilin Wang

Centre for Technology in Water and Wastewater, School of Civil and Environmental Engineering, the University of Technology Sydney, Ultimo, NSW, 2007, Australia

Reviewer #1 (Remarks to the Author):

This study evaluates models to predict COVID-19 hospitalizations in the United States using sewage data and other publicly available sources. This is an important topic and could potentially be a useful contribution. However, further work is needed to clarify the methods and present the results in such a way that will be useful and acceptable for the public health community.

We have thoroughly revised our manuscript to accommodate all the suggestions. For better cross-reference purposes, we have labeled the comments from Reviewer 1 in numbers.

1. The description of the model development and evaluation is not clear. Are the training/test/validation sets based on groups of counties or groups of time points? Is the training done in a way that is blind to the future (e.g., by sequentially adding time points). If not, the validity of the models and validation data would be in question. It sounds like this is maybe what is being done with the progressive learning model.

To better clarify this, we have revised the manuscript in three stage: (1) model establishment, (2) model evaluation, (3) model transferability.

In the model establishment stage, the data from 99 counties in June 2021- May 2022 (12 months) was used for finding an appropriate structure for the model. To accommodate the comments from reviewers, we have made the following changes in model establishment stage for WBE-based predictions:

- Indicators: we included three types of hospitalization indicators: 1) weekly new admission (Hos_wn), 2) the total number of patients who stayed in an inpatient bed during the week (census inpatient sum, Hos_cs), and 3) the daily average number of patients who stayed in an inpatient bed in the week (census inpatient average, Hos_ca). However, due to the unavailability of ICU data, we removed the predictions for ICU admission indicators that were present in the previous manuscript.
- Leading times: we changed the leading times to four types, covering the upcoming week (Hos1w), as well as the second (Hos2w), third (Hos3w), and fourth weeks (Hos4w). This change was made based on suggestions from reviewers to avoid potential overlaps in our previous manuscript (0-7 days, 0-14, ..., 0-28 days in our previous

version). In the revised version, wastewater data from week i was used to predict hospitalization indicators for week $i+1$, $i+2$, $i+3$, and $i+4$.

In model establishment, the main goal is to find an appropriate model structure to describe the relationship between the targets (hospitalization indicators under different leading times) and the explanatory factors (input variables). The data points obtained during June 2021-May 2022 (12 months) were used for model establishment. Training, test, and validation data sets were randomly selected from the data based on a ratio of 70%, 15%, and 15%, respectively, regardless of their counties and time points. The reason for the randomized data selection is to make sure that the model established can describe the generalized pattern within the datasets (from June 2021-May 2022). The training set was used to train the random forest models, while the validation set was used in conjunction to optimize model structures. The test set was used to evaluate the model's prediction capability over unseen data during this period (June 2021-May 2022). Till a satisfying performance was achieved by the validation and test data sets, the established model for each target was further assessed for their capability in predicting future data in the model evaluation stage.

In model evaluation stage, the model established above (batch model) was used to predict the hospitalization indicators (Hos_wn, Hos_cs, Hos_ca) during June 2022-January 2023 ('future' to the models). The prediction capability and performance of these models toward different targets were compared. The hospitalization indicator with the best performance (Hos_wn based on results) was further used to evaluate the necessity of periodic updates based on the most up-to-date information and the transferability of the model to other counties (not included in the model establishment). For progressively learning models, the data used for model establishment was progressively updated every four weeks from June 2022. This means, at week i , a new set of models was established using the datasets till week $i-1$ and used for the prediction till the next update (in week $i+4$). The performance of the batch model and progressively learning model was compared and then the progressively learning models were further used for testing the transferability of the model in another 60 counties from June 2022 to January 2023. To clarify this, we have revised the method section.

Line 536-539 in *Model establishment using random forest algorithm*:

For model establishment, data from June 2021 to May 2022 (3162 data points for each target, 12 months) were utilized to describe the patterns for each target through the random forest algorithm in R (ver 4.2.0, R Foundation for Statistical Computing, <http://www.R-project.org/>).

Line 556-562 in *Model establishment using random forest algorithm*:

the data was randomly divided into three parts, training set (70% of data), validation set (15% of data), and test set (15% of data), regardless of their counties and time points. The training set was used to train the random forest models, and the validation set was used in conjunction to optimize model structures. The test set was then used to evaluate the model's prediction capability over unseen data during the model establishment stage. The rationale behind the randomized data selection was to ensure that the models developed could accurately describe the generalized patterns within the datasets from June 2021 to May 2022.

Line 594-599 in *Model evaluation and comparison*:

models established using the data from June 2021 to May 2022 were employed to forecast hospitalization indicators from June 2022 to January 2023 ('future' data to the model, 4616 data points for each model) using relevant explanatory factors. The prediction accuracy of the models was evaluated using MAE and NMAE.

Line 601-610 in *Necessity of periodic updates*:

The WBE-based models for Hos_wn (selected based on the model evaluation results) under 4 leading times were further used to investigate the need for periodic updates to the model structure. In progressively learning models, the training dataset used for random forest models was progressively updated every four weeks from June 2022 to January 2023. This means that at week i , a new set of models was established utilizing the data from the previous weeks up to week $i-1$ and used for prediction until the next update (in week $i+4$). The construction of the progressive learning models followed the procedure described in the previous section, with 80% of the data used for training and 20% used for testing.

Line 612-613 in *Transferability of progressive learning models*:

The transferability of progressive learning models established in the section above was tested in another 60 counties from 30 states in the USA from June 2022 to January 2023

2. The structure of the paper is hard to parse. There is a lot of description of the methods in the results section, but not enough detail to evaluate what is being done.

Thanks for the suggestions. We have moved the descriptions of the methods in the results section (which was majorly about the model establishment) into the model establishment

section in the materials section (as shown in the changes mentioned in the response to the 1st comment from Reviewer 1).

3. The authors state that “The underestimation of peaks is likely related to the skewed data distribution to higher ranges for hospitalization and ICU admission records” Why did they opt not to use a log-linked GLM (Poisson or negative binomial) for the random forest, or at a minimum transform the data to be less skewed prior to model fitting?

We used the term "skewed data" here to describe the nature of our dataset, which has fewer data points in higher ranges for hospitalization records, making it difficult for the model to capture the relevant patterns in higher ranges. Generally, tree-based models, such as the random forest algorithm used in our study, are not affected by the skewness of the data, as they do not assume any prior distribution ¹.

To provide further clarity, we compared the model's performance in predicting the census inpatient average in the week using both transformed and non-transformed data in the model establishment stage. We used the Box-Cox transformation to convert the non-normal data to normal distributions for the explanatory variables and targets ². The normality of the data was checked through Jarque-Bera Normality Test ³. The comparison of the model's performance between the transformed and non-transformed data showed no significant difference, indicating that the data distribution did not affect the model's performance (Table S5). To clarify this, we have added Table S5 to the supplementary information and added the following lines to the manuscript.

Table S5. Model performance using non-transformed data and Box-Cox transformed data.

	Non-transformed data				Transformed data			
	Training	test	validation	all	Training	test	validation	all
R	0.98	0.87	0.88	0.95	0.97	0.86	0.87	0.94
MAE	2.80	6.35	5.52	3.74	2.81	6.38	5.34	3.73
NMAE	0.18	0.38	0.37	0.24	0.18	0.38	0.36	0.24

Line 549-552:

As the skewness of the data does not affect the structures and performance of random forest models ¹ (which was also demonstrated in Table S5), transformations for data were not included in our study.

4. The COVID forecasting Hub has established a set of standardized forecasting targets that are used by groups using a variety of methods (https://covid19forecasthub.org/reports/single_page.html#National_level). It would be helpful to align prediction targets to these standardized targets

Thanks for the link and relevant information provided by the reviewer. The COVID-19 forecastHub provides useful information on incident daily hospitalizations at the state and national level in the USA for the next 14 days ⁴. However, hospitalization rates and patterns can vary significantly at the county level due to differences in population demographics, healthcare resources, etc., even within the same state ⁵. More granular insights for predicting hospitalization at county-level are more ideal for practical application.

In our study, our aim is to provide guidance to local health facilities on resource allocation and worker deployment to combat COVID-19. Given the potential long-term scenario of 'living with COVID-19,' we decided to use wastewater-based epidemiology (WBE) to predict hospitalizations on a **weekly basis** at the **county level**. The reason for weekly based prediction (rather than daily) is:

- The shedding dynamic of SARS-CoV-2 RNA to sewers.
The infection status among the population is reflected by the SARS-CoV-2 RNA concentration (C_{RNA}) in wastewater. The viral RNA shedding load from infected individual to sewers (through feces, sputum and other bodily fluids) peaks in the first couple of days before, to a week after, the symptom onset ^{6, 7, 8, 9}. Thus, the changes of C_{RNA} in wastewater samples are more sensitive to the variations in the numbers of COVID-19 infections at their early infection stages. This also means that the contribution of each infected individual can be detected in a certain period (more than a week) rather than a day. This is quite different to the incident case or hospitalization record (or ensembled predictions such as COVID-19 forecastHub), where the case record, or hospitalization record for a certain individual is only reported once in a certain period (until they re-infected or re-admitted to a hospital).
- The potential impact of variant diversity and population demographics in different counties.

Whether and when an infected individual is admitted to a hospital is also dependent on other factors, such as the viral variant, race/ethnicity, vaccination, and chronic conditions ^{10, 11, 12, 13, 14, 15}. SARS-CoV-2 variants evolve over time and exhibit distinct

regional patterns across the nation ¹⁶. Moreover, population demographics and vaccination coverage vary across counties and over time. These factors can potentially affect the time between viral shedding and hospitalization. This is evident by the various leading times reported by previous WBE studies for hospitalization, which vary from 1-5 days to 8-18 days in different regions, during different stages of the outbreak ^{17, 18, 19, 20}. Thus, for a large-scale prediction (both geological and temporal), predicting the hospitalization numbers within a certain period (weekly), rather than daily would be more feasible.

- The turn-over time of wastewater samples.
Considering the logistics of wastewater sampling and laboratory analysis, the turn-over time for wastewater samples can vary from a few hours to several days, depending on the capacity of the testing facility. Therefore, for WBE-based prediction, a lower sampling frequency (weekly rather than daily) and longer prediction period (up to 4 weeks, rather than 14 days) would be more practical and feasible, in terms of analytical cost, time, and effort.
- The weekly resource allocation and staff arrangement in most healthcare systems. Most hospitals allocate their resources and staff on a weekly basis for incoming patients ²¹.

We would also like to acknowledge that our weekly-based predictions have limitations. Although weekly predictions would meet the weekly resource allocation and staff arrangement in most healthcare systems and more viable in most of regions, for certain regions where a high-resolution (such as daily) prediction is required, the case-based or hospitalization-record-based (or ensembled) prediction might be more suitable than WBE-based predictions

To reflect the discussion here, we have added the following lines to the manuscript.

Line 56-61:

Ensembled probabilistic forecasts for daily incident hospitalizations were also provided based on the forecast from multiple teams at state and national levels ⁴. However, hospitalization rates and patterns can vary significantly at the county level due to differences in population demographics, healthcare resources, etc., even within the same state ⁵. More granular insights for predicting hospitalization at county-level are more ideal for practical application.

Line 66-80:

Few studies have reported the association between C_{RNA} in wastewater (or primary sludge) with hospitalizations^{22,23} and endeavored to create surveillance models for forecasting hospital admissions with various leading times ranging from 1 to 8 days^{17, 18, 19}. Nevertheless, these observations and models were developed using data from only a few localities for a short period (a couple of months). A recent study revealed the predictive potential for state-level hospitalization occupancy (census hospitalizations) with a leading time of 8-18 days in Austria²⁰. However, population demographics (such as race/ethnicity, vaccination, chronic conditions, etc.) that have been clinically observed impacting the COVID-19 symptom severity^{10, 11, 12, 13, 14, 15} were not considered in all these precedent prediction models. This limits their temporal and geographic scope, thus making it uncertain whether they (both the model and the hospitalization indicators predicted) could be generalized to other areas. Considering that hospitals/healthcare facilities often allocate their resources and workers on a weekly basis for upcoming patients²¹, a large-scale (temporal and geological) prediction system for hospitalizations at the county level on a weekly basis would be more informative for local healthcare facilities, which unfortunately is lacking.

Line 455-460:

In addition, considering the regional variations in the leading time and the turnover time for sample analysis (up to several days), the WBE-based models predicted hospitalizations on a weekly basis. Although this meets the weekly resource allocation and staff arrangement in most healthcare systems, for certain regions where a high-resolution (such as daily) prediction is required, the case-based or record-based (or ensembled) prediction might be more suitable than WBE-based predictions.

5. Fig 1B would be more useful if it also show distribution of counties not included (this would show if SVI in included counties is representative of entire US)

We have incorporated the CCVI distribution for counties across the entire USA (Fig. S1) in the supplementary information. CCVI indexes for each theme generally followed a normal distribution between 0-1 in all the counties in the USA. The CCVI indexes in these counties we included in our paper ranged from 0.02-0.99, which are representative of most USA counties (Fig. S1).

We have added the Fig.S1 into the supplementary information and added the following lines to the manuscript.

Figure S1. The CCVI of the counties included in the study and other counties in the USA. The CCVI distribution is represented by a box plot (left), individual points (middle), and a density plot (right) for each index.

Line 106-108:

The CCVI indexes in these counties ranged from 0.02-0.99, which are representative of most USA counties (Fig. S1) ^{11, 24}.

6. Fig 1C...I think what this is trying to show is case 0-7, 0-14, 0-21, 0-28 days. This should be relabled for clarity. Not sure panel 1C is particularly useful.

The Fig. 1c from the previous manuscript displayed the distribution of hospitalization indicators, and COVID-19 cases in the next 0-7, 0-14, 0-21, and 0-28 days, along with weekly SARS-CoV-2 RNA in wastewater (C_{RNA}). However, as suggested by reviewers, there were overlaps between the target periods. To address this, we revised our approach and used weekly intervals to predict hospitalization indicators for the upcoming week (Hos1w) and the second (Hos2w), third (Hos3w), and fourth weeks (Hos4w) after the wastewater sampling. This means that wastewater data from week i was used to predict hospitalization indicators for week $i+1$, $i+2$, $i+3$, and $i+4$. The COVID-19 cases were thus changed into weekly basis.

Since we changed the prediction leading time on weekly basis, in the revised version, we present the weekly new admission (Hos_wn) in each county over the 20 months of the study period as Fig. 2c, and included Hos_cs, Hos_ca, weekly COVID-19 cases, and weekly C_{RNA} in

wastewater in supplementary information Fig. S2. To reflect the changes, we have added the following lines to the manuscript.

Line 121-129:

Three indicators for hospitalization numbers were used including: 1) weekly new admission (Hos_wn), 2) total number of patients who stayed in an inpatient bed during the week (census inpatient sum, Hos_cs), and 3) daily average number of patients who stayed in an inpatient bed in the week (census inpatient average, Hos_ca). The Hos_wn, Hos_cs, and Hos_ca indicators had a range of 0-100 patients/100k population, 0-1220 patients/100k population, and 0-175 patients/100k population, respectively. The highest peaks were observed during August 2021 to February 2022 (Fig. 2c, Fig.S2). The C_{RNA} of wastewater samples ranged from 0.4 to 9000 copies/mL (IQR:101.54 - 546.53 copies/mL) (Fig. S2). The weekly new COVID-19 cases ranged from 0- 4065 incidence/100k population (IQR: 48-271 incidence/100k population).

Line 145-147:

The records of these three indicators over the course of the upcoming week (Hos1w), as well as the second (Hos2w), third (Hos3w), and fourth weeks (Hos4w), were used as targets, providing prediction leading times of 1, 2, 3 and 4 weeks, respectively.

Fig. 2: Geological location, COVID-19 Community Vulnerability Index (CCVI), and average weekly new COVID-19-induced hospitalizations in each month in the 99 counties involved.

a. The state (filled in color) and counties (indicated by the dot with dot size reflecting the population size of the county) involved in the model establishment. b. The CCVI of the counties involved in the model establishment is represented by a violin plot for each index. c. The average weekly new hospitalization admission numbers of each month from these 99 counties. The data before June 2022 (12 months) were used for model establishments while data after June 2022 (8 months) were used for model evaluation.

Figure S3. The weekly new COVID-19 cases (cases/100k population), C_{RNA} in wastewater samples, Hos_cs (total number of patients stayed in an impatient bed during the week), and Hos_ca (daily average number of patients stayed in an impatient bed during the week), in each county during the study period. Grey cells in the heatmap indicates missing values. The color gradient in each cell represents the monthly average of each indicator.

7. Descriptive stats of CCVI—these numbers are on an arbitrary scaling and meaningless to most people (e.g., what does a CCVI of 0.7 mean? Is that high? Average?)

For better management and policy-making, COVID-19 Community Vulnerability Index (CCVI) was established by Surgo Foundation and was also used by the Centers for Disease Control and Prevention (CDC) for COVID-19-related response in the USA ²⁵. At the county level, CCVI considers 40 measures from census data, covering 7 themes including i) socioeconomic status; ii) minority status and language, iii) housing type, transportation, household composition, and disability (“household and transportation” hereafter), iv) epidemiological factors, v) healthcare system, vi) high-risk environment and vii) population density, with an overall VI summarizing these 7 aspects ^{11,24}. The CCVI overall score as well as the 7 theme indices range from 0 to 1, with 1 representing the most vulnerable area and 0 representing the least vulnerable area ^{11,24}. To clarify this, we have revised the following lines in the manuscript.

Line 489-491:

For better management and policy-making, COVID-19 Community Vulnerability Index (CCVI) was established by Surgo Foundation and used by CDC for COVID-19-related response in the USA ²⁵.

Line 494-500:

At the county level, CCVI considers 40 measures from census data, covering 7 themes including i) socioeconomic status; ii) minority status and language, iii) housing type, transportation, household composition, and disability (“household and transportation” hereafter), iv) epidemiological factors, v) healthcare system, vi) high-risk environment, and vii) population density, with an overall VI summarizing these 7 themes ^{11,24}. The CCVI overall score as well as the 7 theme indices range from 0 to 1, with 1 representing the most vulnerable area and 0 representing the least vulnerable area ^{11,24}.

8. The Prediction target is effectively a rolling average of hospitalization (e.g., average, 0-28 days out). This means prediction targets are overlapping. If you predict from week 1, window is weeks 2-5, if predict starting week 2, window is 3-6, etc. This might influence things when training model. There might also be other windows of relevance not captured with this approach. For example, if there is a 1 week lag between sewage and hospital increases, then including days 0-6 in the prediction window will affect ability to capture patterns accurately. We subdivided this comment into two parts.

a) The Prediction target is effectively a rolling average of hospitalization and overlapping

We agree with the overlap between the previous targets and have changed the leading time in the study on a weekly basis, as the upcoming week (Hos1w) and the second (Hos2w), third (Hos3w), and fourth weeks (Hos4w) after the wastewater sampling. As mentioned in the response to the 6th comment from Reviewer 1, this means wastewater data from week i was used to predict hospitalization indicators for week $i+1$, $i+2$, $i+3$, and $i+4$. The results for the updated models are provided below (in red for detailed changes in the manuscript).

Briefly, in the model establishment stage, the established WBE-based model well described the pattern of data observed from June 2021-May 2022 for all three types of hospitalization indicators (i.e. Hos_wn, Hos_cs, and Hos_ca), with overall R values over 0.90 and NMAE within 0.30 (Table S2). When applying the established batch models for predicting the future hospitalization indicators from June 2022 to January 2023, the prediction for Hos_wn (R=0.81-0.82, NMAE=0.32-0.37) outperformed that of Hos_cs (R=0.59-0.67, NMAE=0.53-0.76) and Hos_ca (R=0.66-0.69, NMAE=0.51-0.65) regardless of the leading times (Table 1). This indicates that WBE-based predictions are likely more capable of capturing the weekly new admissions in the following weeks rather than the total or daily average number of inpatients. The prediction for Hos_wn was further progressively updated every four weeks. This reduced the MAE to from 4 patients/100k population in the batch models to 3 patients/100k population in progressively learning models, and NMAE from 0.32-0.37 in the bath models to 0.28-0.29 in progressively learning models. In the model transferability evaluation stage, the progressively learning models reasonably predicted the Hos_wn in these 60 counties (not included in the model establishment) in the next 1-4 weeks after the wastewater sampling, with an average NMAE of 0.43-0.48. We further periodically updated the model with the data from these 60 new counties into the progressively learning model under the same update frequency

(4 weeks). With the data of new counties included, the average NMAE reduced to 0.31-0.35 for the next 1-3 weeks, and 0.45 for the next 4th week.

b) There might also be other windows of relevance not captured with this approach

The model established in our study aims to provide guidance for local health facilities in allocating resources and workers under COVID-19 pandemic, rather than finding an exact lagging time between the SARS-CoV-2 RNA concentration in wastewater and hospital admission. As mentioned in the response to the 4th comment from Reviewer 1, the lag time in previous studies can vary from 1-18 days in different regions and during different stages of the outbreak^{17, 18, 19, 20}. For instance, Peccia, Zulli¹⁹ observed a 1-4 day lag time between RNA load and new hospitalizations, while Kaplan, Wang¹⁷ found a lag time of 3 to 5 days based on a 3-month monitoring data. Galani, Aalizadeh¹⁸ revealed that new hospitalizations can be predicted with a leading time of 8 days, while a recent study showed that state-level hospitalization occupancy (census hospitalizations) can be predicted with a leading time of 8-18 days²⁰. The variations in the lag time may be due to population demographics, such as race/ethnicity, vaccination, and chronic conditions, which impact COVID-19 symptom severity, as observed in clinical settings^{10, 11, 12, 13, 14, 15}. However, these factors were not considered in previous models. Although these studies are crucial in confirming the early warning capability of WBE, the variations in lagging time and the limited temporal and geological scale of these studies limits their applications and transferability.

Considering that hospitals allocate their resources and workers on a weekly basis for upcoming patients, a large-scale prediction system for hospitalizations at the county level on a weekly basis would be more informative for local healthcare facilities. Thus, we choose to predict the county-level weekly COVID-19-induced hospitalization using WBE in this study. To clarify this, we have added the following lines in the manuscript.

Line 66-80:

Few studies have reported the association between C_{RNA} in wastewater (or primary sludge) with hospitalizations^{22,23} and endeavored to create surveillance models for forecasting hospital admissions with various leading times ranging from 1 to 8 days^{17, 18, 19}. Nevertheless, these observations and models were developed using data from only a few localities for a short period (a couple of months). A recent study revealed the predictive potential for state-level hospitalization occupancy (census hospitalizations) with a leading time of 8-18 days in Austria²⁰. However, population demographics (such as race/ethnicity, vaccination, chronic conditions,

etc.) that have been clinically observed impacting the COVID-19 symptom severity^{10, 11, 12, 13, 14, 15} were not considered in all these precedent prediction models. This limits their temporal and geographic scope, thus making it uncertain whether they (both the model and the hospitalization indicators predicted) could be generalized to other areas. Considering that hospitals/healthcare facilities often allocate their resources and workers on a weekly basis for upcoming patients²¹, a large-scale (temporal and geological) prediction system for hospitalizations at the county level on a weekly basis would be more informative for local healthcare facilities, which unfortunately is lacking.

Line 172-209:

Performance and leading time of the established models in predicting future admissions

WBE-based prediction models were established for all twelve targets (3 indicators × 4 leading times) using the data obtained from June 2021-May 2022 (Fig. 1). The model performance was evaluated using correlation coefficients (R), mean absolute error (MAE), and normalized MAE (NMAE) between model predictions and targets. For all three types of hospitalization indicators (i.e. Hos_wn, Hos_cs, and Hos_ca), the established WBE-based model well described the pattern of data observed from June 2021-May 2022 with overall R values over 0.90 and NMAE within 0.30 (Table S2). When applying the established batch models for predicting the future hospitalization indicators in June 2022-January 2023, the model performance for Hos_wn was greatly better than Hos_cs, and Hos_ca (Table 1). The prediction accuracy achieved R of 0.81-0.82 and NMAE of 0.32-0.37 for predicting Hos_wn, but only R of 0.59-0.67 and NMAE of 0.53-0.76 for Hos_cs and R of 0.66-0.69 and NMAE of 0.51-0.65 for Hos_ca (Table 1). This indicates that WBE-based predictions are likely more capable of capturing the weekly new admissions rather than the census average or sum of inpatients in the week.

Line 260-264:

The MAE reduced from 4 patients/100k population in the batch models to 3 patients/100k population in the progressively learning models, and the NMAE decreased from 0.32-0.37 in the batch models to 0.28-0.29 in the progressive learning models.

Line 319-322 in ***Transferability of the progressively updated WBE-based models:***

From June 2022 to January 2023, the progressively learning models reasonably predicted the Hos_wn in these 60 counties in the next 1-4 weeks after the wastewater sampling, with an

average MAE of 7-8 patients/100 k population and an average NMAE of 0.43-0.48.

Line 329-333:

We further included the data of these 60 different counties from June 2022-January 2023 into the progressively learning models with the same update frequency (4 weeks). With the data of new counties included, the MAE of the prediction for these 60 counties reduced to 4-5 patients/100 k population with an average NMAE of 0.31-0.35 for the next 1-3 weeks, and MAE of 6 patients/100 k population and NMAE of 0.45 for Hos4w.

9. A time series plots of wastewater and hospitalization data would be useful, perhaps from some representative counties.

We have included the time series plot for weekly new hospital admission (Fig. 2c) and SARS-CoV-2 RNA concentrations in wastewater (Fig. S2) for all 99 counties on monthly basis as shown in the response to the 6th comment from reviewer 1.

10. 4B—some further guidance on interpreting these figures is needed. This is not a commonly used analysis/plot type in the epidemiological literature, so some guidance in the text on interpretation would be helpful.

The figure 4B in the previous manuscript was the two-factor partial dependence plot based on C_{RNA} and other explanatory factors with significant contributions. We have updated the relevant results for the partial dependence based on new leading times as Fig. 4b and 4c. In two-factor partial dependence plot, the horizontal axis represents the values of C_{RNA} , whereas the vertical axis represents the values of the other explanatory factor (as shown in the title for each sub-figure). The color gradients in the figure indicate the partial dependence of the model (predicted target) concerning a specific x-value and y-value combination.

Explanations for the figure have been added to the caption. Additionally, the explanation for partial dependence has been added to the methods sections.

Line 576-592:

The partial dependence depicts the marginal effect of one or two explanatory factors on the outputs while controlling for other explanatory factors ²⁶. Mathematically, the partial dependence function for regression is defined as (Eq. 3).

$$\widehat{f}_S(x_S) = E_{X_C}[\widehat{f}(x_S, X_C)] = \int \widehat{f}(x_S, X_C) dP(X_C) \quad (3)$$

The x_S are the features of explanatory factors that we are interested in, and X_C are the other explanatory factors used in the machine learning model \widehat{f} . The mathematical expectation is denoted by E and probability by P . The partial function $\widehat{f}_S(x_S)$ shows the relationship between x_S feature and the predicted targets. The partial function $\widehat{f}_S(x_S)$ is estimated by calculating averages in the training data, also known as Monte Carlo method as Eq. 4:

$$\widehat{f}_S(x_S) = \frac{1}{N} \sum_{i=1}^N \widehat{f}(x_S, X_{iC}) \quad (4)$$

Where $\{X_{1C}, X_{2C}, \dots, X_{NC}\}$ are the values of other variables X_C in the dataset, N is the number of instances. The partial dependence method works by averaging the machine learning model output over the distribution of the features in set C , allowing the function to illustrate the relationship between the features in set S (of interest) and the predicted outcome. By averaging over the other features, we obtain a function that is dependent solely on the features in set S . In other words, partial dependence reveals the relationship between the targets (outputs) and the explanatory factors in x_S (explanatory factors that we are interested).

Fig.4 b and c

Fig. 4: Importance and contribution of the explanatory factors to the model predictions.

b-c: The two-factor partial dependence of Hos_wn at Hos2w (subfigure b) and Hos4w (subfigure c) on C_{RNA} and four significant explanatory factors used in the models. The horizontal axis represents the values of C_{RNA} , whereas the vertical axis represents the values of the other four explanatory factors (as shown in the title). The color gradients in the figure indicate the partial dependence of the predicted target concerning a specific x-value and y-value combination.

11. Are interactions among variables included in the model (rain*sewage)

The interactions between precipitation and concentration of SARS-CoV-2 RNA (C_{RNA}) in wastewater were not considered in our model. This is because the C_{RNA} used in our study was normalized to pepper mild mottle virus, a fecal indicator, to minimize any potential dilution-related variations²⁷. Normalization to a fecal indicator ensures that the variations in C_{RNA} are primarily driven by changes in viral shedding and excretion rather than changes in dilution due to precipitation^{23, 28, 29}. This is also evident by the significance and contributions of the explanatory factors in our study (Fig. 4a). Regardless of the leading time, C_{RNA} was found to be the most important factor for predicting Hos_wn, contributing to a significant increase in MSE (50-67%, $p=0.010$) (Fig. 4a). While, precipitation showed negligible contribution to the model prediction (1-2%, $p=0.3-0.9$). To clarify this, the following lines have been added to the manuscript.

Fig.4: Importance and contribution of the explanatory factors to the model predictions.

a: The importance of explanatory factors was ranked by the increase in %MSE (percent change in mean square error when the explanatory factor is permuted). A higher increase in %MSE corresponds to higher importance. The significance of the explanatory factors was marked as *, **, and *** representing a p value of ≥ 0.01 and < 0.05 , ≥ 0.001 and < 0.01 and < 0.001 , respectively.

Line 228-231:

While precipitation showed negligible contribution to the model prediction (1-2%, $p=0.3-0.9$), which is likely due to that C_{RNA} being normalized to pepper mild mottle virus (a fecal indicator) to minimize any potential dilution-related variations²⁷.

Line 552-554:

Considering that C_{RNA} used in the study was normalized to pepper mild mottle virus (a fecal indicator) to minimize any potential dilution-related variations²⁷, the interaction between precipitation and C_{RNA} was not included as a factor in the WBE-based models.

12. Fig 5 hard to interpret...might be more useful to show time series from representative counties with overlay of different predictions.

Figure 5 in the previous version illustrated the necessity of periodic updates for the model by comparing the prediction results between batch models and progressively learning models. We have included the relevant comparisons in 8 representative counties as new Fig. 5a in the manuscript, along with the overall prediction results (Fig.5b) and error distribution (Fig. 5c). We have also revised the main text to reflect the changes.

Figure 5 in the main text

Fig. 5. Comparison between actual admission records and the prediction results from batch models and progressive learning models for data in June 2022- January 2023.

a. The prediction results from the batch model (in blue) and progressive learning model (in orange) and the actual admission records (in black) for Hos_wn in eight representative counties. b. The prediction results from the batch model (in blue) and progressive learning model (in orange) verse the actual admission records for Hos_wn. c. The error distribution between prediction results and actual admission records for the batch model (in blue) and progressive learning model (in orange) for Hos_wn.

Line 258-264:

The random forest models developed in the previous sections for predicting Hos_wn under different leading times were progressively updated every four weeks between June 2022 and January 2023, considering the healthcare system settings (Fig. 1). The performance of the models improved greatly through progressively learning compared to the batch model (Fig. 5b and 5c). The MAE reduced from 4 patients/100k population in the batch models to 3 patients/100k population in the progressively learning models, and the NMAE decreased from 0.32-0.37 in the batch models to 0.28-0.29 in the progressive learning models

Line 276-288:

Specifically, the prediction performance of the batch and progressively learning models were illustrated in eight representative counties (selected based on population size). Predictions from both batch models and progressively learning models reached good agreements with the actual admission records (Fig. 5a), regardless of the leading time. Compared with batch models, progressively learning models reduced the MAE by 10-70% for a certain county and showed better prediction capability towards the rapid changes in the trends (both sudden rise and drops) (Fig. 5a). The population size in the county did not appear to have a clear impact on the model's accuracy, with most counties achieving comparable NMAE (0.14-0.35) (Fig. 5a). Although both batch models and progressively learning models tended to underestimate some peaks in Harrisonburg city, which had the smallest population size, this was more likely due to higher admission numbers recorded in the county, which were less frequently presented in the datasets (Fig. 2c). This resulted in fewer data points for models to learn and subsequently predict the peaks.

13. Since all of the data are publicly available, analytical code needs to be included in a repository so that other can validate and use this approach.

We have included the analytical code in GitHub as the link below. Relevant changes have been made to the manuscript.

Line 641-643:

Code availability

The code for analysis and figures is provided in the link: https://github.com/xuanbella/RF_COVID

14. There is some precedent of using wastewater data to predict hospitalizations. See Peccia Nature Biotechnology, which used a distributed lags model to link sewage concentrations with hospitalization

Indeed, there are a few precedents using wastewater data to predict hospitalization records ^{17, 18, 19}. Peccia, Zulli ¹⁹ created epidemiological models after measuring the concentration of SARS-CoV-2 RNA in primary sludge over a 3-month period, where 1-4 days of lag time between RNA load and new hospitalizations were observed. Kaplan, Wang ¹⁷ used a differential equation-based epidemiological model based on 3-month monitoring data to demonstrate that new hospitalizations could be anticipated from the SARS-CoV-2 RNA load in primary sludge with a time lag of 3 to 5 days. Galani, Aalizadeh ¹⁸ revealed that new hospitalization can be predicted by RNA load in raw wastewater with a leading time of 8 days. However, all these studies were limited to the area served by a couple of wastewater treatment plants under short-term monitoring (3-6 months). A recent study in Austria revealed the predictive potential for state-level hospitalization occupancy (census hospitalizations) with a leading time of 8-18 days ²⁰. However, population demographics (such as race/ethnicity, vaccination, chronic conditions, etc.) that have been clinically observed impacting the COVID-19 symptom severity ^{10, 11, 12, 13, 14, 15} were not considered in all these precedent prediction models. This limits their temporal and geographic scope, thus making it uncertain whether they (both the model and the prediction targets) could be extrapolated to other areas. To reflect the discussion here, we have added the following lines to the introduction.

Line 66-77:

Few studies have reported the association between C_{RNA} in wastewater (or primary sludge) with hospitalizations ^{22, 23} and endeavored to create surveillance models for forecasting hospital admissions with various leading times ranging from 1 to 8 days ^{17, 18, 19}. Nevertheless, these

observations and models were developed using data from only a few localities for a short period (a couple of months). A recent study revealed the predictive potential for state-level hospitalization occupancy (census hospitalizations) with a leading time of 8-18 days in Austria²⁰. However, population demographics (such as race/ethnicity, vaccination, chronic conditions, etc.) that have been clinically observed impacting the COVID-19 symptom severity^{10, 11, 12, 13, 14, 15} were not considered in all these precedent prediction models. This limits their temporal and geographic scope, thus making it uncertain whether they (both the model and the hospitalization indicators predicted) could be generalized to other areas.

15. The paper needs some grammatical editing.

Thanks for the suggestions. We have thoroughly checked and corrected relevant grammar errors in the manuscript.

Reviewer #2 (Remarks to the Author):

The authors have used Biobot data of wastewater surveillance for SARS-CoV-2 to forecast hospitalizations. It is a natural evolution of the wastewater surveillance data. One big question I have is if the authors have checked in with Biobot who generated all the wastewater surveillance data. I realize it is in the public domain, but as a courtesy the authors should reach out. I have seen some presentations from the Biobot data scientists, and I would be highly surprised if they are not working on forecasting hospitalizations using their data. This feels like a bit of a scoop, without proper acknowledgement/collaboration. I could be wrong – perhaps Biobot is more than happy to see their data analyzed and applied.

The wastewater surveillance data from Biobot (Wastewater Epidemiology | Covid Water Test | Biobot Analytics) was open for public access with a license allowing for reuse with proper citation (which we have cited in the manuscript). The statement for the license is quoted here ‘This work is licensed under CC BY-NC 4.0. This license requires that reusers give credit to the creator. It allows reusers to distribute, remix, adapt, and build upon the material in any medium or format, for noncommercial purposes only.’(GitHub - biobotanalytics/covid19-wastewater-data: Data repository for Biobot Analytics Nationwide Wastewater Monitoring Network)

We have also contacted the Biobot team regarding this paper. They are more than willing for us to use and build relevant models upon their data.

I have a number of further questions regarding the article, as well as a few concerns:

1. Line 23, should be “easing”?

Thanks for correcting this. We have changed the line accordingly.

Line 24-25:

Although the coronavirus disease (COVID-19) emergency status is easing, the COVID-19 pandemic continues to affect healthcare systems globally.

2. Line 25 – I’m not sure I would claim that low-cost COVID-19 prediction is essential. Helpful, certainly.

We agree with this and have changed the line as below.

Line 35-36:

Our study demonstrated the potential of using WBE as an effective method to provide early warnings for healthcare systems.

3. Line 25 – this is not the first time to use wastewater surveillance to predict hospitalizations. I recommend allowing the article to stand on its own merits and not the false claim of “first”.

We agree with this and have removed the ‘first’ as shown below.

Line 26-31:

We evaluated the feasibility of using wastewater-based epidemiology (WBE) to predict COVID-19-induced weekly new hospitalizations in 159 counties across 45 states in the United States of America (USA), covering a population of nearly 100 million. Using county-level weekly wastewater surveillance data (over 20 months), WBE-based models were established through the random forest algorithm.

4. Lines 25-29 – perhaps break up this sentence?

We agree with this and have changed the line as shown in the response above (3rd comment from Reviewer 2).

5. Lines 31-33 – for the results, can the authors place these in terms of relative percentage? With an absolute mean error of 20 hospitalizations, how much percent error is that? Is the average 20 hospitalizations, so a 100% error? Or is the average 100 hospitalizations so only a 20% error? From figure 1C it looks like an error of 20 hospitalizations is quite a lot.

Thanks for pointing out this. We have included normalized mean absolute error (NMAE) to the manuscript to evaluate the model performance. The NMAE is calculated as Eq. 2, which reflects the ratio between the absolute error and the observation values.

$$NMAE = \frac{\sum_{i=1}^n (|y_i - \hat{y}_i|)}{\sum_{i=1}^n (y_i)} \quad (2)$$

where y_i is the i^{th} observation of y and \hat{y}_i the predicted y_i value from the model. The n is the total number of data points.

As mentioned 1st comment from the Reviewer 1, we changed the manuscript into three stages for better clarity. This includes, model establishment, model evaluation, and model transferability. In addition to that, we have made the following changes in model establishment stage for WBE-based predictions to accommodate the comments from reviewers:

- Indicators: we included three types of hospitalization indicators: 1) weekly new admission (Hos_wn), 2) the total number of patients who stayed in an inpatient bed during the week (census inpatient sum, Hos_cs), and 3) the daily average number of patients who stayed in an inpatient bed in the week (census inpatient average, Hos_ca). The performance of WBE-based models on predicting these three indicators were compared and the indicator with the best prediction performance was selected for further evaluation. However, due to the unavailability of ICU data, we removed the predictions for ICU admission indicators that were present in the previous manuscript.
- Leading times: we changed the leading times to four types, covering the upcoming week (Hos1w), as well as the second (Hos2w), third (Hos3w), and fourth weeks (Hos4w). This change was made based on suggestions from reviewers to avoid potential overlaps (0-7 days, 0-14, ..., 0-28 days in our previous version). In the revised version, wastewater data from week i was used to predict hospitalization indicators for week $i+1$, $i+2$, $i+3$, and $i+4$.

In the revised manuscript, models were established using data from 99 counties in June 2021-May 2022 (model establishment stage) and then used as ‘batch model’ for predicting the hospital admissions in these 99 counties in June 2022-January 2023 (‘future’ data to the model) for model evaluation. The hospitalization indicator predicted by WBE with the best performance (weekly new admissions, Hos_wn) was selected for further evaluation. The batch model for Hos_wn was further periodically updated every 4 weeks in the model evaluation stage as the ‘progressively learning model’ for predicting the hospital admissions in these 99 counties in June 2022-January 2023. In progressively learning models, at week i , a new set of models was established utilizing the datasets up to week $i-1$ and used for prediction until the next update (in week $i+4$). The performance of the batch model and progressively learning model was compared and then the progressively learning models were further used for testing

the transferability of the model in another 60 counties (not included in the model establishment stage) from June 2022 to January 2023.

Specifically, for NMAE, in the model establishment stage, the established WBE-based model well described the pattern of data observed from June 2021-May 2022 for all three types of hospitalization indicators (i.e. Hos_wn, Hos_cs, and Hos_ca), with overall R values over 0.90 and NMAE within 0.30. When applying the established batch models for predicting the future hospitalization indicators from June 2022 to January 2023, the prediction for Hos_wn (R=0.81-0.82, NMAE=0.32-0.37) outperformed that of Hos_cs (R=0.59-0.67, NMAE=0.53-0.76) and Hos_ca (R=0.66-0.69, NMAE=0.51-0.65) regardless of the leading times. This indicates that WBE-based predictions are likely more capable of capturing the weekly new admissions in the following weeks rather than the total or daily average number of inpatients.

The prediction for Hos_wn was further progressively updated every four weeks. This reduced the MAE to from 4 patients/100k population in the batch models to 3 patients/100k population in progressively learning models, and NMAE from 0.32-0.37 in the bath models to 0.28-0.29 in progressively learning models. In the model transferability evaluation stage, the progressively learning models reasonably predicted the Hos_wn in these 60 counties (not included in the model establishment) in the next 1-4 weeks after the wastewater sampling, with an average NMAE of 0.43-0.48. We further periodically updated the model with the data from these 60 new counties into the progressively learning model under the same update frequency (4 weeks). With the data of new counties included, the average NMAE reduced to 0.31-0.35 for the next 1-3 weeks, and 0.45 for the next 4th week. Detailed changes in the manuscript are listed below.

Changes in methods section:

Line 536-539 in *Model establishment using random forest algorithm*:

For model establishment, data from June 2021 to May 2022 (3162 data points for each target, 12 months) were utilized to describe the patterns for each target through the random forest algorithm in R (ver 4.2.0, R Foundation for Statistical Computing, <http://www.R-project.org/>).

Line 563-568 in *Model establishment using random forest algorithm*:

The performance of the model was evaluated by the correlation coefficient (R), mean absolute error (MAE), and normalized mean absolute error (NMAE) using equations (1) and (2).

$$MAE = \frac{\sum_{i=1}^n (|y_i - \hat{y}_i|)}{n} \quad (1)$$

$$NMAE = \frac{\sum_{i=1}^n (|y_i - \hat{y}_i|)}{\sum_{i=1}^n (y_i)} \quad (2)$$

where y_i is the i^{th} observation of y and \hat{y}_i the predicted y_i value from the model. The n is the total number of data points.

Line 593-599 in *Model evaluation and comparison*:

models established using the data from June 2021 to May 2022 were employed to forecast hospitalization indicators from June 2022 to January 2023 ('future' data to the model, 4616 data points for each model) using relevant explanatory factors. The prediction accuracy of the models was evaluated using MAE and NMAE.

Line 601-610 in *Necessity of periodic updates*:

The WBE-based models for Hos_wn (selected based on the model evaluation results) under 4 leading times were further used to investigate the need for periodic updates to the model structure. In progressively learning models, the training dataset used for random forest models was progressively updated every four weeks from June 2022 to January 2023. This means that at week i , a new set of models was established utilizing the data from the previous weeks up to week $i-1$ and used for prediction until the next update (in week $i+4$).

Line 601-610 in *Transferability of progressive learning models*:

The transferability of progressive learning models established in the section above was tested in another 60 counties from 30 states in the USA from June 2022 to January 2023

Line 617-623 in *Transferability of progressive learning models*:

Additionally, the study investigated the impact of localized data updates on model transferability. Data in these 60 counties from June 2022 to January 2023 was progressively incorporated into the existing progressive learning model under the same update frequency. This means, at week i , the data in these 60 counties from June 2022 to week $i-1$, was incorporated into the dataset used for establishing the progressively learning model, providing the prediction till the next update (week $i+4$). Model predictions were compared with actual admission records and evaluated using MAE, and NMAE.

Changes in the results section

Line 173-186 in *Performance and leading time of the established models and their capability in predicting future admissions*

WBE-based prediction models were established for all twelve targets (3 indicators \times 4 leading times) using the data obtained from June 2021-May 2022 (Fig. 1). The model performance was evaluated using correlation coefficients (R), mean absolute error (MAE), and normalized MAE (NMAE) between model predictions and targets. For all three types of hospitalization indicators (i.e. Hos_wn, Hos_cs, and Hos_ca), the established WBE-based model well described the pattern of data observed from June 2021-May 2022 with overall R values over 0.90 and NMAE within 0.30 (Table S2). When applying the established batch models for predicting the future hospitalization indicators in June 2022-January 2023, the model performance for Hos_wn was greatly better than Hos_cs, and Hos_ca (Table 1). The prediction accuracy achieved R of 0.81-0.82 and NMAE of 0.32-0.37 for predicting Hos_wn, but only R of 0.59-0.67 and NMAE of 0.53-0.76 for Hos_cs and R of 0.66-0.69 and NMAE of 0.51-0.65 for Hos_ca (Table 1). This indicates that WBE-based predictions are likely more capable of capturing the weekly new admissions rather than the census average or sum of inpatients in the week.

Line 263-264 in *The necessity of periodical updates of WBE-based models*

the NMAE decreased from 0.32-0.37 in the batch models to 0.28-0.29 in the progressive learning models

Line 319-322 in *Transferability of the progressively updated WBE-based models*

From June 2022 to January 2023, the progressively learning models reasonably predicted the Hos_wn in these 60 counties in the next 1-4 weeks after the wastewater sampling, with an average MAE of 7-8 patients/100 k population and an average NMAE of 0.43-0.48.

Line 329- 333 in *Transferability of the progressively updated WBE-based models*

We further included the data of these 60 different counties from June 2022-January 2023 into the progressively learning models with the same update frequency (4 weeks). With the data of new counties included, the MAE of the prediction for these 60 counties reduced to 4-5 patients/100 k population with an average NMAE of 0.31-0.35 for the next 1-3 weeks, and MAE of 6 patients/100 k population and NMAE of 0.45 for Hos4w.

Table S2 for model establishment has been provided in **the supplementary information**

Table S2. Model performance during model establishment stage June 2021-May 2022

Indicators	Model	Hos1w			Hos2w			Hos3w			Hos4w		
		R	MAE	NMAE	R	MAE	NMAE	R	MAE	NMAE	R	MAE	NMAE
Hos_wn	WBE	0.91	2.87	0.22	0.90	2.77	0.22	0.92	3.15	0.24	0.90	3.48	0.27
	Record	0.86	3.18	0.25	0.87	4.85	0.38	0.83	5.40	0.42	0.80	5.57	0.43
	Case	0.86	3.18	0.25	0.87	4.39	0.34	0.85	4.88	0.38	0.81	5.37	0.41
Hos_cs	WBE	0.89	34.81	0.30	0.89	32.23	0.29	0.96	31.18	0.28	0.93	24.87	0.30
	Record	0.93	28.38	0.25	0.89	35.42	0.32	0.86	39.12	0.35	0.94	27.33	0.34
	Case	0.89	31.39	0.28	0.88	36.26	0.34	0.86	35.57	0.32	0.95	23.72	0.31
Hos_ca	WBE	0.95	3.97	0.26	0.97	3.43	0.22	0.95	3.54	0.23	0.94	3.76	0.24
	Record	0.96	3.49	0.23	0.91	5.03	0.33	0.88	5.84	0.38	0.86	6.24	0.40
	Case	0.91	2.87	0.22	0.90	2.77	0.22	0.92	3.15	0.24	0.90	3.48	0.27

Table 1 for batch model performance in predicting future targets in June 2022-January 2023 are provided in **the main text**.

Table 1. Performance WBE-based, case-based and record-based batch models predicting the future targets in June 2022-January 2023

Indicators	Model	Hos1w			Hos2w			Hos3w			Hos4w		
		R	MAE	NMAE	R	MAE	NMAE	R	MAE	NMAE	R	MAE	NMAE
Hos_wn	WBE	0.82	3.65	0.35	0.81	3.84	0.37	0.82	3.59	0.34	0.82	3.30	0.32
	Record	0.78	3.90	0.38	0.70	4.05	0.39	0.65	4.20	0.40	0.56	4.63	0.45
	Case	0.51	4.25	0.41	0.41	4.23	0.40	0.40	4.46	0.42	0.44	4.28	0.41
Hos_cs	WBE	0.60	61.74	0.76	0.59	58.37	0.72	0.67	46.61	0.57	0.62	42.71	0.53
	Record	0.78	25.78	0.32	0.69	32.82	0.40	0.61	35.68	0.43	0.47	38.25	0.47
	Case	0.56	34.39	0.43	0.80	33.71	0.42	0.56	34.99	0.43	0.63	34.11	0.42
Hos_ca	WBE	0.69	7.26	0.65	0.68	6.84	0.61	0.67	6.22	0.55	0.66	5.77	0.51
	Record	0.87	3.71	0.34	0.56	4.74	0.42	0.55	5.46	0.48	0.54	5.11	0.45
	Case	0.69	4.49	0.40	0.65	4.61	0.41	0.53	7.58	0.67	0.62	4.63	0.41

6. From the abstract, I am left wondering how well wastewater forecasts compare to forecasts using clinical surveillance data. Clinical surveillance is “free” to local health departments (cost passed onto individual health insurance) compared to wastewater surveillance which cannot pass this cost onto individual health insurance. And so if the goal is to make a low-cost prediction for hospitalizations, why not use clinical surveillance? How does wastewater compare to test positivity and incidence in terms of accuracy of predicting hospitalizations?

Regarding testing costs, we concur with the reviewer that the majority of testing is typically covered by health insurance in the USA. Nevertheless, we would like to note that there are still millions of individuals in the USA who lack health insurance. The number of uninsured populations was 27.2 M in 2021 and 28.3 M in 2020³⁰. Most of the uninsured population were socioeconomic, racial, or ethnic minorities³⁰, who tend to have a lower testing ratio but higher infections and mortality rates^{31,32}. Moreover, clinical testing is influenced by an individual's motivation to get tested, which can be affected by factors such as symptom, stigma, cost, and accessibility^{32,33}. Thus, clinical testing often captures part of the ‘true’ infection. The Centre for Disease Control and Prevention (CDC) also estimated that only 1 in 4.3 (95% UI 3.7–5.0) of total COVID-19 infections were reported through clinical testing³³.

In terms of **how accurate the WBE-based predictions are, in comparison to the current approaches (cases-based prediction and record-based predictions)**, we have established the county-level case-based and hospitalization-record-based (‘record-based’, hereafter) prediction models. The overall flow is shown as Fig. 1 below.

Fig. 6. Flow chart of the paper methodology, process, and structure

We used the data from 99 counties in 40 states of the USA during June 2021-May 2022 (12 months) for model establishment. For each type of prediction (i.e., WBE-based, case-based, and record-based), 12 random forest models were established (3 indicators×4 leading time=12 models). Overall, a total of 36 models (12 models for each type of prediction) were established in this stage.

In all three type of predictions (i.e. WBE-based models, case-based, record-based models), 13 common explanatory factors were used, including COVID-19 Community Vulnerability Index (CCVI, 8 indexes); county-level vaccination coverage (Vaccine_1st and Vaccine_2nd, %); population size of the county; and weather (air temperature T_a , °C, and precipitation, mm). In addition to these 13 common factors, the weekly new COVID-19 cases (cases/100k population) and test positivity (positive tests/total tests) were used for case-based predictions, C_{RNA} and wastewater temperature (T_w , °C) were used for WBE-based predictions, and hospitalization records for each indicator (i.e., Hos_wn, Hos_cs, Hos_ca) in the week of wastewater sampling were used for record-based prediction. Specifically, for record-based prediction, this means, for example, the new hospital admission in the week i was used for predicting the new hospital admission in the week $i+1$, $i+2$, $i+3$, and $i+4$. The algorithm, methods, and procedure for the model establishment were the same as in the previous manuscript.

After the model establishments, the above 36 models were used for **Model evaluation**. In this stage, these 36 models were used to predict the future hospitalization indicators (i.e. Hos_wn, Hos_cs, Hos_ca) from June 2022 to January 2023 ('future' to the model) under 4 leading times (1, 2, 3, 4 weeks) (Fig. 1). We evaluated the performance of each model using correlation coefficient (R), mean absolute error (MAE, Eq. 1), and normalized mean absolute error (NMAE, Eq. 2) for target selection and prediction type comparison (Fig. 1).

$$MAE = \frac{\sum_{i=1}^n (|y_i - \hat{y}_i|)}{n} \quad (1)$$

$$NMAE = \frac{\sum_{i=1}^n (|y_i - \hat{y}_i|)}{\sum_{i=1}^n (y_i)} \quad (2)$$

where y_i is the i^{th} observation of y and \hat{y}_i the predicted y_i value from the model. The n is the total number of data points.

The prediction accuracy in the model evaluation stage was compared to address *how accurate the WBE-based predictions are, in comparison to the current approaches (cases-based prediction and record-based predictions)*.

In the model establishment stage, the WBE-based model established well described the pattern of data observed from June 2021-May 2022 for all three types of hospitalization indicators (i.e. Hos_wn, Hos_cs, and Hos_ca), with overall R values over 0.90 and NMAE within 0.30 (Table S2). Both case-based models (R=0.81-0.97, NMAE=0.25-0.41) and record-based models (R=0.80-0.96, NMAE= 0.23-0.43) showed comparable or slightly worse performance than WBE-based predictions (R=0.90-0.97, NMAE=0.22-0.30) in describing the patterns in the data for model establishment (Table S2).

When applying the established batch models for predicting the future hospitalization indicators from June 2022 to January 2023, the WBE-based prediction for Hos_wn (R=0.81-0.82, NMAE=0.32-0.37) outperformed that of Hos_cs (R=0.59-0.67, NMAE=0.53-0.76) and Hos_ca (R=0.66-0.69, NMAE=0.51-0.65) regardless of the leading times. This indicates that WBE-based predictions are likely more capable of capturing the weekly new admissions in the following weeks rather than the total or daily average number of inpatients. The case-based or record-based models showed slightly better prediction for Hos_wn than Hos_cs and Hos_ca. The NMAE values achieved from our county-level case-based (0.40-0.42) and record-based (0.38-0.45) models for Hos_wn were comparable to previous case-base or record-based (or ensembled) prediction for daily new admissions at the state or national level in the USA (NMAE=0.35-0.45, leading time of 2-3 weeks)^{34, 35}. However, our WBE-based models outperformed the case-based or record-based models from both our study and previous studies with lower NMAE (0.32-0.37) and longer leading time (1-4 weeks).

The suboptimal performance of case-based predictions may be attributed to the potential bias of clinical testing, where only part of the infections in the community can be captured^{32, 33}. For record-based prediction, the inherent lag between the infection and hospitalization might also affect the prediction accuracy, especially for rapid changes in the infection status³⁵. In contrast, WBE unbiasedly captures the infection status among the population at the early stage of the infection^{6, 7, 8, 9}.

Detailed changes for the stages and relevant results in the manuscript (i.e. model establishment, model evaluation) have been listed in the response to the 5th comment from the Reviewer 2. Specifically, regarding the clinical surveillance, the following lines have been added to the manuscript.

Line 48-56 in Introduction:

To date, the prediction of hospitalization admissions due to COVID-19 majorly relies on

confirmed COVID-19 cases or historical records of daily or weekly COVID-19-induced admissions at the state or national level ^{36,37}. However, with the end of the COVID-19 public health emergency in many countries, changes in test availability, behavior, and reporting strategies reduced the certainty of COVID-19 infection numbers, especially for asymptomatic infections ³⁸. In addition, clinical testing may only capture a portion of the true infections in the community due to factors such as insurance coverage, individual willingness to be tested, and socioeconomic status in the area ^{32,33}. In clinical settings, it is common that some patients have been admitted to hospitals before obtaining positive COVID-19 tests ³⁶.

Line 194-209 in Results:

To facilitate comparison, additional prediction models were established using random forest algorithms based on new COVID-19 cases and test positivity (referred to as case-based predictions) and the relevant records for each hospitalization indicator (referred to as record-based predictions) at the county level. For model establishments, both case-based models (R=0.81-0.97, NMAE=0.25-0.41) and record-based models (R=0.80-0.96, NMAE=0.23-0.43) showed comparable or slightly worse performance than WBE-based predictions (R=0.90-0.97, NMAE=0.22-0.30) in describing the patterns in the data for all three targets (Table S2). When being applied to predict the future targets in June 2022-January 2023, both case-based or record-based models showed slightly better prediction for Hos_wn than Hos_cs and Hos_ca (Table 1). The NMAE values of our county-level case-based (0.40-0.42) and record-based (0.38-0.45) models for Hos_wn were comparable to previous case-base or record-based (or ensembled) prediction for daily new admissions at the state or national level in the USA (NMAE=0.35-0.45, leading time of 2-3 weeks) ^{34,35}. Nonetheless, our WBE-based models showed superior performance compared to case-based or record-based models for Hos_wn prediction, including those from previous studies, with lower NMAE (0.32-0.37) and longer leading time (1-4 weeks).

Line 390-399 in Discussion:

More importantly, our WBE-based predictions (NMAE=0.32-0.37) outperformed the record-based or case-based models in terms of the accuracy and leading time, for county-level predictions (our study, NMAE=0.38-0.45, leading time up to 4 weeks) and state/national-level predictions (previous studies, NMAE= 0.35-0.45, leading time of 2-3 weeks) ^{34,35}. The suboptimal performance of case-based predictions may be attributed to the potential bias of clinical testing, where only part of the infections in the community can be captured ^{32,33}. For

record-based prediction, the inherent lag between the infection and hospitalization might also affect the prediction accuracy, especially for rapid changes in the infection status³⁵. In contrast, WBE unbiasedly captures the infection status among the population at the early stage of the infection^{6, 7, 8, 9}.

7. Lines 54-57 don't follow too well. The changing in test availability, behavior, and reporting is what is driving the poorer clinical surveillance data. Not the change in contact tracing, mask mandates, and vaccine promotion.

We agree with this and have changed the line accordingly.

Line 50-53:

However, with the end of the COVID-19 public health emergency in many countries, changes in test availability, behavior, and reporting strategies reduced the certainty of COVID-19 infection numbers, especially for asymptomatic infections³⁸.

8. Line 68. None of these studies has compared the cost of wastewater surveillance to clinical surveillance. It is an added cost to public health budgets, whereas from a public health budget perspective clinical surveillance is free. Lines 71-74 are more accurate, as compared to a random community sample the cost is lower.

In our previous manuscript, the cited study Weidhaas, Aanderud³⁹ compared the economic value of wastewater surveillance with clinical testing (nasopharyngeal swabs). We agree with the reviewer that clinical testing is covered by health insurance for most of the population, however, we would like to mention the potential bias from the clinical surveillance (as discussed in the response to the 6th comment from reviewer 2).

To avoid confusion, we have changed the line as below.

Line 62-64:

Wastewater-based epidemiology (WBE) is considered an efficient approach for COVID-19 case surveillance, providing unbiased infection estimations at the community level with limited cost (0.7-1% of the population-wide testing)^{40, 41, 42, 43}

9. Lines 74-75. A number of studies have already shown that levels of SARS-CoV-2 in wastewater correlates from hospitalizations, some as early as 2020. The authors need to do a better literature review, cite these studies, and then show how their study builds upon them. Some examples from a very quick and cursory google search: Nattino et al. in JAMA, Zhan et al. in ACS EST Water, Galani et al. in Sci Total Environment.

Indeed, there are a few precedents reporting the association between SARS-CoV-2 RNA in wastewater (or primary sludge) with hospitalizations, and using such data to predict hospitalization records^{22, 17, 18, 19, 23}. Nattino, Castiglioni²², Zhan, Babler²³ observed the association/correlation between SARS-CoV-2 viral loads in wastewater and hospitalizations in a city (or county) although no prediction model was established. Kaplan, Wang¹⁷, Galani, Aalizadeh¹⁸, Peccia, Zulli¹⁹ created WBE-based epidemiological models for predicting new hospitalization with a leading time of 2-8 days. However, all these studies were limited to the area served by a couple of wastewater treatment plants under short-term monitoring (3-6 months). A recent study revealed the predictive potential for state-level hospitalization occupancy (census hospitalizations) with a leading time of 8-18 days in Austria²⁰. However, population demographics (such as race/ethnicity, vaccination, chronic conditions, etc.) that have been clinically observed impacting the COVID-19 symptom severity^{10, 11, 12, 13, 14, 15} were not considered in all these precedent prediction models. This limits their temporal and geographic scope, and thus making it uncertain whether they (both the model and the prediction targets) could be extrapolated to other areas. To reflect the discussion here, we have added the following lines to the introduction.

Line 66-80:

Few studies have reported the association between C_{RNA} in wastewater (or primary sludge) with hospitalizations^{22,23} and endeavored to create surveillance models for forecasting hospital admissions with various leading times ranging from 1 to 8 days^{17, 18, 19}. Nevertheless, these observations and models were developed using data from only a few localities for a short period (a couple of months). A recent study revealed the predictive potential for state-level hospitalization occupancy (census hospitalizations) with a leading time of 8-18 days in Austria²⁰. However, population demographics (such as race/ethnicity, vaccination, chronic conditions, etc.) that have been clinically observed impacting the COVID-19 symptom severity^{10, 11, 12, 13, 14, 15} were not considered in all these precedent prediction models. This limits their temporal and geographic scope, thus making it uncertain whether they (both the model and the

hospitalization indicators predicted) could be generalized to other areas. Considering that hospitals/healthcare facilities often allocate their resources and workers on a weekly basis for upcoming patients ²¹, a large-scale (temporal and geological) prediction system for hospitalizations at the county level on a weekly basis would be more informative for local healthcare facilities, which unfortunately is lacking.

10. The authors are using the Covid-19 community vulnerability index, and cite a pre-print from 2021 that has not been published as of 2023. The CDC created the social vulnerability index which does correlate with COVID-19 measures. How is the CCVI different from the CDC's SVI? I recommend the authors use the SVI as the CCVI is not peer-reviewed.

The COVID-19 Community Vulnerability Index (CCVI) was established by Surgo Foundation (a nonprofit organization) for better management and policy-making in the USA. The CCVI is adapted from Social Vulnerability Index (SVI) from the Centers for Disease Control and Prevention (CDC) in the USA with modifications regarding COVID-19-related risk factors (such as high-risk population and environment) ^{11,24}. At the county level, CCVI considers 40 measures from census data, covering 7 themes including i) socioeconomic status; ii) minority status and language, iii) housing type, transportation, household composition, and disability (“household and transportation” hereafter), iv) epidemiological factors, v) healthcare system, vi) high-risk environment and vii) population density, with an overall VI summarizing these 7 aspects ^{11,24}. The first 3 themes were adapted from SVI, while CCVI adds the last 4 themes considering their associations with COVID-19 outcomes.

Although the pre-print has not been published in peer-reviewed journals (which might be caused by many other reasons), the CCVI indexes are utilized by CDC for COVID-19-related response ²⁵ (publication from COVID-19 Response Team, CDC) and widely used in other studies (all peer-reviewed) for evaluating the epidemiological impacts/responses under COVID-19 ^{44,45,46,47}. Compared with SVI, CCVI has more COVID-19-specific modifications and showed better associations with COVID-19 outcomes (case, mortality) ^{14,48}. Thus, we chose to use CCVI to reflect the population demographic to ensure that the model/approach could be easily adapted to most regions based on their existing management systems, thereby promoting the transferability of the established approach. To reflect this, we have added the following lines to the manuscript.

Line 84-86:

The county-level population demographics were incorporated from COVID-19 Community Vulnerability Index (CCVI) ^{11, 24}, which is in use by the Centers for Disease Control and Prevention (CDC), for easy-adaption and transfer in different regions.

Line 488-505:

County-level population-related and weather data

For better management and policy-making, COVID-19 Community Vulnerability Index (CCVI) was established by Surgo Foundation and used by CDC for COVID-19-related response in the USA ²⁵. The CCVI is adapted from Social Vulnerability Index (SVI) from CDC with modifications regarding COVID-19-related risk factors (such as high-risk population and environment) ^{11, 24}. The CCVI is also widely used for evaluating the epidemiological impacts/responses under COVID-19 ^{44, 46}. At the county level, CCVI considers 40 measures from census data, covering 7 themes including i) socioeconomic status; ii) minority status and language, iii) housing type, transportation, household composition, and disability (“household and transportation” hereafter), iv) epidemiological factors, v) healthcare system, vi) high-risk environment, and vii) population density, with an overall VI summarizing these 7 themes ^{11, 24}. The CCVI overall score as well as the 7 theme indices range from 0 to 1, with 1 representing the most vulnerable area and 0 representing the least vulnerable area ^{11, 24}. The CCVI indexes of the overall score and 7 themes were obtained from the publicly available website (<https://precisionforcovid.org/ccvi>). We chose to use CCVI to reflect the population demographic rather than incorporating multiple measures from population census data to ensure that the model/approach could be easily adapted to most regions based on their existing management systems, thereby promoting the transferability of the established approach.

11. Line 90. Did the authors collect any data? The wastewater data was publicly available – what about the other measures the authors used? It looks like the authors conducted secondary data analyses of publicly available data, a very important distinction.

We conducted secondary data analyses of publicly available data. To clarify this, we have revised the manuscript as below.

Line 630:

We conducted secondary data analyses of publicly available data with data source listed below.

Line 463-465:

Wastewater surveillance data was obtained from the Biobot Nationwide Wastewater Mentoring Network (biobot.io/data), the largest publicly available dataset on SARS-CoV-2 RNA concentrations in wastewater.

Line 482-483:

The data for Hos_wn, Hos_cs, and Hos_ca was retrieved from HealthData.gov.

Line 500-502:

The CCVI indexes of the overall score and 7 themes were obtained from the publicly available website (<https://precisionforcovid.org/ccvi>).

12. Line 494, 500. Should read data were retrieved or obtained, not collected. (As is done in line 504). Wu et al. should be cited in the introduction.

We agree with this. Relevant changes have been made as shown in response to the comment above (11th comment from reviewer 2). *Duvallet and Wu et al.* for describing the analytical methods used in Biobot data (wastewater surveillance data) has been cited in the introduction.

Line 64-66:

Many studies have successfully quantified and correlated SARS-CoV-2 concentrations (C_{RNA}) in wastewater to COVID-19 cases^{27, 40, 41, 42}.

13. Line 512 – the authors state the CCVI was developed by the US CDC. Do they mean the SVI? It looks like it was actually developed by the Surgo Foundation and includes both proprietary and non-proprietary data. Digging into the Melvin et al. article from 2020 it looks like SVI plus human movement. The SVI would be more relevant for the time period the authors are analyzing (once human movement was less disrupted).

Please refer to the 10th comment from Reviewer 2 above, where relevant discussions regarding CCVI and SVI, along with changes in the manuscript have been made.

14. Lines 526-529. Dilution from precipitation would only matter for combined sewers. Do the authors not have access to wastewater treatment plant flow data?

To preserve the anonymity of the participating utilities and to improve their representativeness, the wastewater data (obtained from Biobot) was aggregated based on county and sample amount, providing one SARS-CoV-2 RNA concentration (C_{RNA}) per week for each county. For each sampling location, if there were more than one sample in a week, the concentrations of samples within each week were aggregated using an unweighted average. For each county, in a certain week, the concentrations obtained from each sampling location within the county were aggregated using a weighted average. The weight for a sampling location is relevant to the sewershed population, or 300,000, whichever is smaller. When a sampling location serves multiple counties, the location is associated with the single county that the wastewater operator has provided as the plant's primary service area. Thus, the location and the sewer type for relevant sampling facilities were not provided.

It is worth noting that the C_{RNA} (SARS-CoV-2 RNA concentration in wastewater) we used in the study (obtained from Biobot) was normalized to pepper mild mottle virus (a fecal indicator) to minimize any potential noise caused by the dilution, population size, and wastewater flow²⁷. Through the normalization, the population contributing to the wastewater sample is considered as a sub-group of the area of interest (county in our study) to reflect its infection status. The precipitation information in our study was only summarized and included into the model considering its potential impact. However, due to the randomness of random forest models, the inclusion of additional (or unnecessary) information generally does not affect the model performance^{49, 50}. The limited contribution of precipitation is also observed based on the results in our study. Regardless of the leading time, C_{RNA} was found to be the most important factor for predicting Hos_wn , contributing to a significant increase in MSE (50-67%, $p=0.010$) (Fig. 4a). While, precipitation showed negligible contribution to the model prediction (1-2%, $p=0.3-0.9$). This further confirms that the normalized C_{RNA} is a reliable predictor of hospitalizations and that variations in precipitation do not have a significant impact on the predictive power of the model.

Fig.4: Importance and contribution of the explanatory factors to the model predictions.

a: The importance of explanatory factors was ranked by the increase in %MSE (percent change in mean square error when the explanatory factor is permuted). A higher increase in %MSE corresponds to higher importance. The significance of the explanatory factors was marked as *, **, and *** representing a p value of ≥ 0.01 and < 0.05 , ≥ 0.001 and < 0.01 and < 0.001 , respectively.

To clarify this, the following lines have been added to the manuscript.

Line 228-231 in Results:

While precipitation showed negligible contribution to the model prediction (1-2%, $p=0.3-0.9$), which is likely due to that C_{RNA} being normalized to pepper mild mottle virus (a fecal indicator) to minimize any potential dilution-related variations²⁷.

Line 469-475 in Methods:

The concentration of SARS-CoV-2 RNA detected in each wastewater sample was normalized to pepper mild mottle virus (a fecal indicator) to minimize any potential noise caused by the dilution, population size, and wastewater flow²⁷. The normalized SARS-CoV-2 RNA concentration was further aggregated based on county and sample amount to preserve the

anonymity of participating utilities, providing one SARS-CoV-2 RNA concentration (C_{RNA}) per week for each county (details provided in Supplementary Text 1).

Supplementary Text 1:

To preserve the anonymity of participating utilities and to improve their representativeness, data was aggregated based on county and sample amount²⁷, as detailed below.

For each sampling location, if there is more than one sample in a week, the concentrations of samples within each week were aggregated using an unweighted average.

For each county, in a certain week, the concentrations obtained from each sampling location within the county were aggregated using a weighted average. The weight for a sampling location is relevant to the sewershed population, or 300,000, whichever is smaller. When a sampling location serves multiple counties, the location is associated with the single county that the wastewater operator has provided as the plant's primary service area.

15. The authors have not indicated how they handled the temporal or spatial scales of wastewater. Wastewater data would have varying temporalities with some sites providing weekly data and other sites providing more frequent data. The methods suggest a weekly temporal scale of data analysis using 7, 14, 21, and 28-day measures of hospitalizations. Hospitalizations would be on a daily scale. How have the authors matched those temporal scales? Also, on the spatial scale – how have the authors matched up the county data to the wastewater data? Numerous counties in the Biobot data have multiple wastewater treatment plants. Including them all with a direct match to hospitalization data would artificially inflate the dataset. The authors have also not considered wastewater surveillance coverage either in their analysis nor in their discussion.

To address this comment, we sub-divided this into three comments.

- a) how they handled varying temporalities of wastewater to match the Hospitalizations.

As mentioned in the response to the 14th comment from Reviewer 2 (the comment above), the wastewater concentrations were aggregated at the county level on a weekly basis, providing one SARS-CoV-2 RNA concentration (C_{RNA}) per week for each county. For hospitalization data, we utilized the data from HealthData.gov, where facility-level data for hospital utilization in each county was reported on a weekly basis. The facility-level data were further aggregated

at the county level. This means, both the wastewater data and hospitalization data are at the county level on a weekly basis. The clarification in the manuscript for wastewater data is provided in the response to the 14th comment from Reviewer 2. For hospitalization data, we have added the following lines in the manuscript.

Line 479-485:

Three indicators for hospitalization numbers were used including: 1) weekly new admission (Hos_wn), 2) the total number of patients who stayed in an inpatient bed during the week (census inpatient sum, Hos_cs), and 3) the daily average number of patients who stayed in an inpatient bed in the week (census inpatient average, Hos_ca). The data for Hos_wn, Hos_cs, and Hos_ca was retrieved from HealthData.gov. Briefly, facility-level data for hospital utilization in each county was reported on a weekly basis. The facility-level values for each indicator were further aggregated on a county basis.

b) how have the authors matched up the county data to the wastewater data?

As mentioned in the response to the 14th comment from Reviewer 2 (the comment above), the wastewater concentrations from multiple facilities within the same county under the same week were aggregated based on the facility and population. Please refer to the response there for detailed changes.

c) wastewater surveillance coverage

Indeed, the coverage of wastewater surveillance has always been a challenge for wastewater-based epidemiology. The actual population contributing to a certain wastewater sample may not be equal to the population served by the wastewater treatment plant, or the population in the relevant census district or health district^{28,29}. Thus, population-based normalization using endogenous biomarkers is commonly used in WBE to avoid the noise introduced by population size. In such a way, the SARS-CoV-2 RNA (or other WBE surveillance target) detected in wastewater can be normalized into the capital load, allowing inter-city or inter-county comparisons^{23,28}. In our study, the SARS-CoV-2 RNA wastewater was normalized to pepper mild mottle virus (a fecal indicator) to minimize any potential noise caused by the dilution, population size, and wastewater flow²⁷. Through the normalization, the population contributing to the wastewater sample is considered as a sub-group of the area of interest (county in our study) to reflect its infection status. Such normalization has been demonstrated

to greatly improve the correlation between the SARS-CoV-2 RNA concentrations and outbreak indicators (i.e. cases, hospitalizations, mortality)^{23, 27}.

Another challenge for wastewater surveillance is the presence of tourists or commuters within the studied area (county in our study). It is impossible to distinguish whether part of the SARS-CoV-2 RNA in wastewater had stemmed from a visitor(s) passing through or from the county itself. However, such uncertainty caused by population mobility is also unavoidable in other prediction methods (i.e., case-based, record-based)^{51, 52}. Recent studies have applied mobility surveillance data (such as cell phone mobility data) to improve the prediction accuracy^{35, 53}. Since such data was not available at the county level during the study period, it was not incorporated into our models. However, it is strongly recommended for future studies when the data becomes available. To reflect the discussion here, we have included the following lines in the manuscript.

Line 448-455:

Additionally, although normalization techniques that use endogenous population biomarkers can reduce the potential noise caused by the population size captured by the wastewater sample^{23, 28, 29}, the uncertainty caused by population mobility cannot be avoided in WBE-based predictions, as well as case-based or record-based predictions^{29, 51, 52}. Recently, researchers have employed mobility surveillance data, such as cell phone mobility data, to enhance prediction accuracy^{35, 53}. Although this information is not included in our models due to its unavailability at the county level during the study period, it is highly recommended for future studies when the data becomes accessible.

16. For the out of model predictions, did the authors randomly select data or did they randomly select treatment plants or time periods? I would prefer the authors to randomly select treatment plants or time periods rather than simply randomly selecting data. I see this now in lines 333, but would the authors state so in the methods?

For out of model predictions ('transferability' in our study), the established models were tested in 60 counties that were not included in the model establishment (see details provided in the response to the 5th comment from Reviewer 2, and Fig. 1a as shown in the response to 6th comment from Reviewer 2). Their data in June 2022- January 2023 was used for performance evaluation. To clarify this, we have revised the manuscript as below.

Line 612-613 in Methods section:

The transferability of progressive learning models established in the section above was tested in another 60 counties from 30 states in the USA from June 2022 to January 2023

17. From Figure 2 the correlations between wastewater levels and hospital measures are either 1 or near 1, which to me suggest some type of serial autocorrelation going on – either temporal or spatial. How have the authors accounted for this autocorrelation? Could the artificial duplication of data also be contributing to these hard to believe perfect correlations?

In our previous version, the correlation between the wastewater levels and hospital measures ranged from 0.63-0.68, which is a moderately strong correlation but not close to 1. In the new version, as suggested by reviewers, we changed the leading time on a weekly basis to avoid the overlap in our previous version (i.e. 0-7 d, 0-14 d...0-28 d in the previous version) and include more hospitalization indicators for comparison (detailed discussion provided in the response to 5th comment from Reviewer 2). So, in the new version, we have three hospitalization indicators (i.e. Hos_wn, Hos_cs, and Hos_ca). For each hospitalization indicator, we also have four leading times, the upcoming week (Hos1w), the second (Hos2w), third (Hos3w), and fourth weeks (Hos4w). The correlation between C_{RNA} and these 12 targets (three indicators \times 4 leading times) ranged from 0.46-0.56 (Fig. 3a). It is important to note that correlation analysis aimed to determine which factors should be included in the modeling process, rather than to compare or discuss the correlations between different variables or targets.

Fig. 7a: Correlation between explanatory factors and hospitalization records

a. Spearman's correlation between all the explanatory factors and hospital admission records. The color and circle size indicate the strength of the correlation (bigger circle=stronger correlation; blue color=positive correlation and red color=negative correlation). The significance of the correlation is marked as *, **, and *** representing a p value of ≥ 0.01 and < 0.05 , ≥ 0.001 and < 0.01 and < 0.001 , respectively.

For random forest models, autocorrelation is not typically considered an issue^{49,50}. As a non-parametric machine learning approach, random forest models work by constructing many decision trees on different subsamples of the data (random data points with random explanatory variables), and each tree is trained independently without any knowledge of the other trees. Two approaches are incorporated to ensure the randomness and diversity of the decision trees:

i) bootstrapping the training data so that each tree grows with a different sub-sample; ii) selecting features randomly to generate different subsets of explanatory variables for splitting nodes in a tree⁵⁴. Therefore, the correlations between observations in the data do not affect the individual trees or the final model.

To further demonstrate this, we also generated autocorrelation functions (ACF) plot of the residuals (errors) from the batch and progressively learning models for Hos_wn. The ACF plot checks whether the residuals (errors) are merely white noise with no significant serial correlation and not dependent on an adjacent observation⁵⁵. The ACF starts at a lag of 0, which is the correlation of the residual with itself and therefore results in a correlation of 1. From the lag of 1, if the residuals are from an independent white noise sequence, the sample autocorrelations should lie between the confidence interval (blue dashed lines in the figure). In both the batch model and progressively learning models, the ACF of residuals was within the confidence interval, demonstrating that the residual (prediction accuracy) of the model has no autocorrelation. To reflect the discussion here, we have added the following lines to the manuscript and included the ACF plot in the supplementary information as Figure S7.

Figure S7

Figure S7. Autocorrelation functions (ACF) plot for the residuals from the batch model (a) and progressively learning model (b) for Hos_wn.

Line 267-271:

For each leading time, the peaks of the error distribution were closer to 0 in the progressively learning models than in the batch models (Fig. 5b and 5c). The autocorrelation functions (ACF) ⁵⁵ confirmed that the residuals (errors) are merely white noise with no significant serial correlation and are not dependent on an adjacent observation (Fig. S7).

Line 525-533:

Random forest is a non-parametric machine learning approach to modeling the relationship between the potential explanatory factors (input variables) and the target ^{49, 50}. Random forest algorithm relies on establishing a group of individual decision trees to optimize model fit. Two approaches are incorporated to ensure the randomness and diversity of the decision trees: i) bootstrapping the training data so that each tree grows with a different sub-sample; ii) selecting features randomly to generate different subsets of explanatory variables for splitting nodes in a tree ⁵⁴. The correlations between observations in the data generally do not affect the individual trees or the final model. Thus, autocorrelation is not typically considered an issue for random forest models ^{49, 50}.

18. From Figure 4 it looks like the authors broke out their CCVI – this was not stated in the methods.

The CCVI has seven themes as detailed in the response to the 10th comment from Reviewer 2. Please refer to the discussions and changes in the manuscript as detailed there.

19. Wastewater had the greatest explanatory power for hospitalizations with a 7-day lead time. Why then do the authors select 14- and 28-day lead times to highlight? Considering the biology the authors highlight lines 386-393, the seven-day lead time makes sense. (Fecal shedding about when someone would test positive).

The goal of our study is to provide guidance to local health facilities for allocating resources and workers to combat COVID-19. Indeed, the C_{RNA} of a certain wastewater sample reflects the newly infected patients due to their shedding, however, it potentially also reflects the future COVID-19 patients due to the close contact with the current patients. Sputum and feces have been identified as the major shedding sources of SARS-CoV-2 RNA in wastewater, with the shedding load peaked (10^2 - 10^3 higher than other times) in the first couple of days before, to a week after, the symptom onset ^{6, 7, 8, 9}. Recent meta-analyses revealed that COVID-19 patients

remain contagious for around 12 days and the median time between symptom onset and hospitalization was 7 days (IQR: 5-10 days) ^{13, 56, 57}. Thus, depending on the symptom severity, part of the current infections is likely admitted to a hospital in the next 14 days, and part of the future infections is admitted in the 14-26 days after wastewater sampling. This is also reflected in the contributions of the explanatory factors in our models.

C_{RNA} was found as the most crucial explanatory factor, followed by population-health-related information (i.e. Vaccine_2nd, Vaccine_1st, and CCVI index in epidemiology factors) and COVID-19 transmission-related information (i.e. CCVI in population density and household and transportation). While population-health-related information showed comparable importance regardless of leading times, COVID-19 transmission-related information became increasingly important for longer leading times. As the leading times increased, there was a decrease in the significance of C_{RNA} in predicting Hos_wn, going from 66-67% for Hos1w-Hos3w to 51% for Hos4w. Meanwhile, the importance of CCVI in household and transportation increased from 10-11% for Hos1w-Hos2w to 14-17% for Hos3w-Hos4w, and population density showed an increase from 15-16% for Hos1w-Hos3w to 19% for Hos4w (Fig. 4a). These two factors (i.e. CCVI in population density and household and transportation) reflect the proximity to and interaction with other people and exposure to diseases, which directly relates to the COVID-19 transmission (impacting the number of future cases) ^{10, 11, 12}. While the vaccination status and CCVI in epidemiological factors consider high-risk populations and immunity status in the population, which are critical for both current and future COVID-19-infected patients. This supports the leading time we observed and the potential rationale behind that. To reflect the discussion here, we have revised the following lines in the manuscript.

Line 220-226:

As the leading times increased, there was a decrease in the significance of C_{RNA} in predicting Hos_wn, going from 66-67% for Hos1w-Hos3w to 51% for Hos4w. Meanwhile, the importance of CCVI in household and transportation increased from 10-11% for Hos1w-Hos2w to 14-17% for Hos3w-Hos4w, and population density showed an increase from 15-16% for Hos1w-Hos3w to 19% for Hos4w (Fig. 4a). This suggests that COVID-19 transmission-related information is more critical for predicting Hos_wn in later weeks.

Line 342-380:

The early warning capability of WBE for predicting the weekly new hospital admission in the healthcare system is likely related to viral RNA shedding from COVID-19 patients to sewers and the transmission of COVID-19 within the population. Sputum and feces have been identified as the major shedding sources of SARS-CoV-2 RNA in wastewater, with the shedding load peaked (10^2 - 10^3 higher than other times) in the first couple of days before, to a week after, the symptom onset^{6, 7, 8, 9}. Thus, the changes of C_{RNA} in wastewater samples are more sensitive to the variations in the numbers of COVID-19 infections at their early infection stages. Furthermore, recent meta-analyses revealed that COVID-19 patients remain contagious for around 12 days and the median time between symptom onset and hospitalization was 7 days (IQR: 5-10 days)^{13, 56, 57}. Thus, the C_{RNA} of a certain wastewater sample likely 1) directly reflects the newly infected patients, and 2) indirectly reflects the future COVID-19 patients in the following 12 days due to the close contact with the current patients. Depending on the severity of the symptoms, part of these newly and future infections are likely admitted to hospitals in the next 14 days, and 14-26 days after wastewater sampling, respectively. This is consistent with the 1-4 weeks of leading time in WBE-based predictions for weekly new admissions in our study and also reflected by the contributions of explanatory factors.

C_{RNA} was found as the most crucial explanatory factor, followed by population-health-related information (i.e. Vaccine_2nd, Vaccine_1st, and CCVI index in epidemiology factors) and COVID-19 transmission-related information (i.e. CCVI in population density and household and transportation). While population-health information showed comparable importance regardless of leading times, COVID-19 transmission-related information became increasingly important for longer leading times. Under the same C_{RNA} (infection status), a higher CCVI index in population density or household and transportation increased Hos_wn, especially when they are over 0.5. These two factors reflect the proximity to and interaction with other people and exposure to diseases, which directly relates to the transmission probability (impacting the number of future cases)^{10, 11, 12}. Under the same infection status, a higher Vaccine_2nd, higher Vaccine_1st, or lower CCVI in epidemiological factors reduced the Hos_wn, particularly under Vaccine_2nd >60% or CCVI in epidemiological < 0.5. This is consistent with the clinical observations of over 1 million patients, where a single dose and two doses of any vaccine (i.e. Pfizer-BioNTech, Oxford-AstraZeneca, Moderna that commonly used in the USA) were associated with a 35% and 67% reduction in the risk of hospitalization, respectively⁵⁸. The CCVI in epidemiological factors considers high-risk populations for

COVID-19 such as elderly adults and individuals with underlying health conditions (e.g. respiratory or heart conditions) that have been shown to be associated with more severe COVID-19 symptoms in clinical observations ^{10, 11, 12}. This supports our observations that health-related information was critical for predicting COVID-19-induced hospitalizations under all four leading times, while transmission-related information was more important for models with longer leading times.

20. Figure 1 looks okay. The authors might consider log-transforming the measures as in 1C they are so skewed.

Thanks for the suggestion. We have updated the Fig. 1c as Fig.2c in the new manuscript with log transformation.

Fig. 8: Geological location, COVID-19 Community Vulnerability Index (CCVI), and average weekly new COVID-19-induced hospitalizations in each month in the 99 counties involved.

c. The average weekly new hospitalization admission numbers in each month from these 99 counties. The data before June 2022 (12 months) were used for model establishments while data after June 2022 (8 months) were used for model evaluation.

21. Figure 2 looks good

Thanks for the positive feedback.

22. Table 1. I've never seen an R-squared of 0.98 when modeling hospitalizations – there is just too much random chance. And all of the outcomes are above 0.9. It raises a bit of an eyebrow – it would be wise to dig into the modeling and data to make sure these are real. I have a hard time reconciling that result with Figures 3, 4a, and 5.

Table 1 in our previous manuscript was the evaluation for the model performance over the data used for model establishment (including training, validation, and test data sets). Such a table was used to check whether the model accurately captured the trend over the whole dataset that was used for model establishment. A high R-square is required and pretty common in most machine learning models over the data used for the model establishment^{14, 59}. To make the paper flow clearer, we have further changed the manuscript into the model establishment stage, and model evaluation stage (as detailed in the response to the 5th comment from Reviewer 2). To avoid confusion, the performance table for models in the model establishment stage has been moved to the SI, and the performance table for models in the model evaluation stage is provided in the main text (as the new Table 1, shown in the response to the 5th comment from Reviewer 2).

The figure 3 was also the prediction results from the model establishment stage, which has been removed in the revised manuscript to avoid confusion. Instead, the prediction results in the model evaluation stage are provided as Fig. 5 in the revised manuscript. The figure 4a was the contributions of the explanatory factors, we have changed the figure to a circular bar plot to clarify the content (as new Fig. 4a, provided in the response to the 14th comment from Reviewer 2). Detailed changes are listed below.

Line 176-186:

For all three types of hospitalization indicators (i.e. Hos_wn, Hos_cs, and Hos_ca), the established WBE-based model well described the pattern of data observed from June 2021-

May 2022 with overall R values over 0.90 and NMAE within 0.30 (Table S2). When applying the established batch models for predicting the future hospitalization indicators in June 2022-January 2023, the model performance for Hos_wn was greatly better than Hos_cs, and Hos_ca (Table 1). The prediction accuracy achieved R of 0.81-0.82 and NMAE of 0.32-0.37 for predicting Hos_wn, but only R of 0.59-0.67 and NMAE of 0.53-0.76 for Hos_cs and R of 0.66-0.69 and NMAE of 0.51-0.65 for Hos_ca (Table 1). This indicates that WBE-based predictions are likely more capable of capturing the weekly new admissions rather than the census average or sum of inpatients in the week.

Line 194-206:

To facilitate comparison, additional prediction models were established using random forest algorithms based on weekly new COVID-19 cases and test positivity (referred to as case-based predictions) and the relevant weekly records for each hospitalization indicator (referred to as record-based predictions) at the county level. For model establishments, both case-based models (R=0.81-0.97, NMAE=0.25-0.41) and record-based models (R=0.80-0.96, NMAE=0.23-0.43) showed comparable or slightly worse performance than WBE-based predictions (R=0.90-0.97, NMAE=0.22-0.30) in describing the patterns in the data for all three targets (Table S2, Fig. S4). When being applied to predict the future targets in June 2022-January 2023, both case-based or record-based models showed slightly better prediction for Hos_wn than Hos_cs and Hos_ca (Table 1). The NMAE values of our county-level case-based (0.40-0.42) and record-based (0.38-0.45) models for Hos_wn were comparable to previous case-base or record-based (or ensembled) prediction for daily new admissions at the state or national level in the USA (NMAE=0.35-0.45, leading time of 2-3 weeks)^{34,35}.

Fig. 9. Comparison between actual admission records and the prediction results from batch models and progressive learning models for data in June 2022- January 2023.

a. The prediction results from the batch model (in blue) and progressive learning model (in orange) and the actual admission records (in black) for Hos_wn in eight representative counties. b. The prediction results from the batch model (in blue) and progressive learning model (in orange) versus the actual admission records for Hos_wn. c. The error distribution between prediction results and actual admission records for the batch model (in blue) and progressive learning model (in orange) for Hos_wn.

Reviewer #3 (Remarks to the Author):

Summary

In “Wastewater-based epidemiology predicts COVID-19-induced hospital and ICU admission numbers in over 100 USA counties,” the authors explore the predictability of hospital and ICU rates across the United States using a wide variety of potential predictors including wastewater-based estimates of disease. They explore relationships between the predictor and response variables, construct random forest prediction models, test the fits of these models, and validate their models with additional data held back from the fitting procedure. Overall I find the paper to be well-written and topical, as wastewater based epidemiology is a nascent field and there remain many questions regarding its utility as a surveillance system. However, I have major concerns about the data and methodology as presented in the current work, which I describe in detail below.

Thanks for the positive feedback. We have thoroughly revised our manuscript based on all the suggestions from Reviewers and the Editor. For better cross-reference purpose, we have labelled the comments in numbers.

Major comments

1. Based on the title and text, the study purports to predict county-level COVID-19 hospital and ICU admission counts. I investigated the “University of Minnesota COVID-19 Hospitalization Tracking Project” cited by the authors (note that the reference number cited does not link to the data source), and I am fairly certain that it provides access to hospitalization and ICU census counts, not admission counts. Hospital census describes the raw number of people hospitalized or in the ICU with COVID-19 on a given day, where admissions describe the new patients arriving on to a hospital on a given day. The authors should strongly check which data they are using - from my eye, the values in figure 3 seem to be too large and too smooth to be daily admissions. If I’m correct, then the authors should change the text to reflect this difference. More importantly, this dramatically impacts the interpretation of the results. There are much higher correlations between day-to-day hospital and ICU census counts overall, as the counts from today impact the counts from tomorrow. This causes issues with model fitting, parameter estimation, and forecasting as discussed in (<https://royalsocietypublishing.org/doi/10.1098/rspb.2015.0347>). For these reason, all

forecasting efforts for COVID-19 have focused on hospital admission counts (e.g. <https://github.com/reichlab/covid19-forecast-hub> and <https://www.pnas.org/doi/10.1073/pnas.2111870119>). Otherwise they've focused on daily new reported cases or deaths. The authors could theoretically switch the analysis to use WBE for predicting case counts instead of the hospitalization/ICU census counts.

In our previous version, we used the database established by the University of Minnesota (UM) COVID-19 Hospitalization Tracking Project ⁶⁰ to predict the weekly census hospitalization and ICU admission in the next 0-7, 0-14, 0-21, and 0-28 day. The data source and prediction targets were on a weekly basis instead of a daily basis.

After carefully evaluating reviewers' suggestions regarding the prediction targets (census or new admissions) and leading time (potential overlap between our previous targets), we have changed our paper structure into three stages: model establishment, model evaluation, and model transferability (Fig. 1). The model establishment and model evaluation stages address the question-*which hospitalization indicator can be predicted by WBE-based prediction?*

Fig. 10: Flow chart of the paper methodology, process and structure

In model establishment stage, we used three hospitalization indicators: 1) weekly new admission (Hos_wn), 2) total number of patients stayed in an inpatient bed during the week (census inpatient sum, Hos_cs), and 3) daily average number of patients stayed in an inpatient

bed during the week (census inpatient average, Hos_ca). These hospitalization indicators were chosen because:

Previously, few WBE studies have reported the correlations between these hospitalization indicators with SARS-CoV-2 RNA in wastewater (C_{RNA}) with various leading times ranging from 1 to 18 days in different regions during different stages of the outbreak^{17,18,19,20}. However, previous studies were limited to the area served by a couple of wastewater treatment plants under short-term monitoring (3-6 months). The regional variations in lagging time and limited temporal and geographic scope of previous reports, make it uncertain whether they (the model, lagging time, and hospitalization indicators) could be extrapolated to other areas.

Four leading times were used over the course of the upcoming week (Hos1w), as well as the second (Hos2w), third (Hos3w), and fourth weeks (Hos4w). This means, we are predicting the hospitalization indicators in the week $i+1$, $i+2$, $i+3$, and $i+4$ week based on the information obtained at week i . This was changed as suggested by reviewers, considering the potential overlaps in our previous version (e.g. 0-7 day, 0-14, ..., 0-28 days in our previous version).

The data source for hospitalization records was changed to the official weekly data record from USA government (HealthData.gov, COVID-19 Reported Patient Impact and Hospital Capacity by Facility | HealthData.gov). This is because the previous database (established by UM) focused only on the census inpatient average and stopped updating in August 2022. The prediction for ICU admissions was removed due to the data unavailability.

These three hospitalization indicators were chosen on a weekly basis (rather than daily) because:

- The shedding dynamic of SARS-CoV-2 RNA to sewers.
The infection status among the population is reflected by the SARS-CoV-2 RNA concentration (C_{RNA}) in wastewater. The viral RNA shedding from infected individual to sewers (through feces, sputum and other bodily fluids). The shedding load peaks in the first couple of days before, to a week after, the symptom onset^{6,7,8,9}. This means, the contribution of each infected individual can be detected in a certain period (more than a week) rather than a day. This is quite different to the incident case or hospitalization record (or ensembled predictions such as COVID-19 forecastHub), where the case record, or hospitalization record for a certain individual only reported once in a certain period (until re-infection or re-admission).

- The potential impact of variant diversity and population demographics in different counties.

Whether and when an infected individual is admitted to a hospital is also dependent on other factors, such as the viral variant, race/ethnicity, vaccination, and chronic conditions^{10, 11, 12, 13, 14, 15}. SARS-CoV-2 variants evolve over time and exhibit distinct regional patterns across the nation¹⁶. Moreover, population demographics and vaccination coverage vary across counties and over time. These factors can potentially affect the time between viral shedding and hospitalization. This is evident by the various leading times reported by previous WBE studies for hospitalization, which vary from 1-5 days to 8-18 days in different regions, during different stages of the outbreak^{17, 18, 19, 20}. Thus, for a large-scale prediction (both geographical and temporal), predicting the hospitalization numbers within a certain period (weekly), rather than daily would be more feasible.

- The weekly resource allocation and staff arrangement in most healthcare systems. Most hospitals allocate their resources and staff on a weekly basis for incoming patients²¹.
- The turn-over time of wastewater samples.

Considering the logistics of wastewater sampling and laboratory analysis, the turn-over time for wastewater samples can vary from a few hours to several days, depending on the capacity of the testing facility. Therefore, for WBE-based prediction, a lower sampling frequency (weekly rather than daily) and longer prediction period (up to 4 weeks, rather than 14 days) would be more practical and feasible, in terms of analytical cost, time, and effort.

Specifically, **which hospitalization indicator can be predicted by wastewater-based prediction?**

In our study, in the model establishment stage, data from June 2021 to May 2022 was used for establishing WBE-based models for each hospitalization indicator under each leading time (3 indicators \times 4 leading times). In the model establishment stage, the established WBE-based model well described the pattern of data observed from June 2021-May 2022 for all three types of hospitalization indicators (i.e. Hos_wn, Hos_cs, and Hos_ca), with overall R values over 0.90 and NMAE within 0.30. When applying the established batch models for predicting the hospitalization indicators from June 2022 to January 2023 ('future' to the model), the

prediction performance of Hos_wn (R=0.81-0.82, NMAE=0.32-0.37) outperformed that of Hos_cs (R=0.59-0.67, NMAE=0.53-0.76) and Hos_ca (R=0.66-0.69, NMAE=0.51-0.65) regardless of the leading times.

The better performance of WBE-based predictions for Hos_wn is likely related to the viral shedding pattern. Sputum and feces have been identified as the major shedding sources of SARS-CoV-2 RNA in wastewater, with the shedding load peaked (10^2 - 10^3 higher than other times) in the first couple of days before, to a week after, the symptom onset^{6, 7, 8, 9}. Thus, the changes of C_{RNA} in wastewater samples are more sensitive to the variations in the numbers of COVID-19 patients at their early infection stages. Furthermore, recent meta-analyses revealed that COVID-19 patients remain contagious for around 12 days and the median time between symptom onset and hospitalization was 7 days (IQR: 5-10 days)^{13, 56, 57}. Thus, the C_{RNA} of a certain wastewater sample likely 1) directly reflects the newly infected patients, and 2) indirectly reflects the future COVID-19 patients in the following 12 days due to the close contact with the current patients. Depending on the severity of the symptoms, part of these newly infected and future patients is likely admitted to hospitals in the next 14 days, and 14-26 days after wastewater sampling, respectively.

In contrast, the census admission numbers for a particular week encompass both new admissions and continuing admissions from previous weeks. Hospital stays can vary significantly from a few days to as long as 41 days, depending on factors such as prescribed treatments, chronic conditions (like diabetes and hypertension), nutritional risks (such as body mass index and cognitive impairment), etc.^{61, 62, 63, 64}. Accurately capturing and integrating these variables at the population level into WBE-based predictions (or any other existing approaches) may be challenging. This is also commonly observed in case-based or record-based models (the existing approaches), where better prediction accuracy was achieved for new admissions rather than census inpatient numbers^{35, 65}.

Based on the results presented above, WBE-based models for Hos_wn under all four leading times were further used to investigate whether periodic updates based on the most up-to-date information were necessary (**Model evaluation stage** in Fig. 1). Additionally, the transferability of WBE-based models for Hos_wn prediction to the other 60 counties in the USA was explored (**Transferability stage** in Fig. 1). Further details on the periodic update and model transferability can be found in details in the response to the comment in later sections to avoid repetition here.

To reflect the discussion here, we have revised the manuscript as below.

Line 66-100

Few studies have reported the association between C_{RNA} in wastewater (or primary sludge) with hospitalizations^{22,23} and endeavored to create surveillance models for forecasting hospital admissions with various leading times ranging from 1 to 8 days^{17, 18, 19}. Nevertheless, these observations and models were developed using data from only a few localities for a short period (a couple of months). A recent study revealed the predictive potential for state-level hospitalization occupancy (census hospitalizations) with a leading time of 8-18 days in Austria²⁰. However, population demographics (such as race/ethnicity, vaccination, chronic conditions, etc.) that have been clinically observed impacting the COVID-19 symptom severity^{10, 11, 12, 13, 14, 15} were not considered in all these precedent prediction models. This limits their temporal and geographic scope, thus making it uncertain whether they (both the model and the hospitalization indicators predicted) could be generalized to other areas. Considering that hospitals/healthcare facilities often allocate their resources and workers on a weekly basis for upcoming patients²¹, a large-scale (temporal and geographical) prediction system for hospitalizations at the county level on a weekly basis would be more informative for local healthcare facilities, which unfortunately is lacking.

In this work, we collected county-level weekly WBE data from the recent 20 months (June 2021 to January 2023) covering 159 counties from 45 states in the USA (Fig. 1) with their corresponding county-level hospital admission records, vaccination records, and weather conditions. The county-level population demographics were incorporated from COVID-19 Community Vulnerability Index (CCVI)^{11, 24}, which is in use by the Centers for Disease Control and Prevention (CDC), for easy-adaption and transfer in different regions. Random forest models were established using these factors to predict the county-level hospitalization indicators over the course of the upcoming week, as well as the second, third, and fourth weeks after the wastewater sampling to address the following: (1) The feasibility of using WBE for predicting hospital admission numbers in healthcare systems: which hospitalization indicator can be predicted by WBE-based prediction and how accurate are the predictions in comparison to the current approaches (cases-based prediction and record-based predictions)? (2) The contribution of CCVI indexes, vaccination, and weather factors for the prediction: how are they affecting the WBE-based prediction? (3) For real applications, is a periodic update of the model necessary? (4) The transferability of the models to other counties and states: how accurate is

the model prediction for other counties and how to improve the accuracy? (Fig.1). Our results would help improve the preparedness of healthcare systems and vulnerable counties in the USA in coping with the COVID-19 pandemic or endemic.

Fig. 11: Flow chart of the paper methodology, process, and structure.

Line 478-487 in Methods:

County-level hospitalization data in the USA

Three indicators for hospitalization numbers were used including: 1) weekly new admission (Hos_wn), 2) the total number of patients who stayed in an inpatient bed during the week (census inpatient sum, Hos_cs), and 3) the daily average number of patients who stayed in an inpatient bed in the week (census inpatient average, Hos_ca). The data for Hos_wn, Hos_cs, and Hos_ca was retrieved from HealthData.gov. Briefly, facility-level data for hospital utilization in each county was reported on a weekly basis. The facility-level values for each indicator were further aggregated on a county basis. Considering the preparation window, records for each indicator in the next 1-4 weeks of the wastewater sampling were summarized for each county and used in this study.

Line 536-539 in Methods:

For model establishment, data from June 2021 to May 2022 (3162 data points for each target, 12 months) were utilized to describe the patterns for each target through the random forest algorithm in R (ver 4.2.0, R Foundation for Statistical Computing, <http://www.R-project.org/>).

Line 593-599 in Methods:

Model evaluation and comparison

models established using the data from June 2021 to May 2022 were employed to forecast hospitalization indicators from June 2022 to January 2023 ('future' data to the model, 4616 data points for each model) using relevant explanatory factors. The prediction accuracy of the models was evaluated using MAE and NMAE to compare and select the types of prediction (i.e. WBE-based, case-based, and record-based), hospitalization indicators (i.e. Hos_wn, Hos_cs, Hos_ca) and leading times (i.e. 1-4 weeks).

Line 176-186:

For all three types of hospitalization indicators (i.e. Hos_wn, Hos_cs, and Hos_ca), the established WBE-based model well described the pattern of data observed from June 2021-May 2022 with overall R values over 0.90 and NMAE within 0.30 (Table S2). When applying the established batch models for predicting the future hospitalization indicators in June 2022-January 2023, the model performance for Hos_wn was greatly better than Hos_cs, and Hos_ca (Table 1). The prediction accuracy achieved R of 0.81-0.82 and NMAE of 0.32-0.37 for predicting Hos_wn, but only R of 0.59-0.67 and NMAE of 0.53-0.76 for Hos_cs and R of 0.66-0.69 and NMAE of 0.51-0.65 for Hos_ca (Table 1). This indicates that WBE-based predictions are likely more capable of capturing the weekly new admissions rather than the census average or sum of inpatients in the week.

Line 342-358 in Discussion:

The early warning capability of WBE for predicting the weekly new hospital admission in the healthcare system is likely related to viral RNA shedding from COVID-19 patients to sewers and the transmission of COVID-19 within the population. Sputum and feces have been identified as the major shedding sources of SARS-CoV-2 RNA in wastewater, with the shedding load peaked (10^2 - 10^3 higher than other times) in the first couple of days before, to a week after, the symptom onset^{6, 7, 8, 9}. Thus, the changes of C_{RNA} in wastewater samples are more sensitive to the variations in the numbers of COVID-19 infections at their early infection stages. Furthermore, recent meta-analyses revealed that COVID-19 patients remain contagious for around 12 days and the median time between symptom onset and hospitalization was 7 days (IQR: 5-10 days)^{13, 56, 57}. Thus, the C_{RNA} of a certain wastewater sample likely 1) directly reflects the newly infected patients, and 2) indirectly reflects the future COVID-19 patients in

the following 12 days due to the close contact with the current patients. Depending on the severity of the symptoms, part of these newly and future infections are likely admitted to hospitals in the next 14 days, and 14-26 days after wastewater sampling, respectively. This is consistent with the 1-4 weeks of leading time in WBE-based predictions for weekly new admissions in our study and also reflected by the contributions of explanatory factors.

Line 381-390:

The WBE-based predictions more accurately captured the `Hos_wn` compared to the daily census average or census sum patient numbers in the week. The census admission numbers for a particular week encompass both new admissions and continuing admissions from previous weeks. Hospital stays can vary significantly from a few days to as long as 41 days, depending on factors such as prescribed treatments, chronic conditions (like diabetes and hypertension), nutritional risks (such as body mass index and cognitive impairment), etc.^{61,62,63,64}. Accurately capturing and integrating these variables at the population level into WBE-based predictions (or any other existing approaches) may be challenging. This is also commonly observed in case-based or record-based models (the existing approaches), where better prediction accuracy was achieved for new admissions rather than census inpatient numbers^{35,65}.

Table S2 for model establishment has been provided in the supplementary information

Table S2. Model performance during model establishment stage June 2021-May 2022

Indicators	Model	Hos1w			Hos2w			Hos3w			Hos4w		
		R	MAE	NMAE	R	MAE	NMAE	R	MAE	NMAE	R	MAE	NMAE
Hos_wn	WBE	0.91	2.87	0.22	0.90	2.77	0.22	0.92	3.15	0.24	0.90	3.48	0.27
	Record	0.86	3.18	0.25	0.87	4.85	0.38	0.83	5.40	0.42	0.80	5.57	0.43
	Case	0.86	3.18	0.25	0.87	4.39	0.34	0.85	4.88	0.38	0.81	5.37	0.41
Hos_cs	WBE	0.89	34.81	0.30	0.89	32.23	0.29	0.96	31.18	0.28	0.93	24.87	0.30
	Record	0.93	28.38	0.25	0.89	35.42	0.32	0.86	39.12	0.35	0.94	27.33	0.34
	Case	0.89	31.39	0.28	0.88	36.26	0.34	0.86	35.57	0.32	0.95	23.72	0.31
Hos_ca	WBE	0.95	3.97	0.26	0.97	3.43	0.22	0.95	3.54	0.23	0.94	3.76	0.24
	Record	0.96	3.49	0.23	0.91	5.03	0.33	0.88	5.84	0.38	0.86	6.24	0.40
	Case	0.91	2.87	0.22	0.90	2.77	0.22	0.92	3.15	0.24	0.90	3.48	0.27

Table 1 for batch model performance in predicting future targets in June 2022-January 2023 are provided in the main text.

Table 2. Performance WBE-based, case-based and record-based batch models predicting the future targets in June 2022-January 2023

Indicators	Model	Hos1w			Hos2w			Hos3w			Hos4w		
		R	MAE	NMAE	R	MAE	NMAE	R	MAE	NMAE	R	MAE	NMAE
Hos_wn	WBE	0.82	3.65	0.35	0.81	3.84	0.37	0.82	3.59	0.34	0.82	3.30	0.32
	Record	0.78	3.90	0.38	0.70	4.05	0.39	0.65	4.20	0.40	0.56	4.63	0.45
	Case	0.51	4.25	0.41	0.41	4.23	0.40	0.40	4.46	0.42	0.44	4.28	0.41
Hos_cs	WBE	0.60	61.74	0.76	0.59	58.37	0.72	0.67	46.61	0.57	0.62	42.71	0.53
	Record	0.78	25.78	0.32	0.69	32.82	0.40	0.61	35.68	0.43	0.47	38.25	0.47
	Case	0.56	34.39	0.43	0.80	33.71	0.42	0.56	34.99	0.43	0.63	34.11	0.42
Hos_ca	WBE	0.69	7.26	0.65	0.68	6.84	0.61	0.67	6.22	0.55	0.66	5.77	0.51
	Record	0.87	3.71	0.34	0.56	4.74	0.42	0.55	5.46	0.48	0.54	5.11	0.45
	Case	0.69	4.49	0.40	0.65	4.61	0.41	0.53	7.58	0.67	0.62	4.63	0.41

2. The time periods for the analysis are confusing as written, hindering the ability to evaluate all of the results fully. It appears that the authors have split up the time period for training, testing, and validation based on the table, but they also appear to show in-sample fits for the whole time period (Figure 3). On top of that they also describe 5-fold cross validation for understanding the importance of explanatory factors. I think it would greatly enhance the clarity of the paper if the authors outline how all of these components fit together with one another alongside including the dates of analysis for each figure/table caption as well. For example, the predictions in Figure 5 for the batch model don't appear to perform that well, but it was my understanding that is the same prediction model described in the above figures showing strong predictive ability.

In the previous manuscript, figure 3 was utilized to depict the model's performance over the data in the model establishment stage, while figure 5 was for predicting "future" data (future to the model) for model evaluation. Therefore, the performance presented in these two figures was not identical, which might be confusing.

To address this issue and enhance the clarity of the manuscript, we have reorganized the paper flow into three stages, as discussed in response to the 1st from Reviewer 3. To avoid confusions, the results for model establishment (previous Fig 3) was removed from the manuscript. The main manuscript now only presents the results for predicting future data using the established model (new Fig. 5).

The detailed changed for the three stages are shown in the response to the 1st comment from the Reviewer 3. The new Fig. 5, along with the results illustrated are provided below.

Line 258-264:

The random forest models developed in the previous sections for predicting Hos_wn under different leading times were progressively updated every four weeks between June 2022 and January 2023, considering the healthcare system settings (Fig. 1). The performance of the models improved greatly through progressively learning compared to the batch model (Fig. 5b and 5c). The MAE reduced from 4 patients/100k population in the batch models to 3 patients/100k population in the progressively learning models, and the NMAE decreased from 0.32-0.37 in the batch models to 0.28-0.29 in the progressive learning models

Line 276-296:

Specifically, the prediction performance of the batch and progressively learning models were illustrated in eight representative counties (selected based on population size). Predictions from both batch models and progressively learning models reached good agreements with the actual admission records (Fig. 5a), regardless of the leading time. Compared with batch models, progressively learning models reduced the MAE by 10-70% for a certain county and showed better prediction capability towards the rapid changes in the trends (both sudden rise and drops) (Fig. 5a).

Fig. 12. Comparison between actual admission records and the prediction results from batch models and progressive learning models for data in June 2022- January 2023.

a. The prediction results from the batch model (in blue) and progressive learning model (in orange) and the actual admission records (in black) for Hos_wn in eight representative counties. b. The prediction results from the batch model (in blue) and progressive learning model (in orange) verse the actual admission records for Hos_wn. c. The error distribution between prediction results and actual admission records for the batch model (in blue) and progressive learning model (in orange) for Hos_wn.

3. The predictions shown in Figure 3 and in the table are remarkably good. However, it is difficult to fit these results into the findings of the larger forecasting field. Are the authors claiming that their model can make extremely accurate 4 week predictions? If so, this would be above and beyond what other teams have been able to do in the COVID-19 forecast hub, for example see: <https://forecasters.org/blog/2021/09/28/on-the-predictability-of-covid-19/>. Given this performance, it would be useful to understand how these predictions compare with alternative models such as the null model used in the forecast hub, alongside models that are built on other predictors. For example, are the hospitalization predictions equally as good if one only uses previous hospitalization data to make the predictions, or does one really need

WBE data? In general, the question is, why have the author developed such accurate forecasts, is it the data, the model, something else? Any explanation and comparison with the field would be helpful. Furthermore, It would be useful to include more of the predictor variables in the analysis of variable importance. Would WBE still be chosen as most important over the more traditional data streams (e.g. cases, hospitalizations, or ICU counts)?

We have subdivided this comment for better clarity.

- a) predictions shown in Figure 3 and in the table are remarkably good making it difficult to fit these results into the findings of the larger forecasting field.

The results in previous Figure 3 and Table 1 were from the model establishment stage, which describes the model performance over the data used for model establishment (including training, validation, and test data sets). Such a table and figure were used to check whether the model accurately captured the trend over the whole dataset that was used for model establishment. A very accurate performance is required and pretty common in most machine learning models over the data used for the model establishment ^{14, 59}.

To make the paper flow clearer, we have further changed the manuscript into the model establishment stage, and model evaluation stage (as detailed in the response to the 1st comment from Reviewer 3 and the response above). To avoid confusion, the performance table in the model establishment stage has been moved to SI (Table S2 as shown in 1st comment of Reviewer 3), and the performance table for models predicting the ‘future’ hospitalizations in the model evaluation stage is provided in the main text (as the new Table 1). The previous Figure 3 (for the model establishment stage) has been removed, instead, the performance of the models in predicting the ‘future’ targets is illustrated in Fig.5. The new Table 1, Fig 5 and relevant results have been provided in the response to the 1st and 2nd comment from Reviewer 3.

- b) How is the accuracy of WBE-based models in comparison to previous case-based or record-based models?

As shown in Fig1 (provided in the response to the 1st comment from Reviewer 3), we also established the county-level prediction models using cases and test positivity (case-based prediction) and hospitalization record (record-based prediction) for predicting three hospitalization indicators (Hos_wn, Hos_ca, Hos_cs) in four leading times over the course of

the upcoming week (Hos1w), as well as the second (Hos2w), third (Hos3w), and fourth weeks (Hos4w).

The reason for establishing such models is:

Previously, through the hospital admission records or cases (or ensembled), prediction models have been established to predict the daily (or weekly) new admissions in state-³⁵ or national-level (COVID-19 Forecast Hub)⁶⁶ in the USA. However, hospitalization patterns can differ significantly within the same state, due to variations in factors such as population demographics and healthcare resources^{5, 67}. There is a lack of county-level predictions for hospitalization indicators using case-based or hospitalization-record-based ('record-based' hereafter) predictions.

How these models were established and evaluated:

We used the data from 99 counties in 40 states of the USA during June 2021-May 2022 (12 months) for model establishment. For each type of prediction (i.e., WBE-based, case-based, and record-based), 12 random forest models were established (3 indicators×4 leading time=12 models). Overall, a total of 36 models (12 models for each type of prediction) were established in this stage.

In all three type of predictions (i.e. WBE-based models, case-based, record-based models), 13 common explanatory factors were used, including COVID-19 Community Vulnerability Index (CCVI, 8 indexes); county-level vaccination coverage (Vaccine_1st and Vaccine_2nd, %); population size of the county; and weather (air temperature T_a , °C, and precipitation, mm). In addition to these 13 common factors, the weekly new COVID-19 cases (cases/100k population) and test positivity (positive tests/total tests) were used for case-based predictions, C_{RNA} and wastewater temperature (T_w , °C) were used for WBE-based predictions, and hospitalization records for each indicator (i.e., Hos_wn, Hos_cs, Hos_ca) in the week of wastewater sampling were used for record-based prediction. Specifically, for record-based prediction, this means, for example, the new hospital admission in the week i was used for predicting the new hospital admission in the week $i+1$, $i+2$, $i+3$, and $i+4$. The algorithm, methods, and procedure for the model establishment were the same as in the previous manuscript.

After the model establishments, the above 36 models were used for **Model evaluation**. In this stage, these 36 models were used to predict the future hospitalization indicators (i.e. Hos_wn, Hos_cs, Hos_ca) from June 2022 to January 2023 ('future' to the model) under 4 leading times

(1, 2, 3, 4 weeks) (Fig. 1). We evaluated the performance of each model using correlation coefficient (R), mean absolute error, and normalized mean absolute error for target selection and prediction type comparison (Fig. 1).

How accurate are the predictions in comparison to the current approaches (cases-based prediction and record-based predictions)?

In the model establishments stage, both case-based models (R=0.81-0.97, NMAE=0.25-0.41) and record-based models (R=0.80-0.96, NMAE= 0.23-0.43) showed comparable or slightly worse performance than WBE-based predictions (R=0.90-0.97, NMAE=0.22-0.30) in describing the patterns in the data for all three targets (Table S2). When being applied to predict the future targets in June 2022-January 2023, both case-based or record-based models showed slightly better prediction for Hos_wn than Hos_cs and Hos_ca. The NMAE values achieved from our county-level case-based (0.40-0.42) and record-based (0.38-0.45) models for Hos_wn were comparable to previous case-base or record-based (or ensembled) prediction for daily new admissions at the state or national level in the USA (NMAE=0.35-0.45, leading time of 2-3 weeks)^{34,35}. However, our WBE-based models outperformed the case-based or record-based models for Hos_wn prediction from both our study and previous studies with lower NMAE (0.32-0.37) and longer leading time (1-4 weeks) (Table 1). Table 1 and Table S2 are provided in the response to the 1st comment to the Reviewer 3.

The suboptimal performance of case-based predictions may be attributed to the bias of clinical testing, where only part of the infection in the community can be captured^{32,33}. For record-based prediction, the inherent lags between the infection and hospitalization might also affect the prediction accuracy, especially for rapid changes in the infection status³⁵. In contrast, WBE unbiasedly captures the infection status among the population at the early stage of the infection.

It is worth noting the limitation of the WBE-based predictions in our study, considering the regional variations in the leading time^{17,18,19} and the turnover time for sample analysis (a couple days), the WBE-based models predict hospitalization on a weekly basis. Although this meets the weekly resource allocation and staff arrangement in most healthcare systems, for certain regions where a high-resolution (such as daily) prediction is required, the case-based or record-based prediction might be more suitable than WBE-based predictions.

To reflect the discussion here, we have added the following lines in the manuscript.

Line 48-61:

To date, the prediction of hospitalization admissions due to COVID-19 majorly relies on confirmed COVID-19 cases or historical records of daily or weekly COVID-19-induced admissions at the state or national level^{36,37}. However, with the end of the COVID-19 public health emergency in many countries, changes in test availability, behavior, and reporting strategies reduced the certainty of COVID-19 infection numbers, especially for asymptomatic infections. In addition, clinical testing may only capture a portion of the true infections in the community due to factors such as insurance coverage, individual willingness to be tested, and socioeconomic status in the area^{32,33}. In clinical settings, it is common that some patients have been admitted to hospitals before obtaining positive COVID-19 tests³⁶. Ensembled probabilistic forecasts for daily incident hospitalizations were also provided based on the forecast from multiple teams at state and national levels⁴. However, hospitalization rates and patterns can vary significantly at the county level due to differences in population demographics, healthcare resources, etc., even within the same state⁵. More granular insights for predicting hospitalization at county-level are more ideal for practical application.

Line 194-209:

To facilitate comparison, additional prediction models were established using random forest algorithms based on weekly new COVID-19 cases and test positivity (referred to as case-based predictions) and the relevant records for each hospitalization indicator (referred to as record-based predictions) at the county level. For model establishments, both case-based models (R=0.81-0.97, NMAE=0.25-0.41) and record-based models (R=0.80-0.96, NMAE= 0.23-0.43) showed comparable or slightly worse performance than WBE-based predictions (R=0.90-0.97, NMAE=0.22-0.30) in describing the patterns in the data for all three targets (Table S2, Fig. S4). When being applied to predict the future targets in June 2022-January 2023, both case-based or record-based models showed slightly better prediction for Hos_wn than Hos_cs and Hos_ca (Table 1). The NMAE values of our county-level case-based (0.40-0.42) and record-based (0.38-0.45) models for Hos_wn were comparable to previous case-base or record-based (or ensembled) prediction for daily new admissions at the state or national level in the USA (NMAE=0.35-0.45, leading time of 2-3 weeks)^{34,35}. Nonetheless, our WBE-based models showed superior performance compared to case-based or record-based models for Hos_wn prediction, including those from previous studies, with lower NMAE (0.32-0.37) and longer leading time (1-4 weeks).

Line 394-399:

The suboptimal performance of case-based predictions may be attributed to the potential bias of clinical testing, where only part of the infections in the community can be captured^{32,33}. For record-based prediction, the inherent lag between the infection and hospitalization might also affect the prediction accuracy, especially for rapid changes in the infection status³⁵. In contrast, WBE unbiasedly captures the infection status among the population at the early stage of the infection^{6,7,8,9}.

Line 455-460:

In addition, considering the regional variations in the leading time and the turnover time for sample analysis (up to several days), the WBE-based models predict hospitalizations on a weekly basis. Although this meets the weekly resource allocation and staff arrangement in most healthcare systems, for certain regions where a high-resolution (such as daily) prediction is required, the case-based or record-based prediction might be more suitable than WBE-based predictions.

- a) It would be useful to include more of the predictor variables in the analysis of variable importance. Would WBE still be chosen as most important over the more traditional data streams?

Thanks for the suggestions. We would like to clarify that the importance of explanation factors (input variables) is evaluated by permuting the value of each explanatory factor through 5-fold cross-validation for a certain random forest model⁶⁸. This means, for a certain set of data, when the value of an explanatory factor is permuted (rearranged) while the others remain the same, the increase in the mean squared error (MSE, %) of predictions is considered as the importance of the explanatory factor. This would be helpful to evaluate the contribution of the explanatory factors in capturing the trend of the data (used for model establishment) but cannot guarantee the prediction accuracy of the model for future data when a variable is not included. Thus, we established the case-based and record-based models as detailed in the response above, to compare the importance of WBE information and the case, or hospitalization-based information in the predictive capability for future hospitalizations.

To clarify this, we have added the following lines to the manuscript.

Line 572-574:

For a certain set of data, the importance score for each explanatory factor was determined as the percentage increase in mean square error (%MSE) observed when the value of an explanatory factor was permuted, compared to when no metrics were permuted.

Line 213-215:

The importance of explanatory factors for models established for Hos_wn prediction was evaluated by the increase in mean squared error (MSE, %) of predictions when the value of a certain explanatory factor was permuted⁶⁸

Minor comments:

4. For the statement: “Our study demonstrated the potential of using WBE as a cost-effective method to provide early warnings for healthcare systems.” The authors did not include any cost-effective analysis comparing predictor variables, so I would suggest removing that from the claim.

We agree with this and have changed the line accordingly.

Line 35-36:

Our study demonstrated the potential of using WBE as an effective method to provide early warnings for healthcare systems.

5. It would be extremely helpful to include some example time-series for all of the time-dependent predictor variables. I am surprised that WBE performs as well as case counts in predicting hospitalizations/ICU census, and it would be useful to see the time-series to better visualize the relationship.

We have illustrated time-serial data for the predictor variables in each county, including C_{RNA} , and weekly new cases in Fig. S2. The figure for Hos_wn is provided as Fig. 2c, while Hos_cs, Hos_ca are shown in Fig S2. Relevant changes have been made in the manuscript.

Line 124-131:

The Hos_wn, Hos_cs, and Hos_ca indicators had a range of 0-100 patients/100k population, 0-1220 patients/100k population, and 0-175 patients/100k population, respectively. The highest peaks were observed during August 2021 to February 2022 (Fig. 2c, Fig.S2). The C_{RNA}

of wastewater samples ranged from 0.4 to 9000 copies/mL (IQR:101.54 - 546.53 copies/mL) (Fig. S2). The weekly new COVID-19 cases ranged from 0- 4065 incidence/100k population (IQR: 48-271 incidence/100k population). The hospitalization indicators and C_{RNA} were skewed to higher ranges (Fig. S2), which is consistent with the inherent development of the outbreak.

Fig 2c in the main text:

Fig. 13: Geological location, COVID-19 Community Vulnerability Index (CCVI), and average weekly new COVID-19-induced hospitalizations in each month in the 99 counties involved.

c. The average weekly new hospitalization admission numbers in each month from these 99 counties. The data before June 2022 (12 months) were used for model establishments while data after June 2022 (8 months) were used for model evaluation.

Figure S2 in the supplementary information.

Figure S14. The weekly new COVID-19 cases (cases/100k population), C_{RNA} in wastewater samples, Hos_cs (total number of patients stayed in an impatient bed during the week), and Hos_ca (daily average number of patients stayed in an impatient bed during the week), in each county during the study period. The grey cells in the heatmap indicate missing values. The color gradient in each cell represents the monthly average of each indicator.

6. Fig 1C shows time-series data as points and a box-plot. It is difficult to glean any information from these plots and I would suggest that simple time-series might be more interpretable. Also, shouldn't the 7d, 14d, 21d, and 28d healthcare values all have the same appearance? A 14d value is just a 7d value lagged by 7 days.

We agree with this. As mentioned in the response to the first comment from Reviewer 3, in the previous version, the timespan for the prediction was 0-7 day, 0-14 day, 0-21 day, and 0-28 day. In the new version, we have changed the leading time to a weekly basis to avoid the overlap, as the following week (Hos1w), and the 2nd, 3rd, and 4th week after the wastewater sampling (detailed in the response to the 1st comment from Reviewer 3). Thus, Fig 1c has been updated as the average weekly new hospitalization admission numbers in each month in each county as shown in the response above (5th comment from Reviewer 3).

7. Fig 2 - I think percent positivity is more interpretable and common than the reverse positive metric. This would also help put the correlation for that metric positive than the negative one currently shown

We agree with this. As we added new hospitalization indicators as prediction targets, we have moved the correlation analysis for case-based and record-based models to the supplementary information (Text 3 in supplementary information). It should be noted here that the correlation analysis was majorly used to decide which factor shall be included in the model, rather than discussing the strength of the correlation.

Relevant changes were also made for the main text.

Line 543-545:

In addition to these 13 common factors, the weekly new COVID-19 cases (cases/100k population) and test positivity (positive tests/total tests) were used for case-based predictions

Line 548-549:

The correlation between the hospitalization indicators and the explanatory factors for case-based and record-based models were provided in the supplementary Text 3.

Supplementary Text 3 Correlation between prediction targets and explanatory factors in case-based and record-based predictions

For case-based predictions, the weekly new cases and the positive rate of the testing showed a moderately strong correlation ($R=0.42-0.64$, $p<0.001$) with the all three hospitalization indicators under all four leading times (Fig. S8). Under the same leading time, the weekly new cases and the positive rate showed comparable or slightly stronger correlation with Hos_cs ($R=0.45-0.64$) than Hos_ca ($0.44-0.63$) and Hos_wn ($0.41-0.62$). For each hospitalization indicators, the correlation between the hospitalization indicators and weekly new cases or the positive rate reduced along with the increase of leading time.

For record-based predations, the hospitalization records for each indicator (i.e., Hos_wn, Hos_cs, Hos_ca) in the week of wastewater sampling (Hos0w in the Fig. S8) positively correlated with the future values of these indicators in the next 1-4 weeks (Fig. S8). The correlation between Hos0w and Hos_ca ($R=0.53-0.75$) was stronger than that of Hos_cs ($R=0.46-0.71$) and Hos_wn ($0.37-0.64$) (Fig. S8). The correlation for each indicator also reduced along with the increase of the leading time, with the least correlation coefficient achieved at Hos4w.

Other explanatory factors, including population size, and factors associated with vaccination, CCVI, and the weather showed significant correlations ($|R|$ of $0.01-0.27$) with at least one of the targets (Fig. S8). Considering the randomness of random forest algorithm, all these explanatory factors were used for establishing case-based or record-based models.

Figure S8. Spearman's correlation between explanatory factors and hospitalization records in the data used for case-based and record-based models. The color and circle size indicate the strength of the correlation (bigger circle=stronger correlation; blue color=positive correlation and red color=negative correlation). The significance of the correlation is marked as *, **, and *** representing a p value of ≥ 0.01 and < 0.05 , ≥ 0.001 and < 0.01 and < 0.001 , respectively.

8. Fig 4 needs further explanation. Why do explanatory factors go to 100? What does explanatory factors mean? What is partial dependence and how is it defined?

Explanatory factors are input variables. As discussed in the response to the 3rd comment from Reviewer 3, the importance of explanatory factors (input variables) is evaluated by permuting the value of each explanatory factor for a certain random forest model. Thus, the sum of the increase in the mean squared error (MSE, %) due to permuting each explanatory factor does not necessarily equals to 100%. Relevant changes to the manuscript are detailed in the response to the 3rd comment from Reviewer 3.

Partial dependence depicts the marginal effect of one or two explanatory variables on the outputs while controlling for other explanatory variables ²⁶. Mathematically, the partial dependence function for regression is defined as (Eq. 3).

$$\widehat{f}_S(x_S) = E_{X_C}[\widehat{f}(x_S, X_C)] = \int \widehat{f}(x_S, X_C)dP(X_C) \quad (3)$$

The x_S are the features for which the partial dependence function should be plotted and X_C are the other features used in the machine learning model \widehat{f} . The mathematical expectation is denoted by E and probability by P . The partial function $\widehat{f}_S(x_S)$ shows the relationship between x_S feature and the predicted targets. The partial function $\widehat{f}_S(x_S)$ is estimated by calculating averages in the training data, also known as Monte Carlo method as Eq. 4:

$$\widehat{f}_S(x_S) = \frac{1}{N} \sum_{i=1}^N \widehat{f}(x_S, x_{iC}) \quad (4)$$

Where $\{X_{1C}, X_{2C}, \dots, X_{NC}\}$ are the values of other variables X_C in the dataset, N is the number of instances. The partial dependence method works by averaging the machine learning model output over the distribution of the features in set C, allowing the function to illustrate the relationship between the features in set S (of interest) and the predicted outcome. By averaging over the other features, we obtain a function that is dependent solely on the features in set S. In other words, partial dependence reveals the relationship between the targets (outputs) and the explanatory factors in x_S (explanatory factors that we are interested). To clarify this, we have added the following lines to the manuscript.

Fig.4 b and c

Fig. 15: Importance and contribution of the explanatory factors to the model predictions.

b-c: The two-factor partial dependence of Hos_wn at Hos2w (subfigure b) and Hos4w (subfigure c) on C_{RNA} and four significant explanatory factors used in the models. The horizontal axis represents the values of C_{RNA} , whereas the vertical axis represents the values of the other four explanatory factors (as shown in the title). The color gradients in the figure indicate the partial dependence of the predicted target concerning a specific x-value and y-value combination.

Line 574-592:

Partial dependence depicts the marginal effect of one or two explanatory variables (input variables) on the outputs while controlling for other explanatory variables²⁶. Mathematically, the partial dependence function for regression is defined as (Eq. 3).

$$\widehat{f}_S(x_S) = E_{X_C}[\widehat{f}(x_S, X_C)] = \int \widehat{f}(x_S, X_C) dP(X_C) \quad (3)$$

The x_S are the features for which the partial dependence function should be plotted and X_C are the other features used in the machine learning model \widehat{f} . The mathematical expectation is denoted by E and probability by P . The partial function $\widehat{f}_S(x_S)$ shows the relationship between x_S feature and the predicted targets. The partial function $\widehat{f}_S(x_S)$ is estimated by calculating averages in the training data, also known as Monte Carlo method as Eq. 4:

$$\widehat{f}_S(x_S) = \frac{1}{N} \sum_{i=1}^N \widehat{f}(x_S, x_{ic}) \quad (4)$$

Where $\{X_{1C}, X_{2C}, \dots, X_{NC}\}$ are the values of other variables X_C in the dataset, N is the number of instances. The partial dependence method works by averaging the machine learning model output over the distribution of the features in set C, allowing the function to illustrate the relationship between the features in set S (of interest) and the predicted outcome. By averaging over the other features, we obtain a function that is dependent solely on the features in set S. In other words, partial dependence reveals the relationship between the targets (outputs) and the explanatory factors in x_S (explanatory factors that we are interested).

9. It's not clear that Fig 6 is showing strong out-of-sample prediction performance. There is a clear bias towards positive residuals for hospitalizations and negative residuals for ICU. Also, the blue outlier county may be throwing off evaluation metrics for hospitalizations. For example, if you remove that county, then the relationship for hospitalizations looks pretty much like a horizontal line. In general I would suggest using a larger data set for this out-of-sample analysis.

We have extended the geographical (to 60 new counties) and temporal scope (to 8 months) of the out-of-sample analysis, which we called 'transferability' in the manuscript.

As mentioned in the 1st comment from the Reviewer 3, the progressively learning model for Hos_wn prediction in the next 1-4 weeks was used to test the model transferability. The performance of the models was evaluated in predicting another 60 counties from 30 states in the USA (details provided in Supplementary Table S4) from June 2022 to January 2023. Additionally, the study also investigated the impact of localized data updates on model transferability. Data in these 60 counties from June 2022 to January 2023 was progressively incorporated into the existing progressive learning model under the same update frequency. This means, at week i , the data in these 60 counties from June 2022 to week $i-1$, was incorporated into the dataset used for establishing the progressively learning model, providing the prediction till the next update (week $i+4$). Details for the methods, results and discussion are provided below.

Line 612-613 in *Transferability of progressive learning models*:

The transferability of progressive learning models established in the section above was tested in another 60 counties from 30 states in the USA from June 2022 to January 2023

Line 617-623 in *Transferability of progressive learning models*:

Additionally, the study investigated the impact of localized data updates on model transferability. Data in these 60 counties from June 2022 to January 2023 was progressively incorporated into the existing progressive learning model under the same update frequency. This means, at week i , the data in these 60 counties from June 2022 to week $i-1$, was incorporated into the dataset used for establishing the progressively learning model, providing the prediction till the next update (week $i+4$). Model predictions were compared with actual admission records and evaluated using MAE, and NMAE.

Line 313-337 in the Results section:

Transferability of the progressively updated WBE-based models

The progressive learning models (established in above sections) using the data from 99 counties were applied for predicting the Hos_wn in another 60 different counties from 30 states in the USA with a population size ranging from 0.2 M to 10.0 M (nearly 40M population in total, Table S4). These 60 counties were all unknown to the model (not included in the model establishment process), and 7 of them were from 5 new states (i.e. DC, GA, NM, ND, SD, Table S4). From June 2022 to January 2023, the progressively learning models reasonably predicted the Hos_wn in these 60 counties in the next 1-4 weeks after the wastewater sampling (Fig. 6b, 6c), with an average MAE of 7-8 patients/100 k population and an average NMAE of 0.43-0.48 (Fig. 6a). In six representative counties, although the progressively learning models captured the overall trends of the data, the models were insensitive to sudden changes in the patterns (drop or rise), especially for the counties from a new state to the model (the first three counties in Fig. 7) at longer leading times (Hos3w and Hos4w) (Fig. 7).

Fig. 16. The performance of progressively learning models with and without the data from 60 new counties in June 2022-January 2023.

a. The MAE of the progressively learning models with and without the data from new counties for predicting the Hos_wn in these 159 counties (99 original counties and 60 new counties) in June 2022-January 2023. The 60 new counties are labeled with yellow dot on the left. The color of each cell in the main heatmap indicates the MAE between the prediction and the actual admission record in each county. The box plot in the right presents the weekly new admissions (patients/ 100k population) in each county during June 2022-January 2023, while the top box plot summarizes the NMAE for the prediction in different leading times (Hos1w, Hos2w, Hos3w, and Hos4w) for the original 99 counties (in purple) and the 60 new counties (in orange). b. The prediction results from the progressively learning models with (on the right) and without (on the left) the data from new counties for predicting the Hos_wn in the original 99 counties (in purple) and the 60 new counties (in orange). c. The error distribution between prediction results and actual admission records from the progressively learning models with (on the right) and without (on the left) the data from new counties for predicting the Hos_wn in the original 99 counties (in purple) and the 60 new counties (in orange).

Fig. 17. The prediction results from the progressively learning model without (in purple) and with (in orange) the data from new counties for Hos_wn in six representative counties.

We further included the data of these 60 different counties from June 2022-January 2023 into the progressively learning models with the same update frequency (4 weeks). With the data of new counties included, the MAE of the prediction for these 60 counties reduced to 4-5 patients/100 k population with an average NMAE of 0.31-0.35 for the next 1-3 weeks, and MAE of 6 patients/100 k population and NMAE of 0.45 for Hos4w. The inclusion of data from new counties did not affect the prediction performance for the original 99 counties with comparable MAE at 3 patients/100 k population and NMAE of 0.27-0.28 for Hos1w, Hos2w and Hos3w, but slightly increased the MAE to 4 patients/100 k population (NMAE of 0.35) for Hos4w.

Line 407-429 in Discussions

The progressive learning models also showed reasonable transferability to other 60 counties from 30 states in the USA, with slightly higher NMAE of 0.43-0.48. After incorporating the data from new counties on a monthly basis into the progressively learning models, the updated model reached comparable prediction accuracy towards all 159 counties, with a NMAE of 0.31-0.35 for the next 1-3 weeks, and 0.45 for Hos4w. Thus, for future applications, the progressive learning model with the most recent datasets from relevant counties is highly recommended, and the methodology established in our study has a huge potential to be applied in other regions/counties.

The necessity of periodic updates of localized data from relevant counties is likely related to the variation and evolution of immunity and SARS-CoV-2 variants in different counties, as well as the nature of machine-learning approaches. As discussed in the above sections, vaccination coverage showed a significant contribution to predicting Hos_wn. However, the effect of vaccination on immune protection typically declines over time due to antibody neutralization⁶⁹. The effectiveness of Pfizer or Moderna vaccines decreased from around 65-70% to approximately 10%, 20 weeks after the second dose⁵⁸. Moreover, SARS-CoV-2 variants evolve over time and exhibit distinct regional patterns across the nation¹⁶. Reduced risks of progression to severe clinical outcomes (i.e. hospitalization) were observed with Omicron infections than with Delta infections⁷⁰. Even during the Omicron infections, the effectiveness of vaccines and the probability of hospitalization also varied against different Omicron subvariants^{70, 71, 72}. Thus, the number of hospitalizations under the same infection status may also depend on the remaining immunity from vaccinations and subvariants of

infections in each county over time. The progressively learning model provides the most up-to-date information, allowing the model to adjust its structure to accommodate new changes.

Line 430-442 in Discussions

There are several limitations in this study. The community's immunity is affected by several factors, such as booster shots' recipient coverage and the time interval between booster shots and the second dose of vaccination, as well as infection-induced immunity⁷³. Unfortunately, such information on booster shots was not available at the county level, and the effectiveness and duration of infection-induced immunity remain largely unknown^{69, 70}. Thus, such information was not included in our models. For future research, it is recommended to incorporate time-weighted vaccination and prior infections to evaluate community immunity to predict hospital admissions. Additionally, immune protection from vaccination or prior infections varies against different subvariants⁷⁰. Since reports on the proportion of infections from different variants/subvariants often delay due to the time required for clinical and wastewater analyses (which can take up to months depending on analytical capabilities), such information was not included in our study. However, it is encouraged for future investigations when timely information becomes available.

Reference

1. Hong S, Lynn HS. Accuracy of random-forest-based imputation of missing data in the presence of non-normality, non-linearity, and interaction. *BMC medical research methodology* **20**, 1-12 (2020).
2. Atkinson AC, Riani M, Corbellini A. The box-cox transformation: Review and extensions. (2021).
3. Asar Ö, İlk O, Dag O. Estimating Box-Cox power transformation parameter via goodness-of-fit tests. *Communications in Statistics-Simulation and Computation* **46**, 91-105 (2017).
4. Ray EL, *et al.* Ensemble Forecasts of Coronavirus Disease 2019 (COVID-19) in the U.S. *medRxiv*, 2020.2008.2019.20177493 (2020).
5. Mokhtari A, Mineo C, Kriseman J, Kremer P, Neal L, Larson J. A multi-method approach to modeling COVID-19 disease dynamics in the United States. *Scientific reports* **11**, 1-16 (2021).

6. Li X, *et al.* SARS-CoV-2 shedding sources in wastewater and implications for wastewater-based epidemiology. *Journal of Hazardous Materials* **432**, 128667 (2022).
7. Crank K, Chen W, Bivins A, Lowry S, Bibby K. Contribution of SARS-CoV-2 RNA shedding routes to RNA loads in wastewater. *Science of The Total Environment* **806**, 150376 (2022).
8. Miura F, Kitajima M, Omori R. Duration of SARS-CoV-2 viral shedding in faeces as a parameter for wastewater-based epidemiology: Re-analysis of patient data using a shedding dynamics model. *Science of the Total Environment* **769**, (2021).
9. Jones DL, *et al.* Shedding of SARS-CoV-2 in feces and urine and its potential role in person-to-person transmission and the environment-based spread of COVID-19. *Science of the Total Environment* **749**, (2020).
10. León TM. COVID-19 cases and hospitalizations by COVID-19 vaccination status and previous COVID-19 diagnosis—California and New York, May–November 2021. *MMWR Morbidity and Mortality Weekly Report* **71**, (2022).
11. Surgo-Foundation. The COVID-19 Community Vulnerability Index (CCVI). (ed Ventures S) (2020).
12. Krumel Jr T. The Meatpacking Industry in Rural America During the COVID-19 Pandemic. *US Department of Agriculture, Economic Research Service, Washington DC*, (2020).
13. Pouw N, *et al.* Clinical characteristics and outcomes of 952 hospitalized COVID-19 patients in The Netherlands: A retrospective cohort study. *PLoS One* **16**, e0248713 (2021).
14. Tiwari A, Dadhania AV, Ragunathrao VAB, Oliveira ERA. Using machine learning to develop a novel COVID-19 Vulnerability Index (C19VI). *Science of The Total Environment* **773**, 145650 (2021).
15. Nordström P, Ballin M, Nordström A. Risk of infection, hospitalisation, and death up to 9 months after a second dose of COVID-19 vaccine: a retrospective, total population cohort study in Sweden. *The Lancet* **399**, 814-823 (2022).
16. Hodcroft EB. CoVariants: SARS-CoV-2 Mutations and Variants of Interest. (ed covariants.org) (2021).

17. Kaplan EH, Wang D, Wang M, Malik AA, Zulli A, Peccia J. Aligning SARS-CoV-2 indicators via an epidemic model: application to hospital admissions and RNA detection in sewage sludge. *Health Care Management Science* **24**, 320-329 (2021).
18. Galani A, *et al.* SARS-CoV-2 wastewater surveillance data can predict hospitalizations and ICU admissions. *Science of The Total Environment* **804**, 150151 (2022).
19. Peccia J, *et al.* Measurement of SARS-CoV-2 RNA in wastewater tracks community infection dynamics. *Nature biotechnology* **38**, 1164-1167 (2020).
20. Schenk H, *et al.* Prediction of hospitalisations based on wastewater-based SARS-CoV-2 epidemiology. *Science of The Total Environment* **873**, 162149 (2023).
21. Rossman H, *et al.* Hospital load and increased COVID-19 related mortality in Israel. *Nature communications* **12**, 1904 (2021).
22. Nattino G, *et al.* Association between sars-cov-2 viral load in wastewater and reported cases, hospitalizations, and vaccinations in milan, march 2020 to november 2021. *Jama* **327**, 1922-1924 (2022).
23. Zhan Q, *et al.* Relationships between SARS-CoV-2 in wastewater and COVID-19 clinical cases and hospitalizations, with and without normalization against indicators of human waste. *Acs Es&T Water* **2**, 1992-2003 (2022).
24. Smittenaar P, *et al.* A COVID-19 community vulnerability index to drive precision policy in the US. *medRxiv*, (2021).
25. Wolkin A, *et al.* Comparison of national vulnerability indices used by the Centers for Disease Control and Prevention for the COVID-19 response. *Public Health Reports* **137**, 803-812 (2022).
26. Cheng L, Chen X, De Vos J, Lai X, Witlox F. Applying a random forest method approach to model travel mode choice behavior. *Travel Behaviour and Society* **14**, 1-10 (2019).
27. Duvallet C, *et al.* Nationwide Trends in COVID-19 Cases and SARS-CoV-2 RNA Wastewater Concentrations in the United States. *ACS ES&T Water*, (2022).
28. Chen C, Kostakis C, Gerber JP, Tschärke BJ, Irvine RJ, White JM. Towards finding a population biomarker for wastewater epidemiology studies. *Science of the Total Environment* **487**, 621-628 (2014).

29. Sims N, Kasprzyk-Hordern B. Future perspectives of wastewater-based epidemiology: Monitoring infectious disease spread and resistance to the community level. *Environment International* **139**, 105689 (2020).
30. Keisler-Starkey K, Bunch LN. Health insurance coverage in the United States: 2019. *Washington, DC: US Census Bureau*, (2020).
31. Mackey K, *et al.* Racial and ethnic disparities in COVID-19–related infections, hospitalizations, and deaths: a systematic review. *Annals of internal medicine* **174**, 362-373 (2021).
32. Li X, *et al.* Correlation between SARS-CoV-2 RNA concentration in wastewater and COVID-19 cases in community: A systematic review and meta-analysis. *Journal of Hazardous Materials* **441**, 129848 (2023).
33. Reese H, *et al.* Estimated incidence of coronavirus disease 2019 (COVID-19) illness and hospitalization—United States, February–September 2020. *Clinical Infectious Diseases* **72**, e1010-e1017 (2021).
34. Rosenfeld R, Tibshirani RJ. From the Cover: Beyond Cases and Deaths: The Benefits of Auxiliary Data Streams In Tracking the COVID-19 Pandemic: Epidemic tracking and forecasting: Lessons learned from a tumultuous year. *Proceedings of the National Academy of Sciences of the United States of America* **118**, (2021).
35. Fox SJ, *et al.* Real-time pandemic surveillance using hospital admissions and mobility data. *Proceedings of the National Academy of Sciences* **119**, e2111870119 (2022).
36. Deschepper M, Eeckloo K, Malfait S, Benoit D, Callens S, Vansteelandt S. Prediction of hospital bed capacity during the COVID– 19 pandemic. *BMC Health Services Research* **21**, 468 (2021).
37. Ferstad JO, *et al.* A model to forecast regional demand for COVID-19 related hospital beds. *MedRxiv*, (2020).
38. Reese H, *et al.* Estimated Incidence of Coronavirus Disease 2019 (COVID-19) Illness and Hospitalization—United States, February–September 2020. *Clinical Infectious Diseases* **72**, e1010-e1017 (2020).
39. Weidhaas J, *et al.* Correlation of SARS-CoV-2 RNA in wastewater with COVID-19 disease burden in sewersheds. *Science of The Total Environment* **775**, 145790 (2021).
40. Weidhaas J, *et al.* Correlation of SARS-CoV-2 RNA in wastewater with COVID-19 disease burden in sewersheds. *Science of the Total Environment* **775**, (2021).

41. Feng S, *et al.* Evaluation of Sampling, Analysis, and Normalization Methods for SARS-CoV-2 Concentrations in Wastewater to Assess COVID-19 Burdens in Wisconsin Communities. *ACS ES&T Water* **1**, 1955-1965 (2021).
42. Jiang G, *et al.* Artificial neural network-based estimation of COVID-19 case numbers and effective reproduction rate using wastewater-based epidemiology. *Water Research* **218**, 118451 (2022).
43. Li X, Zhang S, Shi J, Luby SP, Jiang G. Uncertainties in estimating SARS-CoV-2 prevalence by wastewater-based epidemiology. *Chemical engineering journal (Lausanne, Switzerland : 1996)* **415**, 129039 (2021).
44. Brown CC, Young SG, Pro GC. COVID-19 vaccination rates vary by community vulnerability: A county-level analysis. *Vaccine* **39**, 4245-4249 (2021).
45. Melvin SC, Wiggins C, Burse N, Thompson E, Monger M. The Role of Public Health in COVID-19 Emergency Response Efforts From a Rural Health Perspective. *Prev Chronic Dis* **17**, E70 (2020).
46. Tipirneni R, Schmidt H, Lantz PM, Karmakar M. Associations of 4 Geographic Social Vulnerability Indices With US COVID-19 Incidence and Mortality. *American Journal of Public Health* **112**, 1584-1588 (2022).
47. Acharya R, Porwal A. A vulnerability index for the management of and response to the COVID-19 epidemic in India: an ecological study. *The Lancet Global Health* **8**, e1142-e1151 (2020).
48. Karaye IM, Horney JA. The impact of social vulnerability on COVID-19 in the US: an analysis of spatially varying relationships. *American journal of preventive medicine* **59**, 317-325 (2020).
49. Ho TK. The random subspace method for constructing decision forests. *IEEE transactions on pattern analysis and machine intelligence* **20**, 832-844 (1998).
50. Breiman L. Random forests. *Machine learning* **45**, 5-32 (2001).
51. Kraemer MU, *et al.* The effect of human mobility and control measures on the COVID-19 epidemic in China. *Science* **368**, 493-497 (2020).
52. Klein B, *et al.* Forecasting hospital-level COVID-19 admissions using real-time mobility data. *Communications Medicine* **3**, 25 (2023).

53. Thomas KV, Amador A, Baz-Lomba JA, Reid M. Use of mobile device data to better estimate dynamic population size for wastewater-based epidemiology. *Environmental science & technology* **51**, 11363-11370 (2017).
54. Breiman L. Bagging predictors. *Machine learning* **24**, 123-140 (1996).
55. Brockwell PJ, Davis RA. *Introduction to time series and forecasting*. Springer (2002).
56. He X, *et al.* Temporal dynamics in viral shedding and transmissibility of COVID-19. *Nature medicine* **26**, 672-675 (2020).
57. MacIntyre CR, Costantino V, Trent M. Modelling of COVID-19 vaccination strategies and herd immunity, in scenarios of limited and full vaccine supply in NSW, Australia. *Vaccine* **40**, 2506-2513 (2022).
58. Public-health-England. SARS-CoV-2 variants of concern and variants under investigation in England. *Technical briefing* **28**, (2021).
59. Gocheva-Ilieva S, Ivanov A, Stoimenova-Minova M. Prediction of Daily Mean PM10 Concentrations Using Random Forest, CART Ensemble and Bagging Stacked by MARS. *Sustainability* **14**, 798 (2022).
60. Gupta S, Georgiou A, Sen S, Simon K, Karaca-Mandic P. US Trends in COVID-19–Associated Hospitalization and Mortality Rates Before and After Reopening Economies. *JAMA Health Forum* **2**, e211262-e211262 (2021).
61. Chiam T, *et al.* Hospital length of stay among COVID-19-positive patients. *J Clin Transl Res* **7**, 377-385 (2021).
62. Eimer J, *et al.* Tocilizumab shortens time on mechanical ventilation and length of hospital stay in patients with severe COVID-19: a retrospective cohort study. *J Intern Med* **289**, 434-436 (2021).
63. Mendes A, *et al.* Nutritional risk at hospital admission is associated with prolonged length of hospital stay in old patients with COVID-19. *Clinical Nutrition* **41**, 3085-3088 (2022).
64. Wang Z, *et al.* What are the risk factors of hospital length of stay in the novel coronavirus pneumonia (COVID-19) patients? A survival analysis in southwest China. *Plos one* **17**, e0261216 (2022).

65. King AA, Domenech de Cellès M, Magpantay FM, Rohani P. Avoidable errors in the modelling of outbreaks of emerging pathogens, with special reference to Ebola. *Proceedings of the Royal Society B: Biological Sciences* **282**, 20150347 (2015).
66. Cramer EY, *et al.* The United States COVID-19 Forecast Hub dataset. *Scientific Data* **9**, 462 (2022).
67. Whittaker R, *et al.* Length of hospital stay and risk of intensive care admission and in-hospital death among COVID-19 patients in Norway: a register-based cohort study comparing patients fully vaccinated with an mRNA vaccine to unvaccinated patients. *Clinical Microbiology and Infection* **28**, 871-878 (2022).
68. Archer E, Archer ME. Package 'rfPermute'. *Vienna: R Core Team*, (2016).
69. El-Shabasy RM, Nayel MA, Taher MM, Abdelmonem R, Shoueir KR. Three wave changes, new variant strains, and vaccination effect against COVID-19 pandemic. *International Journal of Biological Macromolecules*, (2022).
70. Lewnard JA, Hong VX, Patel MM, Kahn R, Lipsitch M, Tartof SY. Clinical outcomes associated with SARS-CoV-2 Omicron (B. 1.1. 529) variant and BA. 1/BA. 1.1 or BA. 2 subvariant infection in southern California. *Nature Medicine*, 1-1 (2022).
71. Tseng HF, *et al.* Effectiveness of mRNA-1273 vaccination against SARS-CoV-2 omicron subvariants BA. 1, BA. 2, BA. 2.12. 1, BA. 4, and BA. 5. *Nature Communications* **14**, 1-10 (2023).
72. Wang Q, *et al.* Antibody evasion by SARS-CoV-2 Omicron subvariants BA. 2.12. 1, BA. 4, & BA. 5. *Nature*, 1-3 (2022).
73. Qu P, *et al.* Neutralization of the SARS-CoV-2 omicron BA. 4/5 and BA. 2.12. 1 subvariants. *New England Journal of Medicine* **386**, 2526-2528 (2022).

REVIEWERS' COMMENTS

Reviewer #2 (Remarks to the Author):

The authors have done an exceptional job responding to my many questions and suggestions. I also found the responses to the other reviewers quite well done. This work is an important piece that highlights the utility of wastewater surveillance for public health benefit. I have no further questions or suggestions.

Reviewer #3 (Remarks to the Author):

Summary

In "Wastewater-based epidemiology predicts COVID-19-induced weekly new hospital admissions in over 150 USA counties" the authors have responded to a number of my previous concerns. I very much appreciate the effort the authors took in doing so, and I believe the manuscript has been greatly enhanced. I still think that the authors could improve the paper through concerted efforts of comparing the current results with other forecasting models and using proper scoring metrics like the weighted interval score (WIS). However, I believe the work to be publishable once two minor aspects are addressed (described below)

Minor comments

- The authors switched to using the facility-level hospital admission data set provided by HHS, but they have not described fully how they aggregated the various record-based indicators from facilities to the county-level. Further, data from many facilities on many weeks is not available as they hide hospital admission counts less than a certain number. It is important that the authors include some analysis describing the extent to which there is missing data in their analysis.
- I was not able to replicate the analysis as there was no data in the github repository. The authors should ensure that all components are contained in the linked repository before publication.

Dear Reviewers,

We are grateful for the constructive comments received from the editor and reviewers, which have helped us to further improve the quality and clarity of the manuscript.

We appreciate the opportunity to revise this manuscript and have carefully evaluated and addressed all comments and amended the manuscript accordingly. Manuscript ID: NCOMMS-22-51169A.

Below are our detailed responses to the Reviewers' comments point by point. The comments from the editor are in **black**, responses from the authors are in **blue**, and revisions to the manuscript are in **red**.

We would be happy to address any further comments that you or the reviewers might have.

Kind regards,

Prof. Qilin Wang

Centre for Technology in Water and Wastewater, School of Civil and Environmental Engineering, University of Technology Sydney, Ultimo, NSW, 2007, Australia

REVIEWERS' COMMENTS

Reviewer #2 (Remarks to the Author):

The authors have done an exceptional job responding to my many questions and suggestions. I also found the responses to the other reviewers quite well done. This work is an important piece that highlights the utility of wastewater surveillance for public health benefit. I have not further questions or suggestions.

We appreciate the positive feedback and recognition from the Reviewer 2.

Reviewer #3 (Remarks to the Author):

Summary

In “Wastewater-based epidemiology predicts COVID-19-induced weekly new hospital admissions in over 150 USA counties” the authors have responded to a number of my previous concerns. I very much appreciate the effort the authors took in doing so, and I believe the manuscript has been greatly enhanced. I still think that the authors could improve the paper through concerted efforts of comparing the current results with other forecasting models and using proper scoring metrics like the weighted interval score (WIS). However, I believe the work to be publishable once two minor aspects are addressed (described below)

We appreciate the positive feedback received from Reviewer 3. Reviewer 3 also suggest to improve our study through a comprehensive comparison of our current results with other forecasting models using proper scoring metrics like the weighted interval score (WIS). Indeed, such a comparison would be beneficial for evaluating the accuracy of our model. However, we would like to note that there is a lack of county-level predictions for hospitalization indicators in previous studies or prediction models. Previous predictions (and models) rely on the hospital admission records or cases to predict the daily (or weekly) new admissions at **state-¹** or **national-level** (COVID-19 Forecast Hub) ² in the USA. However, hospitalization patterns can differ significantly even within the same state, due to variations in factors such as population demographics and healthcare resources ^{3,4}.

In contrast, our study aims to predict hospitalization numbers at the county level using wastewater-based epidemiology (WBE), to facilitate targeted resource allocation.

Consequently, it was not possible for us to compare our prediction results with those of previous models for the same county. Instead, we addressed this limitation by developing county-level prediction models using hospital admission records or cases, which are the primary indicators employed in existing models. This allowed us to compare the accuracy of our WBE-based predictions with case-based or record-based predictions in our study.

Further, previous studies primarily reported normalized mean absolute error (NMAE)^{1,5} rather than the weighted interval score (WIS) in their results. To compare the accuracy of our study to previous studies, we thus adopted NMAE as the major evaluation parameters.

Our results showed that WBE-based models (NMAE=0.32-0.37) outperformed the case-based or record-based models for weekly new admission (our study, NMAE=0.38-0.45, leading time up to 4 weeks) and state/national-level predictions (previous studies, NMAE= 0.35-0.45, leading time of 2-3 weeks)^{1,5}. This demonstrates the feasibility and accuracy of our WBE-based model for predicting hospitalization admissions at county-level. To better clarify this, we have changed the following lines in the manuscript.

Line 48-50:

To date, the prediction of hospitalization admissions due to COVID-19 is majorly at the state or national level, relying on the confirmed COVID-19 cases or historical records of daily or weekly COVID-19-induced admissions as the key indicators^{6,7}.

Line 610-613:

The performance of the model was evaluated by the correlation coefficient (R), mean absolute error (MAE), and normalized mean absolute error (NMAE) using equations (1) and (2). These evaluation criteria, especially NMAE, have been widely used in previous prediction studies^{1,5}, facilitating inter-study comparisons.

Minor comments

- The authors switched to using the facility-level hospital admission data set provided by HHS, but they have not described fully how they aggregated the various record-based indicators from facilities to the county-level. Further, data from many facilities on many weeks is not available as they hide hospital admission counts less than a certain number. It is important that the authors include some analysis describing the extent to which there is missing data in their analysis.

We subdivided this comment into 2 to better address this.

- (a) They have not described fully how they aggregated the various record-based indicators from facilities to the county-level.

For our study, we obtained the hospital utilization data from official governmental records available on HealthData.gov, the designated website for United States government health data. This dataset consists of facility-level hospitalization records on a weekly basis, alongside the corresponding county where each facility is located. In order to obtain county-level admissions, we aggregated (summed) the records for facilities within the same week and county. To clarify this, we have added the following lines to the manuscript.

Line 524-525:

Briefly, facility-level data for hospital utilization in each county was reported on a weekly basis, along with the corresponding county where each facility is located.

Line 528-529:

The county-level values for each indicator were then obtained from the aggregation of facilities within the same week and county.

- (b) data from many facilities on many weeks is not available as they hide hospital admission counts less than a certain number. It is important that the authors include some analysis describing the extent to which there is missing data in their analysis.

Our study utilized the official governmental records available on HealthData.gov for the hospital admission count. This dataset is derived from reports with facility-level granularity across two main sources: (1) the Department of Health and Human Services (HHS) TeleTracking, and (2) reporting provided directly to HHS Protect by state/territorial health departments on behalf of their healthcare facilities. By combining data from these sources, the dataset from HealthData.gov ensured a comprehensive collection approach that encompassed various means, including electronic health record systems, manual reporting, and other data submission processes.

It is important to note that HHS TeleTracking also leverages data from state health departments, public health agencies, and other healthcare organizations to supplement and validate the data received directly from hospitals. This meticulous and extensive data collection process helps

ensure the thoroughness of the data collection and minimizes the possibility of intentional data concealment.

Indeed, there are some missing data in the dataset from HealthData.gov. When there are fewer than 4 patients in a data field, the cell is redacted and replaced with -999999 (COVID-19 Reported Patient Impact and Hospital Capacity by Facility | HealthData.gov). This value was chosen to ensure that users would not make the mistake of quickly “averaging” a column to conclude. To ensure the accuracy of the prediction, we removed such missing values from the data in our study. To clarify this, we have added the following lines.

Line 518-528:

The data for weekly new hospitalizations, census inpatient sum, and census inpatient average was retrieved from HealthData.gov. This dataset is derived from reports with facility-level granularity across two main sources: (1) the Department of Health and Human Services (HHS) TeleTracking, and (2) reporting provided directly to HHS Protect by state/territorial health departments on behalf of their healthcare facilities. By combining data from these sources, the dataset from HealthData.gov ensured a comprehensive and validated data collection. Briefly, facility-level data for hospital utilization in each county was reported on a weekly basis, along with the corresponding county where each facility is located. In the dataset, when there are fewer than 4 patients in a data field, the cell is redacted and replaced with -999999. To ensure the accuracy of the prediction, we removed such missing values from the data in our study.

- I was not able to replicate the analysis as there was no data in the github repository. The authors should ensure that all components are contained in the linked repository before publication.

The data used in this study are sourced from open accesses with links provided in the Data availability section. The authors are not permitted to share the third-party raw data used in the analysis. Secondary data (wastewater surveillance data and relevant weather, CCVI and hospitalization data) used in the analyses could be shared by contacting the corresponding authors upon reasonable request. To clarify this, we have added the following lines to the data availability section

Line 640-641:

Secondary data (wastewater surveillance data and relevant weather, CCVI, and hospitalization data) used in the analyses could be shared by contacting the corresponding authors upon reasonable request.

Reference

1. Fox SJ, *et al.* Real-time pandemic surveillance using hospital admissions and mobility data. *Proceedings of the National Academy of Sciences* **119**, e2111870119 (2022).
2. Cramer EY, *et al.* The United States COVID-19 Forecast Hub dataset. *Scientific Data* **9**, 462 (2022).
3. Mokhtari A, Mineo C, Kriseman J, Kremer P, Neal L, Larson J. A multi-method approach to modeling COVID-19 disease dynamics in the United States. *Scientific reports* **11**, 1-16 (2021).
4. Whittaker R, *et al.* Length of hospital stay and risk of intensive care admission and in-hospital death among COVID-19 patients in Norway: a register-based cohort study comparing patients fully vaccinated with an mRNA vaccine to unvaccinated patients. *Clinical Microbiology and Infection* **28**, 871-878 (2022).
5. Rosenfeld R, Tibshirani RJ. From the Cover: Beyond Cases and Deaths: The Benefits of Auxiliary Data Streams In Tracking the COVID-19 Pandemic: Epidemic tracking and forecasting: Lessons learned from a tumultuous year. *Proceedings of the National Academy of Sciences of the United States of America* **118**, (2021).
6. Deschepper M, Eeckloo K, Malfait S, Benoit D, Callens S, Vansteelandt S. Prediction of hospital bed capacity during the COVID-19 pandemic. *BMC Health Services Research* **21**, 468 (2021).
7. Ferstad JO, *et al.* A model to forecast regional demand for COVID-19 related hospital beds. *MedRxiv*, (2020).